# Proteína: Scaling Flow-based Protein Structure Generative Models

Tomas Geffner[1,*]      Kieran Didi[1,*]      Zuobai Zhang[1,2,3,†,*]      Danny Reidenbach[1]

Zhonglin Cao[1]      Jason Yim[1,4]      Mario Geiger[1]      Christian Dallago[1]

Emine Kucukbenli[1]      Arash Vahdat[1]      Karsten Kreis[1,*]

[1]NVIDIA  [2]Mila - Québec AI Institute  [3]Université de Montréal  [4]Massachusetts Institute of Technology

*Project page:* https://research.nvidia.com/labs/genair/proteina/

## Abstract

Recently, diffusion- and flow-based generative models of protein structures have emerged as a powerful tool for de novo protein design. Here, we develop *Proteína*, a new large-scale flow-based protein backbone generator that utilizes hierarchical fold class labels for conditioning and relies on a tailored scalable transformer architecture with up to $5\times$ as many parameters as previous models. To meaningfully quantify performance, we introduce a new set of metrics that directly measure the distributional similarity of generated proteins with reference sets, complementing existing metrics. We further explore scaling training data to millions of synthetic protein structures and explore improved training and sampling recipes adapted to protein backbone generation. This includes fine-tuning strategies like LoRA for protein backbones, new guidance methods like classifier-free guidance and autoguidance for protein backbones, and new adjusted training objectives. Proteína achieves state-of-the-art performance on de novo protein backbone design and produces diverse and designable proteins at unprecedented length, up to 800 residues. The hierarchical conditioning offers novel control, enabling high-level secondary-structure guidance as well as low-level fold-specific generation.

## 1 Introduction

De novo protein design, the rational design of new proteins from scratch with specific functions and properties, is a grand challenge in molecular biology (Richardson & Richardson, 1989; Huang et al., 2016; Kuhlman & Bradley, 2019). Recently, deep generative models emerged as a novel data-driven tool for protein design. Since a protein's function is mediated through its structure, a popular approach is to directly model the distribution of three-dimensional protein structures (Ingraham et al., 2023; Watson et al., 2023; Yim et al., 2023b; Bose et al., 2024; Lin & Alquraishi, 2023), typically with diffusion- or flow-based methods (Ho et al., 2020; Lipman et al., 2023). Such protein structure generators usually synthesize backbones only, without sequence or side chains, in contrast to protein language models, which often model sequences instead (Elnaggar et al., 2022; Lin et al., 2023; Alamdari et al., 2023), and sequence-to-structure folding models like AlphaFold (Jumper et al., 2021).

Previous unconditional protein structure generative models have only been trained on small datasets, consisting of no more than half a million structures at maximum (Lin et al., 2024). Moreover, their neural networks do not offer any control during synthesis and are usually small, compared to modern generative AI systems in domains such as natural language, image or video generation. There, we have witnessed major breakthroughs thanks to scalable neural network architectures, large training datasets, and fine semantic control (Esser et al., 2024; Brooks et al., 2024; OpenAI, 2024). This begs the question: can we similarly scale and control protein structure diffusion and flow models, taking lessons from the recent successes of generative models in computer vision and natural language?

Here, we set out to scale protein structure generation and develop a new flow matching-based protein backbone generative model called *Proteína*. In vision and language modeling, generative models are typically prompted through semantic text or class inputs, offering enhanced controllability. Analogously, we enrich our training data with hierarchical fold class labels following the CATH Protein

---

*Core contributor.
†Work done during internship at NVIDIA.

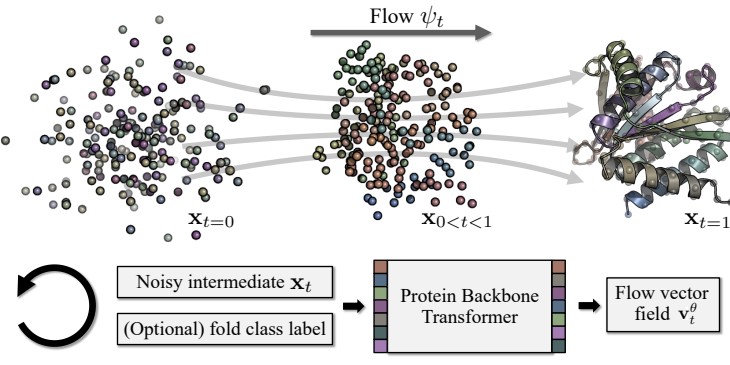

Figure 1: **Proteína.** We use flow-matching and learn a flow to transform a Gaussian distribution over initial protein backbone coordinates (residues' $C_\alpha$ atoms) into realistic protein structures. Proteína relies on a scalable transformer-based architecture and can be conditioned on hierarchical fold class labels for improved controllability and complex protein structure design tasks.

Structure Classification scheme (Dawson et al., 2016). Our novel hierarchical fold class conditioning offers both high-level control, for instance over secondary structure content, as well as low-level guidance with respect to specific fold classes. This can be leveraged, for instance, to dramatically increase the number of $\beta$-sheets in generated proteins. We also explore scaling the training data and train on up to 21 million protein structures, a $35\times$ increase of training data compared to previous work.

Next, we develop a scalable transformer architecture. We opt for a non-equivariant design inspired by recent diffusion transformers in vision (Peebles & Xie, 2023; Ma et al., 2024). For boosted performance, we optionally include triangle layers, a powerful albeit computationally expensive network component common in the protein literature (Jumper et al., 2021). Crucially, though, our models also achieves top performance without any triangle layer-based pair representation updates. This allows us to train on very large proteins and generate backbones of up to 800 residues, while maintaining designability and diversity, significantly outperforming all previous works. Further, while non-equivariant diffusion models have recently been used as part of AlphaFold3 (Abramson et al., 2024), equivariant methods have been dominant in the unconditional protein structure generation literature. We show that large-scale non-equivariant flow models also succeed on unconditional protein structure generation. We train versions of Proteína with more than 400M parameters, more than $5\times$ larger than RFDiffusion (Watson et al., 2023), to the best of our knowledge the largest existing protein backbone generator.

Protein structure generators are typically evaluated based on their generated proteins' diversity, novelty and designability (see Sec. 3.5). However, none of these metrics rigorously scores models at the distribution level, although the task of generative modeling is to learn a model of a data distribution. Hence, we introduce new metrics that directly score the learnt distribution instead of individual samples. Similar to the Fréchet Inception Distance in image generation (Heusel et al., 2017), we compare sets of generated samples against reference distributions in a non-linear feature space. Since our feature extractor is based on a fold class predictor, we further quantify models' diversity over fold classes as well as the similarity of the generated class distribution compared to reference data's classes.

Further, we adjust the flow matching objective to protein structure generation and explore stage-wise training strategies. For instance, using low-rank adaptation (LoRA, Hu et al. (2022)) we fine-tune Proteína models on natural, designable proteins. We also develop novel guidance schemes for hierarchical fold class conditioning and successfully showcase autoguidance (Karras et al., 2024) to enhance protein designability. Experimentally, Proteína achieves state-of-the-art protein backbone generation performance, vastly outperforming all baselines especially in long chain synthesis, and we demonstrate superior control compared to previous models through our novel fold class conditioning.

**Main contributions**: *(i)* We present Proteína, a novel flow-based generative protein structure foundation model using a new scalable non-equivariant transformer architecture, which we scale to more than 400M parameters. *(ii)* We incorporate hierarchical fold class conditioning into Proteína and develop tailored training algorithms and guidance schemes, leading to unprecedented semantic controllability over protein structure generation. In particular, we showcase fold-specific synthesis as well as a controlled enhancement of $\beta$-sheets in generated structures. *(iii)* We introduce several new protein structure generation metrics to complement existing metrics and to better analyze and compare existing models. *(iv)* We scale training data to almost 21M high-quality synthetic protein structures, and show successful training of models with very high designability on such large data. *(v)* We achieve state-of-the-art designable and diverse protein backbone generation performance for unconditional and fold class-conditional generation as well as motif-scaffolding. Thanks to our efficient transformer architecture, we scale to an unprecedented length of 800 residues, still producing diverse and designable proteins, vastly outperforming previous works. *(vi)* For the first time, we demonstrate LoRA-based fine-tuning and autoguidance for flow-based protein structure generative models.

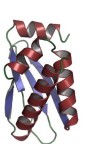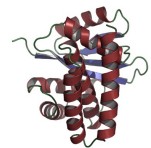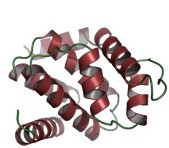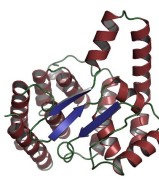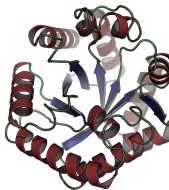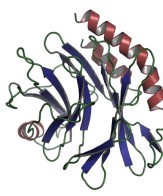

Figure 2: **Proteína Samples.** Designable backbones generated unconditionally by $\mathcal{M}_{\text{FS}}$ model ($<$250 residues).

## 2 BACKGROUND AND RELATED WORK

Proteína relies on flow-matching (Lipman et al., 2023; Liu et al., 2023; Albergo & Vanden-Eijnden, 2023), which models a probability density path $p_t(\mathbf{x}_t)$ that gradually transforms an analytically tractable noise distribution ($p_{t=0}$) into a data distribution ($p_{t=1}$), following a time variable $t \in [0, 1]$. Formally, the path $p_t(\mathbf{x}_t)$ corresponds to a *flow* $\psi_t$ that pushes samples from $p_0$ to $p_t$ via $p_t = [\psi]_t * p_0$, where $*$ denotes the push-forward. In practice, the flow is modelled via an ordinary differential equation (ODE) $d\mathbf{x}_t = \mathbf{v}_t^\theta(\mathbf{x}_t, t)dt$, defined through a learnable vector field $\mathbf{v}_t^\theta(\mathbf{x}_t, t)$ with parameters $\theta$. Initialized from noise $\mathbf{x}_0 \sim p_0(\mathbf{x}_0)$, this ODE simulates the flow and transforms noise into approximate data distribution samples. The probability density path $p_t(\mathbf{x}_t)$ and the (intractable) ground-truth vector field $\mathbf{u}_t(\mathbf{x}_t)$ are related via the continuity equation $\partial p_t(\mathbf{x}_t)/\partial t = -\nabla_{\mathbf{x}_t} \cdot (p_t(\mathbf{x}_t)\mathbf{u}_t(\mathbf{x}_t))$.

To learn $\mathbf{v}_t^\theta(\mathbf{x}_t, t)$ one can employ conditional flow matching (CFM). In CFM, conditioned on data samples $\mathbf{x}_1 \sim p_1(\mathbf{x}_1)$, we construct *conditional* probability paths $p_t(\mathbf{x}_t|\mathbf{x}_1)$ for which the corresponding ground-truth *conditional* vector field $\mathbf{u}_t(\mathbf{x}_t|\mathbf{x}_1)$ is analytically tractable for simple distributions $p_0(\mathbf{x}_0)$, like Gaussian noise. The CFM objective then corresponds to regressing the neural network-defined approximate vector field $\mathbf{v}_t^\theta(\mathbf{x}_t, t)$ against $\mathbf{u}_t(\mathbf{x}_t|\mathbf{x}_1)$, where the intermediate samples $\mathbf{x}_t$ are drawn from the tractable conditional probability path $p_t(\mathbf{x}_t|\mathbf{x}_1)$ and we marginalize over data $\mathbf{x}_1$ via Monte Carlo sampling. Since in expectation the CFM objective results in the same gradients as directly regressing against the intractable marginal ground-truth vector field $\mathbf{u}_t(\mathbf{x}_t)$, $\mathbf{v}_t^\theta(\mathbf{x}_t, t)$ learns an approximation of the ground-truth $\mathbf{u}_t(\mathbf{x}_t)$.

In practice, the conditional probability paths are defined through an *interpolant* that connects noise $\mathbf{x}_0$ and data samples $\mathbf{x}_1$ and constructs intermediate $\mathbf{x}_t$ via interpolation. We rely on the rectified flow (Liu et al., 2023) (also known as conditional optimal transport (Lipman et al., 2023)) formulation, using a linear interpolant $\mathbf{x}_t = t\mathbf{x}_1 + (1 - t)\mathbf{x}_0$ and the regression target $d\psi_t(\mathbf{x}_0|\mathbf{x}_1)/dt = \mathbf{x}_1 - \mathbf{x}_0$. See Sec. 3.2 for our exact instantiation of the CFM objective. Flow-matching is related to diffusion models (Sohl-Dickstein et al., 2015; Ho et al., 2020; Song et al., 2021), see App. J; for Gaussian flows the frameworks become equivalent up to reparametrizations (Kingma & Gao, 2023; Albergo et al., 2023).

**Related Work.** Two seminal works on protein backbone generation with diffusion models are Chroma (Ingraham et al., 2023) and RFDiffusion (Watson et al., 2023), the latter fine-tuning RoseTTAFold (Baek et al., 2021). FrameDiff (Yim et al., 2023b) performs frame-based (Jumper et al., 2021) Riemannian manifold diffusion (Huang et al., 2022; Bortoli et al., 2022) to model residue rotations. These works were followed by FoldFlow (Bose et al., 2024) and FrameFlow (Yim et al., 2023a), leveraging Riemanning flow matching (Chen & Lipman, 2024). Meanwhile, Genie (Lin & Alquraishi, 2023) and others (Trippe et al., 2023) generate protein backbones diffusing only residues' $C_\alpha$ coordinates. Proteus (Wang et al., 2024) builds on top of FrameDiff, introducing efficient triangle layers. Recently, FoldFlow2 (Huguet et al., 2024) and Genie2 (Lin et al., 2024) extended training data to the AFDB, although with significantly less data than Proteína. MultiFlow (Campbell et al., 2024), building on FrameFlow, jointly generates sequence and structure. Related, Protpardelle (Chu et al., 2024) and the concurrent Pallatom (Qu et al., 2024) generate fully atomistic proteins. The latter uses a similar non-equivariant transformer architecture like AlphaFold3 (Abramson et al., 2024), also related to Proteína's architecture. Meanwhile, masked language models have been trained on structure tokens, with ESM3 (Hayes et al., 2024) being the most recent and prominent model. Chroma showed classifier-based guidance with respect to fold classes. In contrast, we, for the first time, leverage classifier-free guidance using large fold class annotations, and perform thorough quantitative analyses.

## 3 PROTEÍNA

### 3.1 SCALING PROTEIN STRUCTURE TRAINING DATA WITH FOLD CLASSES

Most protein structure generators have been trained on natural proteins, using filtered subsets of the PDB (Berman et al., 2000), resulting in training set sizes in the order of 20k. Recently, some works (Lin et al., 2024; Huguet et al., 2024; Qu et al., 2024) relied on the AFDB (Varadi et al., 2021) and in-

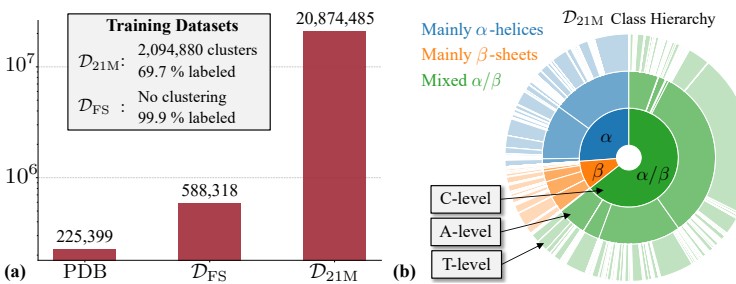

Figure 3: **Dataset Statistics.** *(a)* Dataset size comparisons. *(b)* Sunburst plot of the hierarchical fold class labels in our largest dataset $\mathcal{D}_{21M}$, depicting the hierarchical label structure and the relative sizes of the three hierarchical fold class categories C, A, and T.

cluded synthetic AlphaFold2 structures (Jumper et al., 2021). Genie2 (Lin et al., 2024) used the largest dataset, i.e. ≈0.6M synthetic structures. Inspired by the data scaling success of generative models in areas such as image and video generation and natural language synthesis (Brooks et al., 2024; Esser et al., 2024; OpenAI, 2024), we explore scaling protein structure training data even further. The entire AFDB extends to ≈214M structures, orders of magnitude larger than its small subsets used in previous works. However, not all of these structures are useful for training protein structure generators, as they contain low-quality predictions and other unsuitable data. Our main Proteína models are trained on two datasets, denoted as $\mathcal{D}_{FS}$ and $\mathcal{D}_{21M}$, the latter newly created (data processing details in App. M):

**1. Foldseek AFDB clusters $\mathcal{D}_{FS}$:** This dataset corresponds to the data that was also used by Genie2, based on sequential filtering and clustering of the AFDB with the sequence-based MMseqs2 and the structure-based Foldseek (van Kempen et al., 2024; Barrio-Hernandez et al., 2023). This data uses *cluster representatives only*, i.e. only one structure per cluster. Like Genie2, we use protein lengths between 32 and 256 residues in our main models, leading to 588,318 structures in total.

**2. High-quality filtered AFDB subset $\mathcal{D}_{21M}$:** We filtered all ≈214M AFDB structures for proteins with max. residue length 256, min. average pLDDT of 85, max. pLDDT standard deviation of 15, max. coil percentage of 50%, and max. radius of gyration of 3nm. This led to 20,874,485 structures. We further clustered the data with MMseqs2 (Steinegger & Söding, 2017) using a 50% sequence similarity threshold. During training, we sample clusters uniformly, and draw random structures within.

We use $\mathcal{D}_{FS}$, as, to the best of our knowledge, it represents the largest training dataset used in any previous flow- or diffusion-based structure generators. With $\mathcal{D}_{21M}$ we are pushing the frontier of training data scale for protein structure generation. In fact, $\mathcal{D}_{21M}$ is 35× larger than $\mathcal{D}_{FS}$ (see Fig. 3).

**Hierarchical fold class annotations.** Large-scale generative models in the visual domain typically rely on semantic class- or text-conditioning to offer control or to effectively break down the generative modeling task into a set of simpler conditional tasks (Bao et al., 2022). However, existing protein structure diffusion or flow models are either trained unconditionally, or condition only on partially given local structures, for instance in motif scaffolding tasks (Yim et al., 2024; Lin et al., 2024).

We propose, for the first time, to instead leverage fold class annotations that globally describe protein structures, akin to semantic class or text labels of images. We use *The Encyclopedia of Domains (TED)* data, which consists of structural domain assignments to proteins in the AFDB (Lau et al., 2024b;a). TED uses the CATH structural hierarchy (Dawson et al., 2016) to assign labels, where *C* ("class") describes the overall secondary-structure content of a domain, *A* ("architecture") groups domains with high structural similarity, *T* ("topology/fold") further refines the structure groupings, and *H* ("homol-

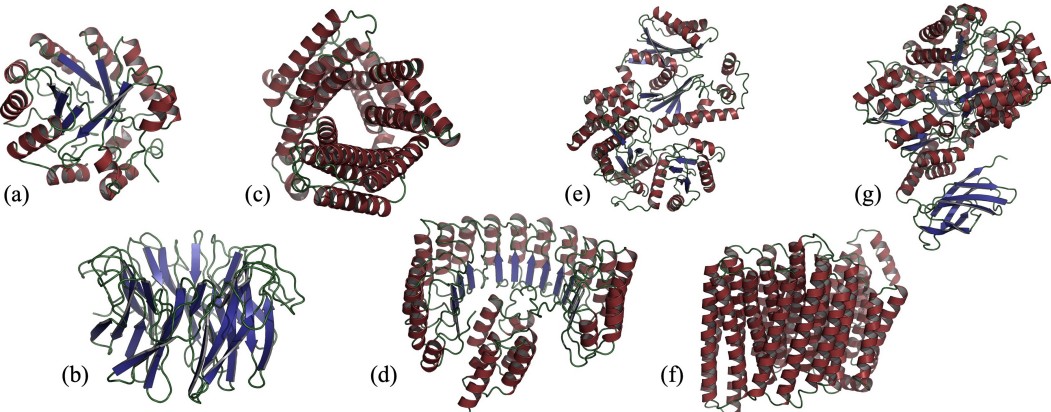

Figure 4: **Long Proteína Samples**. Chain lengths in (a)-(g): [300, 400, 500, 600, 700, 800, 800]. (a) "Mixed $\alpha/\beta$"-guided. (b) "Mainly $\beta$"-guided. (e) "Mixed $\alpha/\beta$"-guided. Others unconditional. All samples designable.

ogous superfamily") labels are only shared between domains with evolutionary relationships. Since we are strictly interested in structural modeling, we discard the H level and leverage only C, A, and T level labels. We assign labels to the proteins in all datasets, but since TED annotated not all of AFDB, some structures lack CAT labels. Moreover, some labels are less common than others (see Fig. 3); we only consider the main "mainly $\alpha$", "mainly $\beta$", and "mixed $\alpha/\beta$" C classes. See App. M for details.

## 3.2 TRAINING OBJECTIVE

We model protein backbones' residue locations through their $C_\alpha$ atom coordinates, similar to Lin & Alquraishi (2023); Lin et al. (2024). Note that many works instead leverage so-called frames (Jumper et al., 2021), additionally capturing residue rotations. However, this requires modeling a generative process over Riemannian rotation manifolds as well as ad hoc modifications to the rotation generation schedule during inference, which are not well understood (Yim et al., 2023a; Bose et al., 2024; Huguet et al., 2024). We purposely avoid such representations to not make the framework unnecessarily complicated, and prioritize simplicity and scalability, relying purely on $C_\alpha$ backbone coordinates.

Consider the vector of a protein backbone's 3D $C_\alpha$ coordinates $\mathbf{x} \in \mathbb{R}^{3L}$, where $L$ is the number of residues. Denote the protein's fold class labels as $\{C_\mathbf{x}, A_\mathbf{x}, T_\mathbf{x}\}_{\text{CAT}}$, and the binned pairwise distance between residues $i$ and $j$ as $D_{b,ij}(\mathbf{x})$. Using $\mathbf{x}_t = t\mathbf{x} + (1-t)\boldsymbol{\epsilon}$, Proteína's objective then is

$$
\min_\theta \mathbb{E}_{\mathbf{x}\sim p_\mathcal{D}(\mathbf{x}), \boldsymbol{\epsilon}\sim\mathcal{N}(\mathbf{0},\boldsymbol{I}), t\sim p(t)} \left[ \frac{1}{L} \underbrace{||\mathbf{v}_t^\theta(\mathbf{x}_t, t, \hat{\mathbf{x}}(\mathbf{x}_t), \{C_\mathbf{x}, A_\mathbf{x}, T_\mathbf{x}\}_{\text{CAT}}) - (\mathbf{x} - \boldsymbol{\epsilon})||_2^2}_{\text{Main conditional flow-matching loss, see Sec. 2.}} \right.
$$
$$
\left. - \frac{\mathbf{1}(t \geq 0.3)}{L^2} \sum_{i,j} \sum_{b=1}^{64} \underbrace{D_{b,ij}(\mathbf{x}) \log p_{b,ij}^\theta(\mathbf{x}_t, t, \hat{\mathbf{x}}(\mathbf{x}_t), \{C_\mathbf{x}, A_\mathbf{x}, T_\mathbf{x}\}_{\text{CAT}})}_{\text{Optional auxiliary binned distogram loss.}} \right]. \tag{1}
$$

Similar to Abramson et al. (2024); Qu et al. (2024), we optionally include a cross entropy-based distogram loss, which discretizes pairwise residue distances into 64 bins. The distogram is predicted via a prediction head attached to our architecture's pair representation and only used if this pair representation is updated (see Sec. 3.3). This loss is generally used only for $t \geq 0.3$. We also train for self-conditioning, conditioning the model on its own clean data prediction $\hat{\mathbf{x}}(\mathbf{x}_t) = \mathbf{x}_t + (1 - t)\mathbf{v}_t^\theta(\mathbf{x}_t, t, \emptyset, \{C_\mathbf{x}, A_\mathbf{x}, T_\mathbf{x}\}_{\text{CAT}})$ with probability 0.5. Furthermore, we design a novel $t$-sampling distribution, $p(t) = 0.02\,\mathcal{U}(0, 1) + 0.98\,\mathcal{B}(1.9, 1.0)$, tailored to flow matching for protein backbone generation (motivation and discussion in App. K, visualization in Fig. 20, ablation studies in App. L).

**Fold-class conditioning.** Our fold class labels describe protein structures at different levels of detail, and we seek the ability to both condition on varying levels of the hierarchy, and to also run the model unconditionally. To this end, we propose to hierarchically drop out different label combinations during training. Specifically, with $p = 0.5$ we drop all labels ($\{\emptyset, \emptyset, \emptyset\}_{\text{CAT}}$), with $p = 0.1$ we only show the $C$ label ($\{C_\mathbf{x}, \emptyset, \emptyset\}_{\text{CAT}}$), with $p = 0.15$ we drop only the $T$ label ($\{C_\mathbf{x}, A_\mathbf{x}, \emptyset\}_{\text{CAT}}$) and with $p = 0.25$ we give the model all labels ($\{C_\mathbf{x}, A_\mathbf{x}, T_\mathbf{x}\}_{\text{CAT}}$). The drop probabilities are chosen such that, on the one hand, we learn a strong unconditional model without any labels. On the other hand, the number of categories increases along the hierarchy, such that we focus training more on the increasingly fine A and T classes, as opposed to conditioning only on the coarser C labels (Fig. 3). Moreover, our approach enables classifier-free guidance (Ho & Salimans, 2021) for all possible levels during inference, combining the unconditional model prediction with any of the label-conditioned predictions (guidance weight $\omega$, see App. I). Note that, while most training proteins have only a single label, if a protein has multiple domains and corresponding hierarchical labels, we randomly feed one of them to the model.

## 3.3 A SCALABLE PROTEIN STRUCTURE TRANSFORMER ARCHITECTURE

While previous protein structure generators typically use small equivariant neural networks, we take inspiration from language and image generation (Peebles & Xie, 2023; Ma et al., 2024; Esser et al., 2024) and design a new streamlined non-equivariant transformer, see Fig. 5. It constructs residue chain and pair representations from the (noisy) protein coordinates, the residue indices, the sequence separation between residues and the (optional) self-conditioning input. The residue chain representation is processed by a stack of conditioned and biased multi-head self-attention layers (Vaswani et al., 2017), using a pair bias via the pair representation, which can be optionally updated, too. At the end, the updated sequence representation is decoded into the vector field prediction $\mathbf{v}_t^\theta$ to model Proteína's flow.

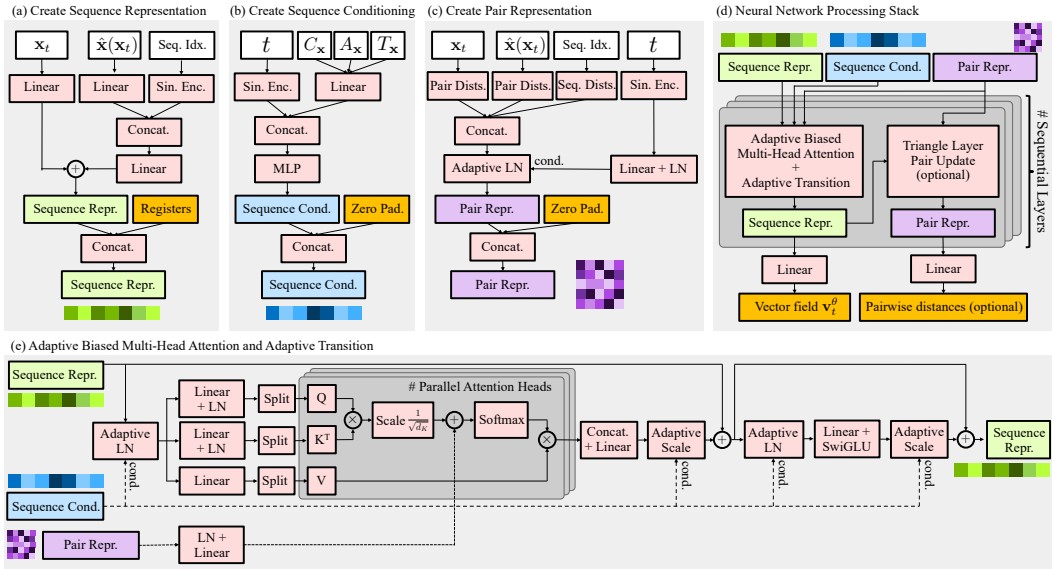

Figure 5: **Proteína's transformer architecture.** *(a)-(c)* We first create a sequence representation, sequence conditioning features, and a pair representation. *(d)* They are processed by conditioned and biased (through the pair representation) multi-head attention layers, described in *(e)*. We use a variant of QK normalization, applying LayerNorm (LN) to the Q and K inputs to the attention operation, before the multi-head split. Optionally, the pair representation can be updated. See App. N for the *Pair Update*, *Adaptive LN*, and *Adaptive Scale* modules.

A related architecture has recently been introduced by AlphaFold3 (Abramson et al., 2024), and is used concurrently in Pallatom (Qu et al., 2024). Our design features some additional components: *(i)* As discussed, we condition on hierarchical fold class labels. They are fed to the model through concatenated learnable embeddings, injected into the attention stack via adaptive layer norms, together with the $t$ embedding. *(ii)* Following best practices from language and vision, we extend our sequence representation with auxiliary tokens, known as registers (Darcet et al., 2024), which can capture global information or act as attention sinks (Xiao et al., 2024) and streamline the sequence processing. *(iii)* We use a variant of QK normalization (Dehghani et al., 2023) to avoid uncontrolled attention logit growth. While our models are smaller than the large models in vision and language, we train with relatively small batch sizes and high learning rates, where similar instabilities can occur (Wortsman et al., 2024). *(iv)* All our attention layers feature residual connections—without, we were not able to train stably (AlphaFold3 is ambiguous regarding their use of such residuals). *(v)* We use triangle multiplicative layers (Jumper et al., 2021) as *optional add-on only* to update the pair representation. While triangle layers have been shown to boost performance (Jumper et al., 2021; Lin et al., 2024; Huguet et al., 2024), they are highly compute and memory intensive, limiting scalability. Hence, in Proteína we avoid their usage as the driving model component and carry out most processing with the main transformer stack.

AlphaFold3 showed that non-equivariant diffusion models can succeed in protein folding, but they rely on expressive amino acid sequence and MSA embeddings. We instead learn the distribution of protein structures *without sequence inputs*. For this task, to the best of our knowledge, almost all related works used equivariant architectures, aside from the concurrent Pallatom (Qu et al., 2024) and Prot-pardelle (Chu et al., 2024). To nonetheless *learn* equivariance, we center training proteins and augment with random rotations; in App. E we show that our model learns an approximately SO(3)-equivariant vector field. We train models with up to ≈400M parameters in the transformer and ≈17M in the triangle layers, which, we believe, represents the largest protein structure flow or diffusion model.

## 3.4 SAMPLING

New protein backbones can be generated with Proteína by simulating the learnt flow's ODE, see Sec. 2. Since our flow is Gaussian, there exists a connection between the learnt vector field and the corresponding score $s(\mathbf{x}_t) := \nabla_{\mathbf{x}_t} \log p_t(\mathbf{x}_t)$ (Albergo et al., 2023; Ma et al., 2024),

$$\mathbf{s}_t^\theta(\mathbf{x}_t, \tilde{c}) = \frac{t\mathbf{v}_t^\theta(\mathbf{x}_t, \tilde{c}) - \mathbf{x}_t}{1 - t}, \tag{2}$$

where we use $\tilde{c}$ as abbreviation for all conditioning inputs (see Sec. 3.2). This allows us to construct a stochastic differential equation (SDE) that can be used as a stochastic alternative to sample Proteína,

$$\mathrm{d}\mathbf{x}_t = \mathbf{v}_t^\theta(\mathbf{x}_t, \tilde{c})\mathrm{d}t + g(t)\mathbf{s}_t^\theta(\mathbf{x}_t, \tilde{c})\mathrm{d}t + \sqrt{2g(t)\gamma}\,\mathrm{d}\mathcal{W}_t, \tag{3}$$

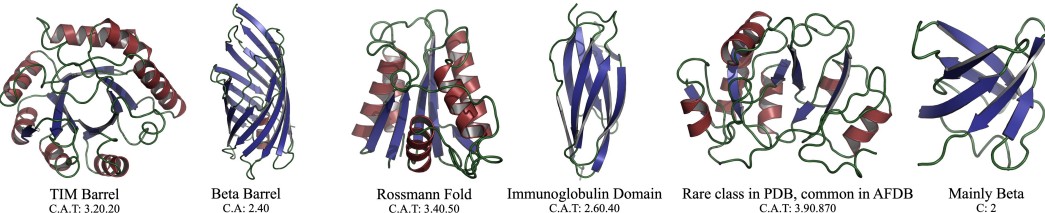

TIM Barrel — C.A.T: 3.20.20    Beta Barrel — C.A: 2.40    Rossmann Fold — C.A.T: 3.40.50    Immunoglobulin Domain — C.A.T: 2.60.40    Rare class in PDB, common in AFDB — C.A.T: 3.90.870    Mainly Beta — C: 2

Figure 6: **Fold class-conditional Generation** with $\mathcal{M}_{\text{FS}}^{\text{cond}}$ model (Sec. 4.3). All samples are designable and correctly re-classified (App. D). The used C.A.T fold class conditioning codes are given below fold names.

where $\mathcal{W}_t$ is a Wiener process and $g(t)$ scales the additional score and noise terms, which corresponds to Langevin dynamics (Karras et al., 2022). Crucially, we have introduced a noise scaling parameter $\gamma$. For $\gamma=1$, the SDE has the same marginals and hence samples from the same distribution as the ODE (Karras et al., 2022; Ma et al., 2024). However, it is common in the protein structure generation literature to reduce the noise scale in stochastic sampling (Ingraham et al., 2023; Wang et al., 2024; Lin et al., 2024). This is not a principled way to reduce the temperature of the sampled distribution (Du et al., 2023), but can be beneficial empirically, often improving designability at the cost of diversity. Fold label conditioning is done via classifier-free guidance (CFG) (Ho & Salimans, 2021), and we also explore *autoguidance* (Karras et al., 2024), where a model is guided using a "bad" version of itself. In a unifying formulation, we can write the guided vector field as

$$\mathbf{v}_t^{\theta,\text{guided}}(\mathbf{x}_t, \tilde{c}) = \omega\, \mathbf{v}_t^{\theta}(\mathbf{x}_t, \tilde{c}) + (1-\omega) \left[(1-\alpha)\mathbf{v}_t^{\theta}(\mathbf{x}_t, \emptyset) + \alpha\, \mathbf{v}_t^{\theta,\text{bad}}(\mathbf{x}_t, \tilde{c})\right] \tag{4}$$

where $\omega \geq 0$ defines the overall guidance weight and $\alpha \in [0, 1]$ interpolates between CFG and autoguidance. An analogous equation holds for the scores $\mathbf{s}_t^{\theta}(\mathbf{x}_t, \tilde{c})$. To the best of our knowledge, no previous works explore CFG or autoguidance for protein structure generation. More details in App. I.

### 3.5 PROBABILISTIC METRICS FOR PROTEIN STRUCTURE GENERATIVE MODELS

Protein structure generators are scored based on their samples' *designability*, *diversity* and *novelty* (see App. F). However, designability relies on auxiliary models, ProteinMPNN (Dauparas et al., 2022) and ESMFold (Lin et al., 2023), with their own biases. Moreover, we cannot necessarily expect to maximize designability by learning a better generative model, because not even all training proteins are designable (Lin et al., 2024). Next, diversity and novelty are usually only computed among designable samples, which makes them dependent on the complex designability metric, and diversity and novelty do otherwise not depend on quality. Therefore, we propose new probabilistic metrics that offer complementary insights. We suggest to more directly quantify how well a model matches a relevant reference distribution. Specifically, we first train a fold class predictor $p_\phi(\cdot|\mathbf{x})$ with features $\phi(\mathbf{x})$ for all CAT hierarchy levels (Sec. 3.1). Leveraging this classifier, we propose three new metrics:

**Fréchet Protein Structure Distance (FPSD).** Inspired by the FID score (Heusel et al., 2017), we embed generated and reference structures into the feature space of the fold class predictor and measure the Wasserstein distance between the feature distributions, modeling them as Gaussians. Defining the generated and the reference set of protein structures as $\{\mathbf{x}\}_{\text{gen}}$ and $\{\mathbf{x}\}_{\text{ref}}$, respectively, we have

$$\text{FPSD}(\{\mathbf{x}\}_{\text{gen}}, \{\mathbf{x}\}_{\text{ref}}) := ||\boldsymbol{\mu}_{\{\phi(\mathbf{x})\}_{\text{gen}}} - \boldsymbol{\mu}_{\{\phi(\mathbf{x})\}_{\text{ref}}}||_2^2 + \text{tr}\left(\Sigma_{\{\phi(\mathbf{x})\}_{\text{gen}}} + \Sigma_{\{\phi(\mathbf{x})\}_{\text{ref}}} - 2(\Sigma_{\{\phi(\mathbf{x})\}_{\text{gen}}}\Sigma_{\{\phi(\mathbf{x})\}_{\text{ref}}})^{\frac{1}{2}}\right).$$

An accurate fold class predictor must learn an expressive feature representation of protein structures. Hence, we argue that these feature embeddings must be well-suited for fine-grained reasoning about protein structure distributions, making a fold class predictor an ideal choice as embedding model.

**Fold Jensen Shannon Divergence (fJSD).** We also directly compare the marginal predicted categorical fold class distributions of generated and reference structures via the Jensen Shannon Divergence,

$$\text{fJSD}(\{\mathbf{x}\}_{\text{gen}}, \{\mathbf{x}\}_{\text{ref}}) := 10 \times \text{JSD}(\mathbb{E}_{\mathbf{x}\sim\{\mathbf{x}\}_{\text{gen}}}p_\phi(\cdot|\mathbf{x})||\mathbb{E}_{\mathbf{x}\sim\{\mathbf{x}\}_{\text{ref}}}p_\phi(\cdot|\mathbf{x})).$$

Note that we can evaluate this fJSD metric at all levels of the predicted CAT fold class hierarchy, allowing us to measure distributional fold class similarity at different levels of granularity. In practice, in this work we report the average over all levels in the interest of conciseness.

**Fold Score (fS).** Inspired by the Inception Score (Salimans et al., 2016), we propose a Fold Score

$$\text{fS}(\{\mathbf{x}\}_{\text{gen}}) := \exp\left(\mathbb{E}_{\mathbf{x}\sim\{\mathbf{x}\}_{\text{gen}}}\left[D_{\text{KL}}\left(p_\phi(\cdot|\mathbf{x})||\mathbb{E}_{\mathbf{x}\sim\{\mathbf{x}\}_{\text{gen}}}p_\phi(\cdot|\mathbf{x})\right)\right]\right).$$

Table 1: Proteína's **unconditional backbone generation performance** compared to baselines. *All models and baselines tuned for designability via noise scaling or inference rotation annealing, not sampling full distribution.* For metric evaluation details see App. F and App. G. Best scores **bold**, second best underlined.

| Model | Design-ability (%)↑ | Diversity Cluster↑ | TM-Sc.↓ | Novelty vs. PDB↓ | AFDB↓ | FPSD vs. PDB↓ | AFDB↓ | fS (C / A / T)↑ | fJSD vs. PDB↓ | AFDB↓ | Sec. Struct. % (α / β ) |
|---|---|---|---|---|---|---|---|---|---|---|---|
| *Unconditional generation. $\mathcal{M}_i^j$ denotes the Proteína model variant, and $\gamma$ is the noise scale for Proteína.* | | | | | | | | | | | |
| FrameDiff | 65.4 | 0.39 (126) | 0.40 | 0.73 | 0.75 | 194.2 | 258.1 | 2.46 / 5.78 / 23.35 | 1.04 | 1.42 | 64.9 / 11.2 |
| FoldFlow (base) | 96.6 | 0.20 (98) | 0.45 | 0.75 | 0.79 | 601.5 | 566.2 | 1.06 / 1.79 / 9.72 | 3.18 | 3.10 | 87.5 / 0.4 |
| FoldFlow (stoc.) | 97.0 | 0.25 (121) | 0.44 | 0.74 | 0.78 | 543.6 | 520.4 | 1.21 / 2.09 / 11.59 | 3.69 | 2.71 | 86.1 / 1.2 |
| FoldFlow (OT) | 97.2 | 0.37 (178) | 0.41 | 0.71 | 0.75 | 431.4 | 414.1 | 1.35 / 3.10 / 13.62 | 2.90 | 2.32 | 82.7 / 2.0 |
| FrameFlow | 88.6 | 0.53 (236) | 0.36 | 0.69 | **0.73** | 129.9 | 159.9 | 2.52 / 5.88 / 27.00 | 0.68 | 0.91 | 55.7 / 18.4 |
| ESM3 | 22.0 | 0.58 (64) | 0.42 | 0.85 | 0.87 | 933.9 | 855.4 | 3.19 / 6.71 / 17.73 | 1.53 | 0.98 | 64.5 / 8.5 |
| Chroma | 74.8 | 0.51 (190) | 0.38 | 0.69 | 0.74 | 189.0 | 184.1 | 2.34 / 4.95 / 18.15 | 1.00 | 1.08 | 69.0 / 12.5 |
| RFDiffusion | 94.4 | 0.46 (217) | 0.42 | 0.71 | 0.77 | 253.7 | 252.4 | 2.25 / 5.06 / 19.83 | 1.21 | 1.13 | 64.3 / 17.2 |
| Proteus | 94.2 | 0.22 (103) | 0.45 | 0.74 | 0.76 | 225.7 | 226.2 | 2.26 / 5.46 / 16.22 | 1.41 | 1.37 | 73.1 / 9.1 |
| Genie2 | 95.2 | 0.59 (281) | 0.38 | **0.63** | **0.69** | 350.0 | 313.8 | 1.55 / 3.66 / 11.65 | 2.21 | 1.70 | 72.7 / 4.8 |
| $\mathcal{M}_{FS}, \gamma=0.35$ | 98.2 | 0.49 (239) | 0.37 | 0.71 | 0.77 | 411.2 | 392.1 | 1.93 / 5.16 / 16.79 | 1.96 | 1.53 | 71.6 / 5.8 |
| $\mathcal{M}_{FS}, \gamma=0.45$ | 96.4 | 0.63 (305) | 0.36 | 0.69 | 0.75 | 388.0 | 368.2 | 2.06 / 5.32 / 19.05 | 1.65 | 1.23 | 68.1 / 6.9 |
| $\mathcal{M}_{FS}, \gamma=0.5$ | 91.4 | **0.71 (323)** | **0.35** | 0.69 | 0.75 | 380.1 | 359.8 | 2.10 / 5.18 / 19.07 | 1.55 | 1.13 | 67.0 / 7.2 |
| $\mathcal{M}_{FS}^{no\text{-}tri}, \gamma=0.45$ | 93.8 | 0.62 (292) | 0.36 | 0.69 | 0.76 | 322.2 | 306.2 | 1.80 / 4.72 / 18.59 | 1.84 | 1.36 | 71.3 / 5.5 |
| $\mathcal{M}_{21M}, \gamma=0.3$ | **99.0** | 0.30 (150) | 0.39 | 0.81 | 0.84 | 280.7 | 319.9 | 2.05 / 5.90 / 19.65 | 1.66 | 1.81 | 62.2 / 9.9 |
| $\mathcal{M}_{21M}, \gamma=0.6$ | 84.6 | 0.59 (294) | **0.35** | 0.72 | 0.77 | 280.7 | 301.8 | 2.31 / 5.76 / 30.11 | 0.89 | 0.95 | 58.7 / 12.0 |
| $\mathcal{M}_{LoRA}, \gamma=0.5$ | 96.6 | 0.43 (208) | 0.38 | 0.75 | 0.78 | 274.1 | 336.0 | 2.40 / 6.26 / 26.93 | 0.79 | 0.93 | 54.3 / 13.0 |

A higher score is desired. The fS is maximized when individual sample's class predictions $p_\phi(\cdot|\mathbf{x})$ are sharp, while the marginal distribution $\mathbb{E}_{\mathbf{x} \sim \{\mathbf{x}\}_{gen}} p_\phi(\cdot|\mathbf{x})$ has high entropy and covers many classes. Hence, this score encourages diverse generation, while individual samples should be of high quality to enable confident predictions under the classifier. The fS can also be evaluated for all CAT levels.

Our new metrics are probabilistic and directly score generated proteins at the distribution level, offering additional insights. They can help model development, but are *not* meant as optimization targets to rank models. A protein designer in practice still cares primarily about designable, diverse and novel proteins. Therefore, we did not indicate bold/underlined scores for these metrics in the evaluation tables in Sec. 4. The new metrics are evaluated with 5,000 samples in practice. In App. G, we provide details and extensively validate the new metrics on benchmarks, to establish their validity and sensitivity.

## 4 EXPERIMENTS

We trained three main Proteína models ($\mathcal{M}$), all with the possibility for conditional and unconditional generation (Sec. 3.2): *(i)* Model $\mathcal{M}_{FS}$ is trained on $\mathcal{D}_{FS}$ with a 200M parameter transformer and 15M parameters in triangle layers. *(ii)* The more efficient $\mathcal{M}_{FS}^{no\text{-}tri}$ is trained on $\mathcal{D}_{FS}$ with a 200M parameter transformer without any triangle layers nor pair representation updates. *(iii)* $\mathcal{M}_{21M}$ is trained on $\mathcal{D}_{21M}$ with a 400M parameter transformer and 15M parameters in triangle layers. Details in App. O.

### 4.1 PROTEIN BACKBONE GENERATION BENCHMARK

In Tab. 1, we compare our models' performance with baselines for protein backbone generation (see Sec. 2). We select all appropriate baselines for which code was available, as we require to generate samples to fairly evaluate metrics and follow a consistent evaluation protocol (described in detail in Apps. F and G). We did not evaluate Genie, as it is outdated since Genie2, and we were not able to compare to the recent FoldFlow2, as no code is available. We also evaluated ESM3 as a state-of-the-art masked language model that can also produce structures. Baseline evaluation and experiment details in Apps. O and P. All models and baselines in Tab. 1 are adjusted for high designability via rotation annealing or reduction of the noise scale during inference. Tab. 1 findings:

**Unconditional generation.** *(i)* $\mathcal{M}_{FS}$ can be tuned during inference for different designability, diversity and novelty trade-offs (varying $\gamma$). It outperforms all baselines in designability and diversity, while performing competitively on novelty, only behind Genie2 and FrameFlow for AFDB novelty (model samples in Fig. 2). *(ii)* $\mathcal{M}_{FS}^{no\text{-}tri}$ still reaches 93.8% designability and outperforms all baselines on diversity, despite not using any expensive triangle layers and no pair track updates—in contrast

Table 2: Proteína's and Chroma's **fold class-conditional backbone generation performance**.

| Model | Design-ability (%)↑ | Diversity Cluster↑ | TM-Sc.↓ | Novelty vs. PDB↓ | AFDB↓ | FPSD vs. PDB↓ | AFDB↓ | fS (C / A / T)↑ | fJSD vs. PDB↓ | AFDB↓ | Sec. Struct. % (α / β ) |
|---|---|---|---|---|---|---|---|---|---|---|---|
| *Fold class-conditional generation with Proteína model $\mathcal{M}_{FS}^{cond}$ and CFG with guidance weight $\omega$ and noise scale $\gamma = 0.4$.* | | | | | | | | | | | |
| Chroma | 57.0 | **0.65 (186)** | 0.37 | **0.68** | **0.73** | 157.8 | 131.0 | 2.36 / 5.11 / 19.82 | 0.84 | 0.77 | 70.2 / 11.1 |
| $\mathcal{M}_{FS}^{cond}, \omega=1.0$ | **91.4** | 0.57 (262) | 0.34 | 0.77 | 0.81 | 121.1 | 127.6 | 2.50 / 6.93 / 31.31 | 0.58 | 0.52 | 57.1 / 13.7 |
| $\mathcal{M}_{FS}^{cond}, \omega=1.5$ | 89.2 | 0.57 (252) | **0.33** | 0.77 | 0.81 | 106.1 | 113.5 | 2.58 / 7.36 / 32.72 | 0.49 | 0.47 | 56.0 / 14.6 |
| $\mathcal{M}_{FS}^{cond}, \omega=2.0$ | 83.8 | 0.54 (225) | **0.33** | 0.78 | 0.82 | 103.0 | 108.3 | 2.62 / 7.55 / 33.74 | 0.45 | 0.43 | 54.5 / 15.7 |

Table 3: Proteína's and GENIE2's backbone generation performance *when evaluated to sample full distribution, i.e. no noise or temperature reduction.* Metric details in Apps. F and G. Best scores **bold**, second best underlined.

| Model | Design-ability (%)↑ | Diversity | | Novelty vs. | | FPSD vs. | | fS | fJSD vs. | | Sec. Struct. % |
|---|---|---|---|---|---|---|---|---|---|---|---|
| | | Cluster↑ | TM-Sc.↓ | PDB↓ | AFDB↓ | PDB↓ | AFDB↓ | (C / A / T)↑ | PDB↓ | AFDB↓ | ($\alpha$ / $\beta$) |
| *Unconditional generation. $\mathcal{M}_i^j$ denotes Proteína model variant. Sampling for Proteína performed using generative ODE (App. I), for GENIE with their approach.* | | | | | | | | | | | |
| Genie2 | 19.0 | 0.81 (77) | 0.33 | **0.66** | **0.72** | 104.7 | 29.94 | 2.24 / 4.49 / 22.83 | 0.75 | 0.16 | 65.0 / 7.5 |
| $\mathcal{M}_{FS}$ | 19.6 | **0.93 (91)** | 0.32 | **0.66** | 0.74 | 85.39 | 21.41 | 2.51 / 5.65 / 27.35 | 0.59 | 0.09 | 48.2 / 13.2 |
| $\mathcal{M}_{21M}$ | 35.4 | 0.65 (115) | 0.34 | 0.74 | 0.79 | 50.14 | 44.98 | 2.51 / 6.46 / 39.65 | 0.32 | 0.23 | 55.7 / 11.8 |
| $\mathcal{M}_{LoRA}$ | **44.2** | 0.58 (129) | 0.35 | 0.73 | 0.75 | 68.56 | 138.6 | 2.61 / 7.19 / 38.64 | 0.31 | 0.82 | 47.2 / 13.4 |
| *Fold class-conditional generation with Proteína model $\mathcal{M}_{FS}^{cond}$ and CFG with guidance weight $\omega$. Sampling is performed using generative ODE (App. I).* | | | | | | | | | | | |
| $\mathcal{M}_{FS}^{cond}, \omega{=}1.0$ | 24.2 | 0.74 (90) | **0.29** | 0.73 | 0.79 | 71.46 | 19.45 | 2.64 / 6.75 / 26.64 | 0.40 | 0.12 | 48.7 / 14.7 |

to all existing models. *(iii)* $\mathcal{M}_{21M}$ achieves state-of-the-art 99.0% designability, while generating less diverse structures. This is expected, as it is trained on the very large, yet strongly filtered $\mathcal{D}_{21M}$. Models trained on $\mathcal{D}_{FS}$ exhibit higher diversity, because no radius of gyration or secondary structure filtering was used during data curation. With $\mathcal{D}_{21M}$ we were able to prove that one can create high-quality datasets, much larger than $\mathcal{D}_{FS}$, from fully synthetic structures that can be used for training generative models producing almost entirely designable structures. Furthermore, our discussed findings represent an important proof that non-equivariant architectures can achieve state-of-the-art performance on protein backbone generation. All baselines use fully equivariant networks.

**PDB-LoRA $\mathcal{M}_{LoRA}$.** We used LoRA (Hu et al., 2022) to fine-tune $\mathcal{M}_{FS}$ on a small dataset of only designable proteins from the PDB (App. M.1). As expected, designability improves, diversity decreases, FPSD and fJSD with respect to PDB decrease, and FPSD and fJSD with respect to AFDB increase. This experiment showcases how a model that is trained only on synthetic data can be successfully fine-tuned on natural proteins, and the metrics validate that the generated samples indeed are closer to the PDB in distribution. Moreover, the amount of $\beta$-sheets doubles, an important aspect, due to the under-representation of $\beta$-sheets in many protein design models. To the best of our knowledge, this is the first time that such LoRA fine-tuning has been demonstrated for protein structure flow or diffusion models.

**Fold-Class conditional generation and new metrics.** Next, we evaluate our fold class-conditional model $\mathcal{M}_{FS}^{cond}$ as well as Chroma, the only baseline that also supports class-conditional sampling (see Tab. 2). We feed the labels from the empirical label distribution of $\mathcal{D}_{FS}$ to the models. This enforces diversity across different fold structures, which is reflected in the metrics. Compared to unconditional generation, our conditional model achieves state-of-the-art TM-Score diversity, while also reaching the best FPSD, fS and fJSD scores, thereby demonstrating fold structure diversity (fS) and a better match in distribution to the references (FPSD, fJSD). Moreover, this is achieved while maintaining very high designability. Further, the effect is enhanced by classifier-free guidance ($\omega \geq 1.0$). Fold class-conditioning also significantly improves the $\beta$-sheet content of the generated backbones. Note that, however, the model does not improve novelty. Novelty can be at odds with learning a better model of the training distribution—the goal of any generative model—as it rewards samples completely outside the training distribution. That motivates our new metrics, which are complementary, as clearly shown in the class-conditioning case. Chroma has very poor designability and is outperformed in TM-score diversity and the number of designable cluster. Moreover, we show in App. D.1 that, in contrast to Proteína, Chroma fails to perform accurate fold class-specific generation by analyzing whether generated proteins correspond to the correct conditioning fold classes.

**Full distribution modeling.** Most models use temperature and noise scale reduction or rotation schedule annealing during inference to increase designability at the cost of diversity. In Tab. 3, we analyze performance when sampling the entire distribution instead, comparing to Genie2 also sampled at full temperature. Genie2 produces the least designable samples. $\mathcal{M}_{FS}$ performs overall on-par with or better than Genie2, but $\mathcal{M}_{21M}$ has much higher designability and LoRA fine-tuning also gives a big boost. Moreover, almost all new distribution metrics (FPSD, fS, fJSD) are significantly improved over Tab. 1, as we now sample the entire distribution. This is only fully captured by our new metrics.

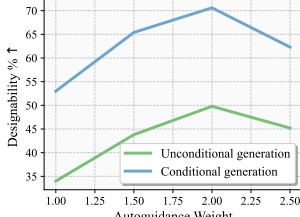

Figure 7: Designability of $\mathcal{M}_{21M}$ ODE samples with autoguidance.

**Autoguidance.** In Fig. 7, we show a case-study of autoguidance (Karras et al. (2024), see App. I) for protein backbone generation with our $\mathcal{M}_{21M}$ model in full distribution mode (ODE), using an early training checkpoint as "bad" guidance checkpoint. We can significantly boost designability, up to 70% in conditional generation, far surpassing the results in Tab. 3. To the best of our knowledge, this is the first proof of principle of autoguidance in the context of protein structure generation.

## 4.2 Long Chain Generation

While our main models are trained on proteins of up to 256 residues, we fine-tune the $\mathcal{M}_{FS}^{no\text{-}tri}$ model on proteins of up to 768 residues (App. O for details). In Fig. 8, we show our model's performance on long protein backbone generation of up to 800 residues (samples in Fig. 4.). While Genie2 exhibits superior diversity at 300 residues, beyond that Proteína significantly outperforms all baselines by a large margin, achieving state-of-the-art results. At very long lengths, all baselines collapse and cannot produce diverse designable proteins anymore. In contrast, for our model most generated backbones are designable even at length 800 and we still generate many diverse proteins, as measured by the number of designable clusters. To the best of our knowledge, no previous protein backbone generators successfully trained on proteins up to that length.

It is possible for us because $\mathcal{M}_{FS}^{no\text{-}tri}$ does not use any expensive triangle layers and no pair track updates, relying only on our novel efficient transformer, whose scalability this experiment validates. We envision that such long protein backbone generation unlocks new large-scale protein design tasks. Note that long length genera-

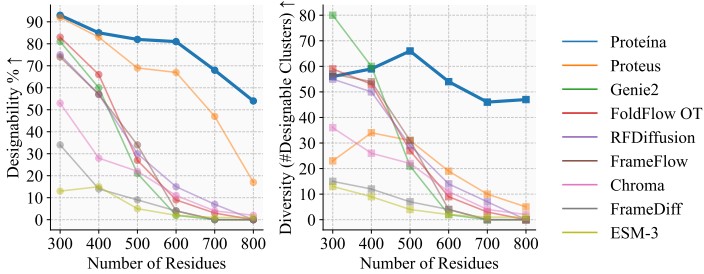

Figure 8: Proteína long backbone generation performance (also App. O.5).

tion can be combined with our novel fold class conditioning, too, offering additional control (Fig. 4).

## 4.3 Fold Class-specific Guidance and Increased $\beta$-Sheets

A problem that has plagued protein structure generators for a long time is that they typically produce much more $\alpha$-helices than $\beta$-sheets (Tabs. 1 and 3). Our fold class conditioning offers a new tool to address this without the need for fine-tuning (Huguet et al., 2024). In Tab. 4, we guide the $\mathcal{M}_{FS}^{cond}$ model with respect to the main high-level C level classes that determine secondary structure content (details App. O). When guiding into the "mixed $\alpha/\beta$" and especially "mainly $\beta$" classes, $\beta$-sheets increase dramatically in contrast to unconditional or "mainly $\alpha$" generation and also compared to all baselines in Tab. 1. Importantly, the samples remain designable. As we restrict generation to specific classes, diversity slightly decreases as expected, but we still generate diverse samples.

Aside from C-level guidance to achieve controlled secondary structure diversity, we can also guide with respect to interesting or relevant A- and T-level classes. In Fig. 6, we show examples of guidance into different fold classes from the CAT

Table 4: Guiding Proteína into the C-level classes.

| Class | Design-ability % ↑ | Diversity | | Novelty vs. | | Sec. Struct. % | | |
|---|---|---|---|---|---|---|---|---|
| | | Foldseek↑ | TM-Sc.↓ | PDB↓ | AFDB↓ | $\alpha$ | $\beta$ | coil |
| Unconditional | 96.4 | 0.63 (305) | 0.36 | 0.69 | 0.75 | 68.1 | 6.9 | 25.0 |
| "Mainly $\alpha$" | 96.6 | 0.37 (179) | 0.42 | 0.77 | 0.82 | 82.5 | 0.6 | 16.9 |
| "Mainly $\beta$" | 90.0 | 0.48 (215) | 0.37 | 0.75 | 0.82 | 14.9 | 33.3 | 51.8 |
| "Mixed $\alpha/\beta$" | 97.8 | 0.42 (207) | 0.37 | 0.73 | 0.78 | 44.1 | 20.5 | 35.4 |

hierarchy, demonstrating that Proteína offers unprecedented control over protein backbone generation. We would also like to point to App. D, where we extensively validate that our novel fold class conditioning correctly works by re-classifying generated conditional samples with our fold class predictor.

*Further Proteína samples in App. A. Proteína also achieves state-of-the-art performance in **motif-scaffolding** (App. B). Speed and efficiency analysis in App. C.2. More experiments in Apps. E and L.*

## 5 Conclusions

We have presented Proteína, a foundation model for protein backbone generation. It features novel fold class conditioning, offering unprecedented control over the synthesized protein structures. In comprehensive unconditional, class-conditional and motif scaffolding benchmarks, Proteína achieves state-of-the-art performance. Our driving neural network component is a scalable non-equivariant transformer, which allows us to scale Proteína to synthesize designable and diverse backbones up to 800 residues. We also curate a 21M-sized high-quality dataset from the AFDB and, scaling Proteína to over 400M parameters, show that highly designable protein generation is achievable even when training on synthetic data at such unprecedented scale. For the first time, we demonstrate not only classifier-free but also autoguidance as well as LoRA-based fine-tuning in protein structure flow models. Finally, we introduce new distributional metrics that offer novel insights into the behaviors of protein structure generators. We hope that Proteína unlocks new large-scale protein design tasks while offering increased control.

## REPRODUCIBILITY STATEMENT

We ensure that our data processing, network architecture design, inference-time sampling, sample evaluations, and baseline comparisons are reproducible. Our Appendix offers all necessary details and provides comprehensive explanations with respect to all aspects of this work.

In addition to Sec. 3.1, in App. M we describe in detail how our $\mathcal{D}_{FS}$ and $\mathcal{D}_{21M}$ datasets are created, processed, filtered and clustered, which includes the hierarchical CAT fold class labels that we use. Dataset statistics are given in Fig. 3, which can serve as reference. Additional tools that we use during data processing and evaluation, such as MMseqs2 (Steinegger & Söding, 2017) and Foldseek (van Kempen et al., 2024; Barrio-Hernandez et al., 2023), are publicly available and we cite them accordingly. Hence, our data processing pipeline is fully reproducible. Next, our new transformer architecture is explained in detail in Sec. 3.3 and App. N, with detailed module visualizations in Figs. 5 and 24 and network hyperparameters in App. O. Inference time sampling is described in Sec. 3.4 with additional algorithmic details in App. I. The corresponding sampling hyperparameters are provided in App. O. Furthermore, how we evaluate the traditional protein structure generation metrics is explained in detail in App. F and App. F.1, while our newly proposed metrics (Sec. 3.5) are validated and explained in-depth in App. G. Moreover, to ensure our extensive baseline comparisons are also reproducible, the corresponding details are described in App. P.

For model and code release, please see Proteína's GitHub repository `https://github.com/NVIDIA-Digital-Bio/proteina/` as well as our project page `https://research.nvidia.com/labs/genair/proteina/`.

## ETHICS STATEMENT

Protein design has been a grand challenge of molecular biology with many promising applications benefiting humanity. For instance, novel protein-based therapeutics, vaccines and antibodies created by generative models hold the potential to unlock new therapies against disease. Moreover, carefully engineered enzymes may find broad industrial applications and serve, for example, as biocatalysts for green chemistry and in manufacturing. Novel protein structures may also yield new biomaterials with applications in materials science. Beyond that, deep generative models encoding a general understanding of protein structures may improve our understanding of protein biology itself. However, it is important to be also aware of potentially harmful applications of generative models for de novo protein design, for instance related to biosecurity. Therefore, protein generative models generally need to be applied with an abundance of caution.

## ACKNOWLEDGMENTS

We would like to thank Pavlo Molchanov, Bowen Jing and Hannes Stärk for helpful discussions. We also thank NVIDIA's compute infrastructure team for maintaining the GPU resources we utilized. Last, not least, thanks to all computational colleagues who make their tools available, to all experimental colleagues who help advancing science by making their data publicly available, and to all those who maintain the crucial databases we build our tools on, like the RCSB PDB, the AFDB, and UniProt.

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

# Appendix

## A  ADDITIONAL PROTEÍNA SAMPLE VISUALIZATIONS

In Fig. 9, we show additional protein backbones generated by Proteína, covering the entire chain length spectrum of our model. These samples are generated without any conditioning. Furthermore, in Fig. 10 we show additional fold class-conditioned samples and in Fig. 12 are visualizations of successful motif-scaffolding.

Note that in all figures all shown samples are designable, according to our definition of designability (see App. F).

## B  MOTIF-SCAFFOLDING WITH PROTEÍNA

To validate the performance of Proteína in conditional tasks beside fold conditioning, we implement motif-scaffolding capabilities for Proteína and test its performance on the RFDiffusion benchmark (Watson et al., 2023).

### B.1  MOTIF-SCAFFOLDING IMPLEMENTATION

To enable Proteína to perform motif-scaffolding, we add two additional features to our model via embedding layers: the motif structure (with coordinates set to the origin for residues that are not part of the motif) and a motif mask (1 for positions that are part of the motif, 0 for positions that are not). In addition, we center the data ($\mathbf{x}_1$) and the noise ($\mathbf{x}_0$) not based on the overall centre of mass, but only on the center of mass calculated over the motif coordinates.

At inference time, sampling is initialized as before but again centered based on the center of mass calculated over the motif coordinates. We train a model with 60M parameters in the transformer layers and 12M parameters in multiplicative triangle layers, using the same dataset and motif training augmentation as Lin et al. (2024). We use a batch size of 5 and add an additional motif structure auxiliary loss with a weight of 5 in addition to the losses discussed in 3.2. Compared to Lin et al. (2024), in addition to specifying the structural constraints of the motif in the model pair representation, we encode the masked motif coordinates in the sequence representation. Thus, given conditional motif coordinates, the model is tasked with inpainting a designable scaffold. At inference time we sample with a reduced noise scale, as done in the unconditional case, using $\gamma = 0.5$.

### B.2  MOTIF-SCAFFOLDING RESULTS

Motif-scaffolding performance is judged by common criteria outlined in previous work (Lin et al., 2024; Watson et al., 2023):

- For each problem in the benchmark set by Watson et al. (2023), 1000 backbones are generated.
- For each backbone, 8 ProteinMPNN sequences are generated with fixed sequences in the motif region following the convention of Lin et al. (2024)

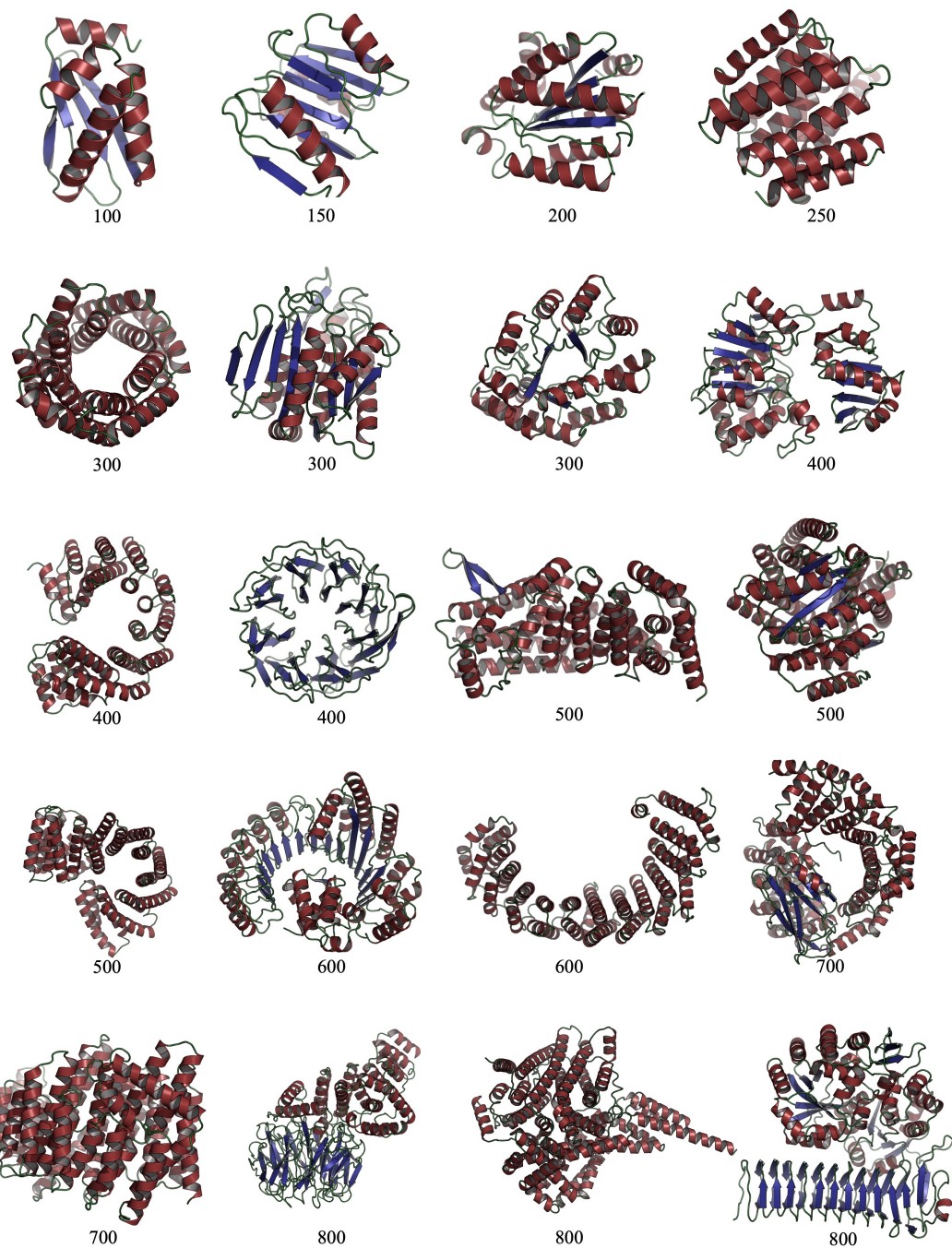

Figure 9: **Unconditional Proteína Samples.** The numbers below the proteins denote the generated proteins' number of residues. All shown proteins are designable.

- All 8 sequences per backbone are fed to ESMFold. The predicted structures are used to compute the scRMSD, which is the $C_\alpha$-RMSD between the designed and predicted backbone, as well as the motifRMSD, which is the full backbone RMSD between the predicted and the ground truth motif.

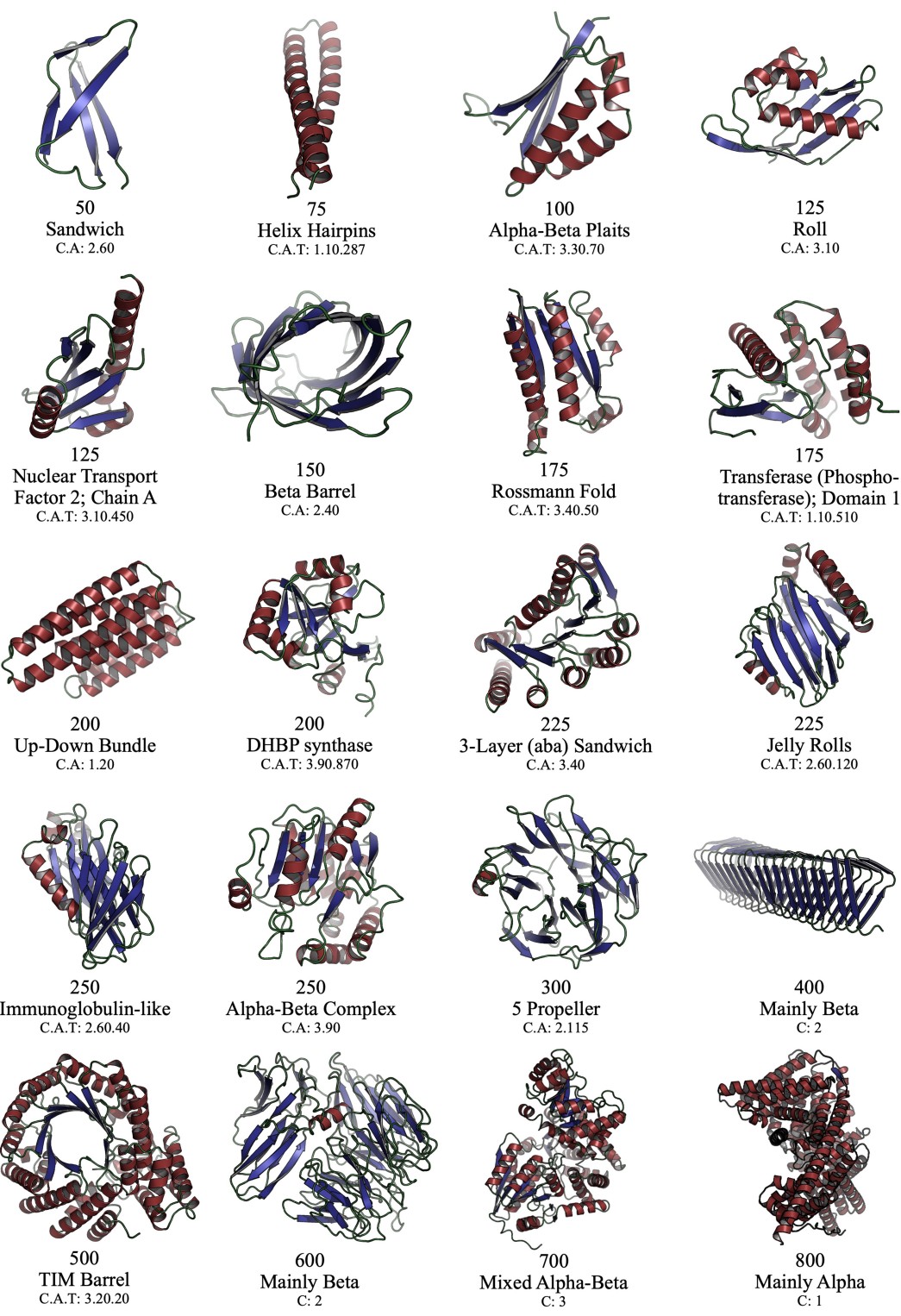

Figure 10: **Fold Class-Conditional Proteína Samples.** The numbers below the proteins denote the generated proteins' number of residues. Moreover, we show the fold class and the corresponding C.A.T fold class code for conditioning. All shown proteins are designable and correctly re-classified into the conditioning fold class.

Table 5: Number of unique successes on the RFDiffusion benchmark for 4 different methods, each generating 1000 backbones.

| Task Name | Proteina | Genie2 | RFDiffusion | FrameFlow |
|---|---|---|---|---|
| 6E6R_long | **713** | 415 | 381 | 110 |
| 6EXZ_long | 290 | 326 | 167 | **403** |
| 6E6R_medium | **417** | 272 | 151 | 99 |
| 1YCR | **249** | 134 | 7 | 149 |
| 5TRV_long | **179** | 97 | 23 | 77 |
| 6EXZ_med | 43 | 54 | 25 | **110** |
| 7MRX_128 | 51 | 27 | **66** | 35 |
| 6E6R_short | **56** | 26 | 23 | 25 |
| 5TRV_med | 22 | **23** | 10 | 21 |
| 7MRX_85 | **31** | 23 | 13 | 22 |
| 3IXT | 8 | **14** | 3 | 8 |
| 5TPN | 4 | **8** | 5 | 6 |
| 7MRX_60 | 2 | **5** | 1 | 1 |
| 1QJG | 3 | 5 | 1 | **18** |
| 5TRV_short | 1 | **3** | 1 | 1 |
| 5YUI | **5** | 3 | 1 | 1 |
| 4ZYP | **11** | 3 | 6 | 4 |
| 6EXZ_short | 3 | 2 | 1 | 3 |
| 1PRW | 1 | 1 | 1 | 1 |
| 5IUS | 1 | 1 | 1 | 0 |
| 1BCF | 1 | 1 | 1 | 1 |
| 5WN9 | 2 | 1 | 0 | **3** |
| 2KL8 | 1 | 1 | 1 | 1 |
| 4JHW | 0 | 0 | 0 | 0 |

- A backbone is classified as a success when one of the sequences generated for it has an scRMSD $\leq 2$Å, a motifRMSD $\leq 1$Å, pLDDT $\geq 70$, and pAE $\leq 5$.

- All successes are clustered via hierarchical clustering with single linkage and a TM-score threshold of 0.6 to reach the final number of unique successes.

Looking at the performance over the entire benchmark (Tab. 6 and Fig. 11), we see that Proteína has the highest number of unique successes overall in the benchmark (2094 compared to 1445 for the second-best method Genie2) and is the sole best method in 8 tasks (compared with the second-best method Genie2 that wins in 5 tasks).

Investigating the performance for each task individually (Tab. 5 and Fig. 13), we see that Proteína outperforms mostly on easy and medium tasks, whereas the hardest tasks with 1 or 0 successes still seem challenging. Successful designs are shown in Fig. 12.

Table 6: Number of unique successes summed over the whole RFDiffusion benchmark and number of times a method was the sole best method for 4 different methods, each generating 1000 backbones. See Fig. 11 for a bar chart of these results.

| Task Name | Proteina | Genie2 | FrameFlow | RFDiffusion |
|---|---|---|---|---|
| #Successes total | 2094 | 1445 | 1099 | 889 |
| #Tasks as best method | 8 | 5 | 4 | 1 |

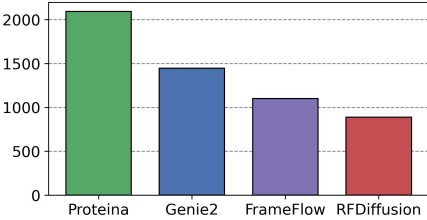
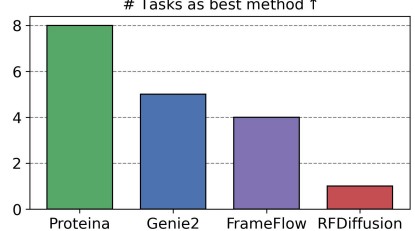

Figure 11: Motif-scaffolding results. Number of unique successes summed over the whole RFDiffusion benchmark and number of times a method was the sole best method for 4 different methods, each generating 1000 backbones. These are the same numbers as in Tab. 6.

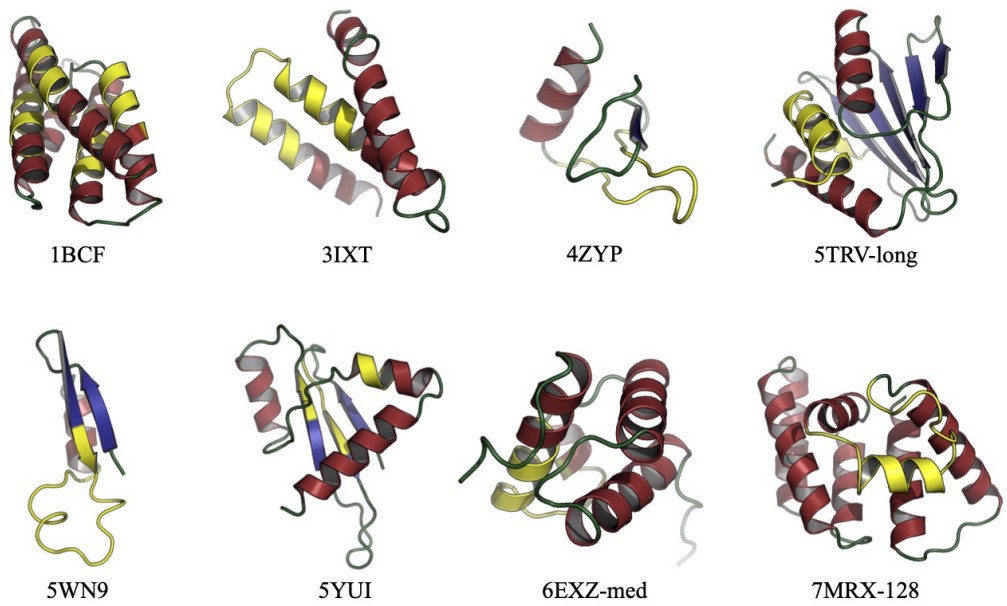

Figure 12: Examples of successful designs in the motif-scaffolding benchmark. All shown samples satisfy the criteria for task success. The task specification is given below the proteins. Motif residues are shown in yellow.

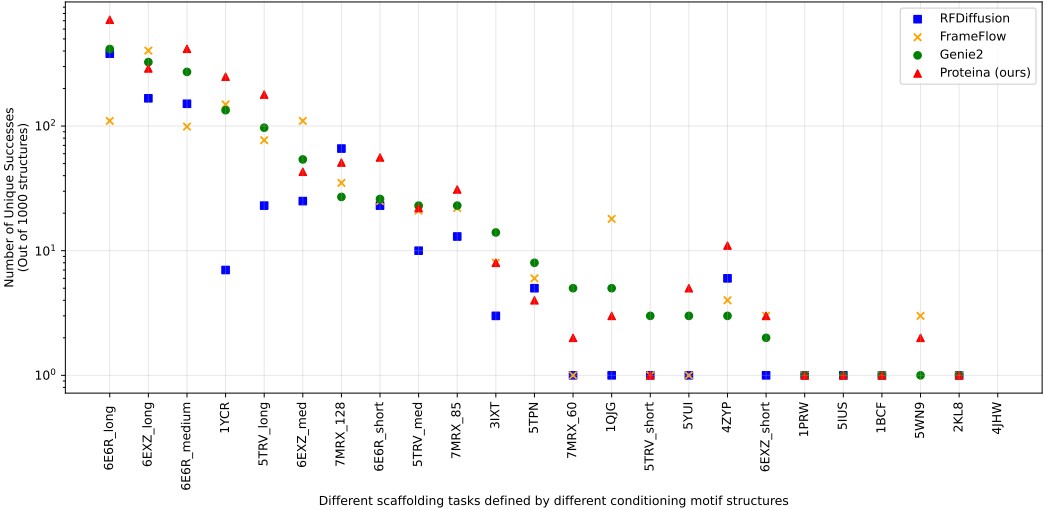

Figure 13: Motif-scaffolding results. Numbers of unique successes on the RFDiffusion benchmark, following the success definition and clustering methodology from Genie2 (Lin et al., 2024).

## C  SCALING AND EFFICIENCY ANALYSIS

### C.1  SCALING FLOW MATCHING TRAINING

In Fig. 14, we study the optimization of Proteína's flow matching objective as function of the number of parameters, using Proteína models without triangular multiplicate layers, scaling the novel non-equivariant transformer architecture. We trained models of various sizes between ≈60M and ≈400M parameters, and we find that we can consistently improve the loss as we scale the model size, thereby validating the scalability of our architecture. This observation is in line with recent work on state-of-the-art image generation (Esser et al., 2024), leveraging a similar flow matching approach.

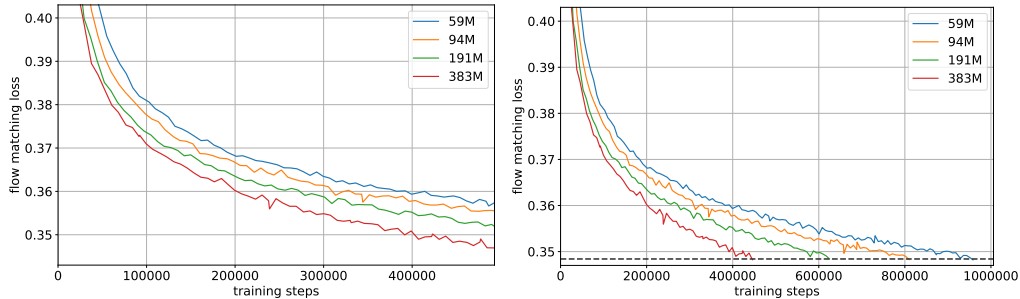

Figure 14: *Left:* Flow matching loss over the course of training for differently sized Proteína models (number of model parameters given at the top right). Batch size 5 for all. *Right:* The same training curves, but we emphasize that when scaling the model the flow matching loss reaches similarly low values (gray dashed line) significantly faster.

Table 7: Unconditional backbone generation performance of the additional, smaller $\mathcal{M}_{FS}^{small}$ Proteína model, side-by-side with the other, larger models that we trained. Results partly copied from Tab. 1.

| Model | Design-ability (%)↑ | Diversity | | Novelty vs. | | FPSD vs. | | fS | fJSD vs. | | Sec. Struct. % |
| | | Cluster↑ | TM-Sc.↓ | PDB↓ | AFDB↓ | PDB↓ | AFDB↓ | (C / A / T)↑ | PDB↓ | AFDB↓ | ($\alpha$ / $\beta$ ) |
| *Unconditional generation. $\mathcal{M}_i^j$ denotes the Proteína model variant, and $\gamma$ is the noise scale for Proteína.* | | | | | | | | | | | |
| $\mathcal{M}_{FS}$, $\gamma$=0.45 | 96.4 | 0.63 (305) | 0.36 | 0.69 | 0.75 | 388.0 | 368.2 | 2.06 / 5.32 / 19.05 | 1.65 | 1.23 | 68.1 / 6.9 |
| $\mathcal{M}_{FS}^{no-tri}$, $\gamma$=0.45 | 93.8 | 0.62 (292) | 0.36 | 0.69 | 0.76 | 322.2 | 306.2 | 1.80 / 4.72 / 18.59 | 1.84 | 1.36 | 71.3 / 5.5 |
| $\mathcal{M}_{FS}^{small}$, $\gamma$=0.45 | 94.8 | 0.55 (273) | 0.35 | 0.72 | 0.78 | 322.3 | 323.3 | 2.21 / 5.91 / 22.83 | 1.53 | 1.24 | 64.7 / 8.0 |

## C.2 MODEL PARAMETERS, SAMPLING SPEED AND MEMORY CONSUMPTION

To compare the parameter counts of different models as well as the practical implications of these parameter counts such as memory consumption and sampling speed, we conduct three analyses:

1. Models are **sampled with batch size 1** and the sampling time is measured. This is run on an A6000-48GB GPU for comparison with previous works (Lin et al., 2024). See Tab. 8 and Fig. 15.

2. For all tested models, we determine the **largest supported batch size** that fits into GPU memory and does not result in out-of-memory errors. This is executed on an A100-80GB GPU. See Tab. 9.

3. Models are sampled with their maximum batch size and the **sampling time** is measured, **normalized** with respect to the batch size. This is executed on an A100-80GB GPU. See Tab. 10.

Each of the linked tables shows all models' number of parameters.

As part of these experiments, we use an additional model $\mathcal{M}_{FS}^{small}$ which only contains around 60M parameters (similar to RFDiffusion), but still performs very competitively, outperforming most baselines like RFDiffusion (Tab. 7). As one would expect due to the smaller model size, it does perform slightly worse than our larger state-of-the-art models, though, showing slightly worse diversity and novelty. The training and sampling of this model follows the setting from $\mathcal{M}_{FS}^{no-tri}$, with the main difference being the number of parameters. For all our models, we leverage the fact that our transformer-based architecture is amenable to hardware optimisations and leverage the torch compilation framework (Ansel et al., 2024) to speed up training and inference. The inference numbers depicted here for Proteína account for inference time of the compiled model.

Looking at sampling time for single protein generation (batch size 1) on an A6000-48GB (Tab. 8), we see that the runtime of Proteína depends on whether we use triangle layers or not: Proteína models with triangle layers are still faster than state-of-the-art tools like RFDiffusion and Genie2, but are slower than FrameFlow at all lengths and slower than Chroma at longer lengths. However, Proteína models without triangle layers are a lot faster and perform competitively even with much smaller models like FrameFlow (with $\mathcal{M}_{FS}^{small}$ running faster than FrameFlow for all lengths). Note that we compare with RFDiffusion, Genie2, FrameFlow and Chroma, as these represent the most competitive baselines.

In practice, one performs inference batch-wise. To compare the performance of Proteína in this setting, we determined the maximum batch size for each method on an A100-80GB GPU (Tab. 9)

Table 8: Sampling time [seconds] for different methods at batch size 1 for samples of varying length (the numbers in the top row indicate protein backbone chain length) on an A6000-48GB GPU.

| Method | # Model parameters | Inference steps | 100 | 200 | 300 | 400 | 500 | 600 | 700 | 800 |
|---|---|---|---|---|---|---|---|---|---|---|---|
| Genie2 | 15.7M | 1000 | 48 | 75 | 135 | 233 | 356 | 536 | 740 | 961 |
| RFDiffusion | 59.8M | 50 | 21 | 41 | 80 | 137 | 214 | 296 | 397 | 531 |
| FrameFlow | 17.4M | 100 | 4 | 6 | 9 | 13 | 18 | 22 | 28 | 35 |
| Chroma | 18.5M | 500 | 22 | 29 | 36 | 42 | 49 | 55 | 63 | 69 |
| $\mathcal{M}_{FS}^{small}$ | 59M | 400 | 3 | 3 | 6 | 8 | 12 | 18 | 25 | 32 |
| $\mathcal{M}_{FS}^{no\text{-}tri}$ | 191M | 400 | 3 | 5 | 9 | 15 | 23 | 32 | 42 | 54 |
| $\mathcal{M}_{FS}$ | 208M | 400 | 8 | 26 | 63 | 119 | 188 | 273 | 370 | 529 |
| $\mathcal{M}_{21M}$ | 397M | 400 | 8 | 24 | 54 | 102 | 159 | 230 | 310 | 408 |

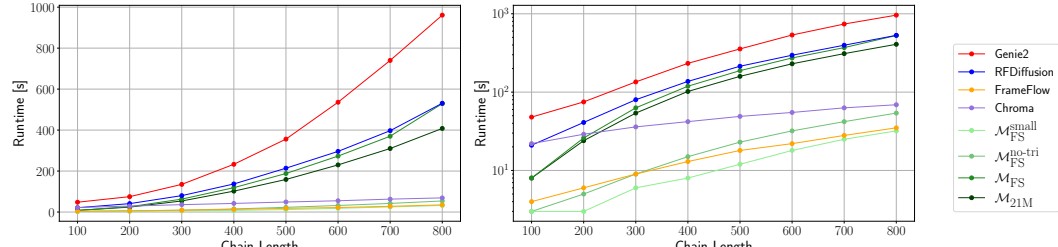

Figure 15: **Single sample runtimes.** The runtimes for different models for batch size 1 on a A6000-48GB GPU. Different scales are used for y-axis (left - linear, right - logarithmic). The same data is shown in Tab. 8.

Table 9: Maximum batch size during inference for different methods for samples of varying length (the numbers in the top row indicate protein backbone chain length) on an A100-80GB GPU.

| Method | # Model parameters | Inference steps | 100 | 200 | 300 | 400 | 500 | 600 | 700 | 800 |
|---|---|---|---|---|---|---|---|---|---|---|---|
| Genie2 | 15.7M | 1000 | 204 | 51 | 22 | 12 | 8 | 5 | 4 | 3 |
| Chroma | 18.5M | 500 | 862 | 435 | 285 | 211 | 162 | 136 | 116 | 101 |
| $\mathcal{M}_{FS}^{small}$ | 59M | 400 | 1599 | 416 | 200 | 194 | 72 | 46 | 36 | 25 |
| $\mathcal{M}_{FS}^{no\text{-}tri}$ | 191M | 400 | 700 | 187 | 85 | 48 | 31 | 21 | 16 | 12 |
| $\mathcal{M}_{FS}$ | 208M | 400 | 199 | 55 | 26 | 14 | 9 | 6 | 4 | 3 |
| $\mathcal{M}_{21M}$ | 397M | 400 | 157 | 44 | 20 | 11 | 7 | 5 | 3 | 2 |

Table 10: Sampling time [seconds] for different methods at max batch size for varying lengths (the numbers in the top row) on an A100-80GB GPU. The time is obtained by dividing the total runtime by the batch size.

| Method | # Model parameters | Inference steps | 100 | 200 | 300 | 400 | 500 | 600 | 700 | 800 |
|---|---|---|---|---|---|---|---|---|---|---|---|
| Genie2 | 15.7M | 1000 | 27.74 | 65.47 | 117.59 | 183.67 | 257.63 | 373.40 | 526.00 | 690.67 |
| Chroma | 18.5M | 500 | 4.81 | 9.56 | 12.09 | 17.58 | 21.99 | 26.31 | 30.84 | 35.17 |
| $\mathcal{M}_{FS}^{small}$ | 59M | 400 | 0.29 | 0.94 | 2.01 | 3.38 | 5.26 | 7.33 | 9.97 | 14.44 |
| $\mathcal{M}_{FS}^{no\text{-}tri}$ | 191M | 400 | 0.59 | 1.88 | 3.87 | 6.54 | 9.96 | 14.04 | 18.87 | 24.33 |
| $\mathcal{M}_{FS}$ | 208M | 400 | 3.74 | 13.05 | 28.31 | 50.14 | 80.89 | 125.33 | 173.25 | 229.00 |
| $\mathcal{M}_{21M}$ | 397M | 400 | 3.29 | 11.20 | 24.35 | 42.64 | 75.57 | 105.80 | 144.33 | 192.00 |

and then determined the normalized sampling times per sequence in this batch setting by dividing the overall batch runtime by the batch size (Tab. 10). No numbers were reported for RFDiffusion and FrameFlow since these methods do not support batched inference, limiting the batch size to 1.

Even with Proteína having more parameters than the baselines, we see that Proteína models with triangle layers can fit similar batch sizes to Genie2. On the other hand, Proteína models without triangle layers can fit very large batches, up to 1.6k proteins of length 100 for $\mathcal{M}_{FS}^{small}$.

Looking at the per-sequence sampling time in the max batch size setting (Tab. 10), we see that Proteína benefits strongly from batched inference, especially for models without triangle layers and shorter sequence lengths. This enables fast batched sample generation, with less than 1 second per chain for short chain lengths.

Our overall conclusion from these experiments is that even though we investigated model size scaling in this work, this scaling does not come at a cost in terms of inference efficiency, thanks to our efficient and scalable architecture. Our models support batches as large as or larger than the baselines and can be sampled as fast as or faster than the baselines, meanwhile leading to state-of-the-art protein backbone generation performance (see main paper).

Table 11: Fold class-conditioned generation: We report the generated proteins' re-classification probabilities of the correct fold class label that was used during conditioning.

| Setup | C | | | A | | | T | | |
|---|---|---|---|---|---|---|---|---|---|
| | $\alpha$ | $\beta$ | $\alpha/\beta$ | Common | Regular | Rare | Common | Regular | Rare |
| *Classifier-free guidance ($\alpha = 0.0$), guidance weight $\omega$ (Proteína model $\mathcal{M}_{21M}^{cond}$, $\gamma = 0.3$).* | | | | | | | | | |
| Proteína, $\omega$=0.0 | 0.585 | 0.017 | 0.450 | 0.128 | 0.014 | 0.000 | 0.032 | 0.002 | 0.000 |
| Proteína, $\omega$=0.5 | 0.914 | 0.479 | 0.784 | 0.437 | 0.204 | 0.119 | 0.336 | 0.114 | 0.006 |
| Proteína, $\omega$=1.0 | 0.986 | 0.887 | 0.961 | 0.701 | 0.334 | 0.226 | 0.570 | 0.209 | 0.010 |
| Proteína, $\omega$=1.5 | 0.993 | 0.962 | 0.977 | 0.772 | 0.363 | 0.242 | 0.611 | 0.225 | 0.012 |
| Proteína, $\omega$=2.0 | 0.992 | 0.975 | 0.976 | 0.788 | 0.383 | 0.233 | 0.638 | 0.230 | 0.012 |
| Proteína, $\omega$=2.5 | 0.993 | 0.979 | 0.997 | 0.842 | 0.366 | 0.298 | 0.636 | 0.224 | 0.012 |
| Chroma | 0.888 | 0.486 | 0.644 | 0.240 | 0.007 | 0.000 | 0.133 | 0.002 | 0.000 |

# D    VALIDATING FOLD CLASS CONDITIONING VIA RE-CLASSIFICATION

To analyze whether our fold class conditioning correctly works, we re-classify generated conditional samples with our fold class predictor and validate whether the generated samples correctly correspond to their conditioning classes; see Tab. 11. We use classifier-free guidance on the model $\mathcal{M}_{21M}^{cond}$ with a noise scale of $\gamma = 0.3$, which yields the best re-classification probabilities. We guide the model to generate 100 samples for each C-level class, 30 samples for each A-level class, and 2 samples for each T-level class. The generated samples are then evaluated using our fold classifier (trained in App. G.3.2) to predict the probability that they belong to the correct class.

We group the classes by their frequency in the training set and calculate the average re-classification probability for each group. Specifically, there are three C-level classes: "Mainly Alpha", "Mainly Beta", and "Mixed Alpha/Beta". For A-level classes, we divide them into three categories: 9 classes with over 500K samples (common), 13 classes with 10K–500K samples (regular), and 17 classes with fewer than 10K samples (rare). For T-level classes, we have 31 classes with over 100K samples (common), 237 classes with 5K–100K samples (regular), and 958 with fewer than 5K samples (rare).

As shown in Tab. 11, Proteína can accurately produce the main C classes. At A- and T-level, where we have an increasingly fine spectrum of classes (Fig. 3), the task becomes more challenging, and on average common folds are generated better than rare ones. Considering the imbalanced label distribution with many rare classes, this result is expected. Moreover, re-classification accuracy generally increases with guidance weight $\omega$, validating our tailored CFG scheme (Sec. 3.2). We conclude that while rare classes can be challenging, as expected, the conditioning generally works well for the three C and the common A and T classes.

## D.1    RE-CLASSIFICATION ANALYSIS OF FOLD CLASS-CONDITIONAL CHROMA SAMPLING

As discussed in the main text, we also evaluated Chroma (Ingraham et al., 2023) on fold class-conditional generation (also see App. P). Chroma uses its own CATH fold class label classifier to guide its generation when conditioning on fold classes. We repeated the re-classification analysis for Chroma and report its correct re-classification probabilities in the last row in Tab. 11. We find that Chroma generally performs poorly compared to Proteína. While Proteína can guide into the three main C classes with almost 100% success rate, Chroma struggles to reliably guide into these high-level classes. Furthermore, when guiding with respect to the more fine-grained A and T classes, Chroma's success plummets. This means that Chroma cannot reliably perform fold class conditioning, in contrast to Proteína.

We would also like to comment on Chroma's results in Tab. 2, where it performs competitively with Proteína. This is because the designability, diversity and novelty metrics do not actually test whether correct protein structures given the labels were generated, but these metrics only score the overall set of generated backbones, irrespective of their labels. Only the re-classification analysis conducted here specifically tests the fold class conditioning capabilities in a fine-grained manner.

# E    EQUIVARIANCE ANALYSIS

In this section we study whether our transformer architecture learns a rotationally equivariant vector field. Since the optimal vector field is known to be rotationally equivariant,[3] studying this may yield insights into our method's performance and behavior. We study this empirically for our $\mathcal{M}_{\text{FS}}$ model in the unconditional sampling setting by comparing clean-sample predictions on rotated versions of a noisy/diffused backbone $\mathbf{x}_t$. More specifically, we compute three metrics. The first one is given by

$$\mathcal{E}^r(t) = \mathop{\mathbb{E}}_{\substack{\mathbf{x}\sim p_{\text{data}} \\ \mathbf{x}_t\sim p(\mathbf{x}_t\,|\,\mathbf{x}) \\ R\sim\text{Unif}(\text{SO}(3))}} \Big[\text{RMSD}\left(\hat{\mathbf{x}}(\mathbf{x}_t), R\,\hat{\mathbf{x}}(R^\top\mathbf{x}_t)\right)\Big], \tag{5}$$

where $\hat{\mathbf{x}}(\mathbf{x}_t) = \mathbf{x}_t + (1-t)\,\mathbf{v}_t^\theta(\mathbf{x}_t, \emptyset)$ is the clean sample prediction (since we use the velocity parameterization). This metric compares the outputs of our model with respects to two inputs (noisy backbones) that are the same up to a rotation $R$. A perfectly equivariant model is guaranteed to achieve $\mathcal{E}^r(t) = 0$, as the two outputs would also be equal up to the same rotation $R$. For a non equivariant model, however, we would have $\mathcal{E}^r(t) > 0$, with greater values corresponding to "less equivariant" models.

The second metric we consider is given by

$$\mathcal{E}^u(t) = \mathbb{E}\Big[\text{RMSD}\left(\hat{\mathbf{x}}(\mathbf{x}_t), U\hat{\mathbf{x}}(R^\top\mathbf{x}_t)\right)\Big], \tag{6}$$

where $U$ in Eq. (6) is the rotation that optimally aligns $\hat{\mathbf{x}}(\mathbf{x}_t)$ and $\hat{\mathbf{x}}(R^\top\mathbf{x}_t)$, that is, $U = \arg\min_{A\in\text{SO}(3)} \|\hat{\mathbf{x}}(\mathbf{x}_t) - A\hat{\mathbf{x}}(R^\top\mathbf{x}_t)\|^2$. This metric has two interesting properties. First, a perfectly equivariant model satisfies $U = R$ and $\mathcal{E}^u(t) = 0$. And second, $\mathcal{E}^u(t) \leq \mathcal{E}^r(t)$, with the two metrics being close when the optimal rotation $U \approx R$. Approximately equivariant models should achieve low values for this metric. Additionally, for approximately equivariant models the gap in $\mathcal{E}^u(t) \leq \mathcal{E}^r(t)$ should be small.

Finally, the third metric is given by

$$\mathcal{E}(t) = \mathbb{E}\Big[\text{RMSD}\left(\hat{\mathbf{x}}(\mathbf{x}_t), \hat{\mathbf{x}}(R^\top\mathbf{x}_t)\right)\Big]. \tag{7}$$

In contrast to the first two metrics, $\mathcal{E}(t)$ is minimized by rotationally invariant models (in fact, $\mathcal{E}(t) = 0$ only for such models). In contrast, equivariant or approximately equivariant models should produce larger values for this metric. Intuitively, approximately equivariant models should satisfy $\mathcal{E}^r(t) \ll \mathcal{E}(t)$.

Results for all three metrics as a function of $t$ are shown in Fig. 16. It can be observed that, while greater than zero, our model achieves $\mathcal{E}^u(t) \approx \mathcal{E}^r(t) < 0.5\text{Å}$ for all $t$. This confirms that while our model does not learn a perfectly equivariant vector field, it is approxiamtely equivariant, thanks to the random rotation augmentations applied to clean samples during training. Additionally, as expected for approximately equivariant models, $\mathcal{E}(t)$ is considerably higher than the other two metrics.

It may also be informative to consider the notion of designability (see App. F), which (broadly) deems a backbone designable if there exists a sequence that folds into a structure withing 2Å (RMSD) of the original backbone. The metric $\mathcal{E}^r$ shows that rotating our model predictions accordingly (on rotated inputs) yields RMSDs values below 0.5Å, significantly below the "similarity" threshold used to measure designability.

# F    ESTABLISHED METRICS: DESIGNABILITY, DIVERSITY, NOVELTY & SECONDARY STRUCTURE

We evaluate models using a set of metrics previously established in the literature, including designability, diversity, novelty, and secondary structure content. These metrics are computed across 500 samples, which include 100 proteins at each of the following lengths: 50, 100, 150, 200, and 250.

---

[3]A fact leveraged by many existing methods, which rely on rotationally equivariant architectures (Yim et al., 2023b;a; Lin & Alquraishi, 2023; Lin et al., 2024; Bose et al., 2024).

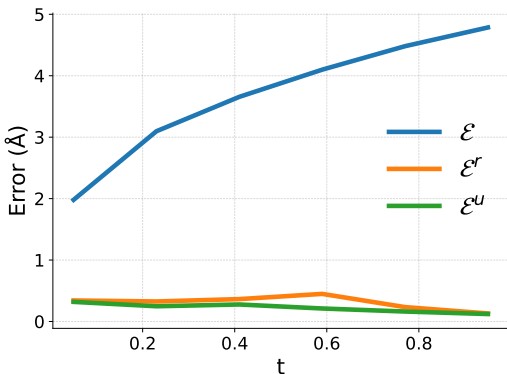

Figure 16: Equivariance analysis. $\mathcal{E}, \mathcal{E}^r$ and $\mathcal{E}^u$ from the captions measure different types of errors, and are formally defined in Eqs. (5), (6) and (7). For a perfectly equivariant model, the green ($\mathcal{E}^r$) and orange ($\mathcal{E}^u$) lines would be exactly zero for all $t$. Approximately equivariant models achieve low values for $\mathcal{E}^r$ and $\mathcal{E}^u$, and large values for $\mathcal{E}$. Our model, despite not being equivariant by construction, follows such a trend.

**Designability.** A protein backbone is considered designable if there exists an amino acid sequence which folds into that structure. Our evaluation of designability follows the methodology outlined by Yim et al. (2023b). For each backbone generated by a model, we produce eight sequences using ProteinMPNN (Dauparas et al., 2022) with a sampling temperature of 0.1. We then predict a structure for each sequence using ESMFold (Lin et al., 2023) and calculate the root mean square deviation (RMSD) between each predicted structure and the model's original structure. A sample is classified as designable if its lowest RMSD—referred to as the self-consistency RMSD (scRMSD)—is under 2Å. The overall designability of a model is computed as the fraction of samples that meet this criterion.

**Diversity (TM-score).** We evaluate diversity in two different ways. The first measure of diversity we report follows the methodology from Bose et al. (2024). For each protein length specified above, we compute the average pairwise TM-score among designable samples, and then aggregate these averages across lengths. Since TM-scores range from zero to one, where higher scores indicate greater similarity, lower scores are preferable for this metric.

**Diversity (Cluster).** The second measure of diversity follows the methodology from Yim et al. (2023b). The designable backbones are clustered based on a TM-score threshold of 0.5. Diversity is then computed by dividing the total number of clusters by the number of designable samples, that is, (number of designable clusters) / (number of designable samples). We perform clustering using Foldseek (van Kempen et al., 2024). Detailed commands for this process are provided in App. F.1. Since more diverse samples imply more clusters, higher scores are preferable for this metric.

**Novelty.** This metric assesses a model's ability to generate structures that are distinct from those in a predefined reference set. For every designable structure we compute its TM-score against each structure in the reference set, tracking the maximum score obtained. We then report the average of these maximum TM-scores (lower is better). In this work we consider two reference sets: the PDB, and $\mathcal{D}_{\text{FS}}$ (Sec. 3.1), the AlphaFold DB subset used by Lin et al. (2024) to train Genie2 (this set was also used to train our $\mathcal{M}_{\text{FS}}$ models). These metrics measure how well a model can produce samples that lack close analogs within the reference sets. We use Foldseek (van Kempen et al., 2024) to evaluate the TM-score of a backbone against these two databases. Detailed commands for this process are provided in App. F.1.

**Secondary structure content.** We use Biotite's (Kunzmann & Hamacher, 2018) implementation of the P-SEA algorithm (Labesse et al., 1997) to analyze the secondary structure content of designable backbones. Specifically, we calculate the proportions of alpha helices ($\alpha$), beta sheets ($\beta$), and coils ($c$) in each sample. The results are reported as normalized values: $\alpha/(\alpha + \beta + c)$ for alpha helices, $\beta/(\alpha + \beta + c)$ for beta sheets, and $c/(\alpha + \beta + c)$ for coils. In the main paper, we sometimes only report the $\alpha$ and $\beta$ percentages in the interest of brevity.

### F.1 FOLDSEEK COMMANDS FOR CLUSTER DIVERSITY AND NOVELTY CALCULATIONS

**Diversity (Cluster).** As mentioned above we use Foldseek to cluster sets of designable backbones. The command used is

```
foldseek easy-cluster <path_samples> <path_tmp>/res <path_tmp>
--alignment-type 1 --cov-mode 0 --min-seq-id 0
--tmscore-threshold 0.5
```

where `<path_samples>` is a directory with all designable samples stored in PDB format and `<path_tmp>` is a directory for temporary files during computation.

**Novelty.** We use Foldseek to evaluate the TM-score of a protein backbone against a reference set. We store the reference sets as Foldseek databases. For the PDB we use Foldseek's precomputed database, and create our own for $\mathcal{D}_{\text{FS}}$. We use the following Foldseek command to compute max TM-scores

```
foldseek easy-search <path_sample> <database_path> <out_file>
<tmp_path> --alignment-type 1 --exhaustive-search
--tmscore-threshold 0.0 --max-seqs 10000000000
--format-output query,target,alntmscore,lddt
```

where `<path_sample>` is the path of the generated structure as a PDB file, `<database_path>` is the path to the Foldseek database, and `<out_file>` and `<tmp_path>` specify the output file and directory for temporary files.

## G   NEW METRICS: FPSD, FS AND FJSD

### G.1   MOTIVATION

Protein structure generators are typically evaluated based on designability, diversity and novelty. Designability measures whether the generated structures can be realistically designed, though with biases inherent in folding and inverse-folding models. While generating diverse and novel proteins is important, these metrics may overlook the quality of the samples—specifically, how closely they resemble realistic proteins. Besides, none of these metrics directly evaluates models at the distribution level, failing to measure how well a model aligns with a reference or target distribution.

To address these limitations, we propose three new metrics that score the learnt distribution rather than individual samples. First, we introduce the Fréchet Protein Structure Distance (FPSD), which compares sets of generated samples to a reference distribution in a non-linear feature space, drawing inspiration from the Fréchet Inception Distance (FID) used in image generation (Heusel et al., 2017). Second, we define the Fold Score (fS), similar to the Inception Score (Salimans et al., 2016), which evaluates both the quality and diversity of generated samples using a trained fold classifier. Finally, we present the Fold Jensen-Shannon Divergence (fJSD) to quantify the similarity of generated samples to reference distributions across predicted fold classes. All new metrics are defined in detail in App. G.2. They all rely on a fold classifier for protein backbones, $p_\phi(\cdot \,|\, \mathbf{x})$, described in App. G.3.

### G.2   METRIC DEFINITION

**Fréchet Protein Structure Distance (FPSD).**   The Fréchet Protein Structure Distance (FPSD) measures the distance between two distributions over protein backbones, the one defined by a generative model and a target reference distribution, leveraging a non-linear feature extractor $\phi(\mathbf{x})$ (in practice, we use the last layer of a fold classifier $p_\phi(\cdot|\mathbf{x})$, see App. G.3).

Let $\{\mathbf{x}\}_{\text{gen}}$ and $\{\mathbf{x}\}_{\text{ref}}$ denote two distributions over protein backbones, defined by a generative model and a reference distribution, respectively. We compute the FPSD between these distributions by measuring the Fréchet Distance between the two Gaussian densities defined as $\mathcal{N}(\boldsymbol{\mu}_{\{\phi(\mathbf{x})\}_{\text{gen}}}, \Sigma_{\{\phi(\mathbf{x})\}_{\text{gen}}})$ and $\mathcal{N}(\boldsymbol{\mu}_{\{\phi(\mathbf{x})\}_{\text{ref}}}, \Sigma_{\{\phi(\mathbf{x})\}_{\text{ref}}})$. In practice this metric is computed following a two-step process:

1. Compute the mean and covariance over features $\phi(\mathbf{x})$ for the generative and reference distributions

$$\boldsymbol{\mu}_{\{\phi(\mathbf{x})\}_{\text{gen}}} = \mathbb{E}_{\mathbf{x} \sim \{\mathbf{x}\}_{\text{gen}}}[\phi(\mathbf{x})], \ \Sigma_{\{\phi(\mathbf{x})\}_{\text{gen}}} = \mathbb{E}_{\mathbf{x} \sim \{\mathbf{x}\}_{\text{gen}}}[(\phi(\mathbf{x}) - \boldsymbol{\mu}_{\{\phi(\mathbf{x})\}_{\text{gen}}})(\phi(\mathbf{x}) - \boldsymbol{\mu}_{\{\phi(\mathbf{x})\}_{\text{gen}}})^\top]$$

$$\boldsymbol{\mu}_{\{\phi(\mathbf{x})\}_{\text{ref}}} = \mathbb{E}_{\mathbf{x} \sim \{\mathbf{x}\}_{\text{ref}}}[\phi(\mathbf{x})], \ \Sigma_{\{\phi(\mathbf{x})\}_{\text{ref}}} = \mathbb{E}_{\mathbf{x} \sim \{\mathbf{x}\}_{\text{ref}}}[(\phi(\mathbf{x}) - \boldsymbol{\mu}_{\{\phi(\mathbf{x})\}_{\text{ref}}})(\phi(\mathbf{x}) - \boldsymbol{\mu}_{\{\phi(\mathbf{x})\}_{\text{ref}}})^\top],$$

2. Measure the Fréchet Distance between the two resulting Gaussian distributions

$$\text{FPSD}(\{\mathbf{x}\}_{\text{gen}}, \{\mathbf{x}\}_{\text{ref}}) := \|\boldsymbol{\mu}_{\{\phi(\mathbf{x})\}_{\text{gen}}} - \boldsymbol{\mu}_{\{\phi(\mathbf{x})\}_{\text{ref}}}\|_2^2 + \text{tr}\left(\Sigma_{\{\phi(\mathbf{x})\}_{\text{gen}}} + \Sigma_{\{\phi(\mathbf{x})\}_{\text{ref}}} - 2(\Sigma_{\{\phi(\mathbf{x})\}_{\text{gen}}}\Sigma_{\{\phi(\mathbf{x})\}_{\text{ref}}})^{\frac{1}{2}}\right).$$

Here, $\|\boldsymbol{\mu}_{\{\phi(\mathbf{x})\}_{\text{gen}}} - \boldsymbol{\mu}_{\{\phi(\mathbf{x})\}_{\text{ref}}}\|_2$ represents the distance between the mean feature vectors, and the trace term captures the differences in covariance matrices. The FPSD reflects how closely the generated structures resemble the reference distribution, as measured by distributional similarity in continuous feature space, with lower values indicating greater similarity.

**Protein Fold Score (fS).** The Protein Fold Score (fS) measures the quality and diversity of generated structures by evaluating how well they align with known fold classes.

Let $\{\mathbf{x}\}_{\text{gen}}$ represent the distribution of generated structures, and let $p_\phi(\cdot|\mathbf{x})$ denote the predicted probability distribution over fold classes for a structure $\mathbf{x}$. The fS is computed in two steps:

1. Compute the marginal distribution over fold classes $p_\phi(\cdot) = \mathbb{E}_{x \sim \{\mathbf{x}\}_{\text{gen}}}[p_\phi(\cdot|\mathbf{x})]$,

2. Calculate the Protein Fold Score

$$\text{fS}(\{\mathbf{x}\}_{\text{gen}}) = \exp\left(\mathbb{E}_{\mathbf{x} \sim \{\mathbf{x}\}_{\text{gen}}}\left[D_{\text{KL}}(p_\phi(\cdot|\mathbf{x})\|p_\phi(\cdot))\right]\right),$$

where $D_{\text{KL}}$ represents the Kullback-Leibler divergence. This score captures the average divergence between the label distribution of each generated sample and the marginal distribution over labels, reflecting both quality and diversity.

A higher fS indicates that the generated protein structures are not only of high quality individually, but also exhibit a diverse range of fold classes, capturing the richness of the generated distribution. Note that fS is calculated separately at the different levels of the label hierarchy, i.e., separately for the C-, A- and T-level classes.

**Protein Fold Jensen-Shannon Divergence (fJSD).** The Protein Fold Jensen-Shannon Divergence (fJSD) quantifies the similarity between the predicted label distribution of generated protein structures and that of a reference set, both derived from the same fold classifier.

Let $\{\mathbf{x}\}_{\text{gen}}$ and $\{\mathbf{x}\}_{\text{ref}}$ represent the distributions of generated and reference structures, respectively, and let $p_\phi(\cdot|\mathbf{x})$ denote the predicted probability distribution over fold classes for a structure $\mathbf{x}$. The fJSD metric is computed in two steps:

1. Compute the marginal predicted distribution over fold classes for the generative and reference distributions

$$p_{\text{gen}}(\cdot) = \mathbb{E}_{x \sim \{\mathbf{x}\}_{\text{gen}}}[p_\phi(\cdot|\mathbf{x})] \quad \text{and} \quad p_{\text{ref}}(\cdot) = \mathbb{E}_{x \sim \{\mathbf{x}\}_{\text{ref}}}[p_\phi(\cdot|\mathbf{x})],$$

2. Calculate the Protein Fold Jensen-Shannon Divergence

$$\text{fJSD}(\{\mathbf{x}\}_{\text{gen}}, \{\mathbf{x}\}_{\text{ref}}) = 10 \times D_{\text{JS}}(p_{\text{gen}}(\cdot)\|p_{\text{ref}}(\cdot)), \tag{8}$$

where $D_{\text{JS}}$ denotes the Jensen-Shannon divergence, defined as

$$D_{\text{JS}}(P\|Q) = \frac{1}{2}D_{\text{KL}}(P\|M) + \frac{1}{2}D_{\text{KL}}(Q\|M), \tag{9}$$

with $M = \frac{1}{2}(P + Q)$. In our case, $P$ represents the distribution $p_{\text{gen}}(\cdot)$ and $Q$ represents the distribution for the reference set $p_{\text{ref}}(\cdot)$. Since the Jensen-Shannon divergence is upper bounded by 1, we multiply it by a factor of 10 for easier reporting of the results.

Lower values of fJSD indicate that the predicted label distribution of generated proteins closely aligns with that from the reference set, reflecting higher fidelity to the expected fold classes. In contrast to FPSD, which measures the similarity between the generated and reference distributions in continuous feature space, fJSD measures the similarity in the categorical label space from the fold classifier. As we empirically find that fJSD values calculated for the C-, A-, and T-level label distributions yield the same ranking across different methods, we decide to report the final metric values as the average of the fJSD scores at the C-, A-, and T-levels. We note, however, that this metric can be reported separately for each of the C, A, and T-levels.

**Reference Datasets.** To evaluate FPSD and fJSD, we construct two reference datasets, one for the PDB and another one for the AFDB. For the PDB reference set, we curate a high-quality single-chain dataset by applying several filters to the PDB: a minimum residue length of 50, a maximum residue length of 256, a resolution threshold of 5.0 Å, a maximum coil proportion of 0.5, and a maximum radius of gyration of 3.0 nm. We then cluster the dataset based on a sequence identity of 50% and select the cluster representatives, resulting in 15,357 samples. For the AFDB reference set, we directly use the Foldseek AFDB clusters, denoted by $\mathcal{D}_{FS}$ in the main text.

These metrics are evaluated independently of existing metrics based on a different set of generated samples. We randomly sample 125 proteins at each length from 60 to 255 residues, with a step size of 5. We use *all* the 5,000 produced samples, *without any designability filter*, for evaluation.

### G.3 Fold Classifier Training

A crucial aspect of defining the new metrics is developing an accurate fold classifier $p_\phi(\cdot|\mathbf{x})$ which embeds alpha-carbon-only structures into the feature space $\phi(\mathbf{x})$. In this subsection we give details behind the classifier we use, including the dataset it is trained on and its architecture.

#### G.3.1 Dataset Processing

For training the classifier, CATH structural labels are utilized for protein domain annotation (Dawson et al., 2016) which includes *C* (class), *A* (architecture), and *T* (topology/fold) labels. We exclude *H* (homologous superfamily) labels to ensure that our classification is based solely on structures.

We extract chains from the PDB dataset, with structures filtered to include a minimum length of 50 residues, a maximum length of 1000 residues, and a maximum oligomeric state of 10. We also discard proteins with a resolution worse than 5Å and those lacking CATH labels. This results in a total of 214,564 structures, categorized into 5 C classes, 43 A classes, and 1,336 T classes. The dataset is randomly divided into training, validation, and test sets at a ratio of 8:1:1, ensuring that at least one protein from each class is included in the test set whenever possible. While the paper primarily focuses on the three main C-level classes ("mainly alpha", "mainly beta", "mixed alpha/beta"), as they are the most interesting and relevant to our study, we still train the classifier on all C-level classes. This ensures the metrics are universally applicable and can be used for future analyses involving any of the C-level classes.

Given that the CATH database annotates protein domains, some proteins may have multiple domains, thus multiple CATH labels. For these proteins, we randomly sample one domain label as the ground truth during training and encourage the model to predict equal probabilities for the labels of all domains. During testing, predicting any of the correct labels is considered a good prediction.

#### G.3.2 GearNet-Based Fold Classifier

To build the fold classifier $p_\phi(\cdot|\mathbf{x})$, we utilize an SE(3)-invariant network, GearNet (Zhang et al., 2023), as our feature extractor $\phi(\mathbf{x})$. GearNet is a geometric relational graph convolutional network specifically designed for protein structure modeling, making it ideal for tasks such as protein classification and fold prediction. While the original GearNet architecture processes both structural and sequential data, we modify it to focus solely on predicting fold classes based on structure. The model components are detailed as follows:

1. **Input and Embedding Layer:** Each $C_\alpha$ atom is treated as a node, and the node features are constructed by concatenating a 256-dimensional atom type embedding (generally corresponding to the $C_\alpha$ atom embedding) with a 256-dimensional sinusoidal positional embedding based on sequence indices.

2. **Graph Construction:** A multi-relation graph is built using both sequential and spatial information. Sequential relations are established by connecting neighboring atoms within a relative sequence distance between -2 an 2, with each relative distance treated as a distinct relation type. Spatial relations connect atoms within a Euclidean distance of 10Å. In total, the graph uses five sequential relation types and one spatial relation type, allowing the model to capture diverse interaction patterns between residues based on both sequence proximity and spatial context.

Table 12: Summary of metric validation experiments.

| Setting | Distribution | Expected Results | Results |
|---|---|---|---|
| *Protein Fold Score (fS)* | | | |
| Balanced dataset | Diverse and balanced label distribution | High fS | Fig. 17 (blue) |
| Homogeneous dataset | Homogeneous label distribution | Low fS | Fig. 17 (red) |
| Imbalanced dataset | Diverse but imbalanced label distribution | Medium fS | Fig. 17 (green) |
| Imbalanced noisy dataset | Noisy and imbalanced distribution | Decreasing fS | Fig. 17 (green) |
| Unseen noisy dataset | Noisy distribution with unseen samples | Decreasing fS | Fig. 17 (orange) |
| *Fréchet Protein Structure Distance (FPSD) and Protein Fold Jensen-Shannon Divergence (fJSD)* | | | |
| Disjoint split datasets | Different structure distributions | High FPSD and fJSD | Fig. 18 (blue) |
| Random split datasets | Similar structure distributions | Low FPSD and fJSD | Fig. 18 (green) |
| Random split noisy datasets | Noisy distributions with seen samples | Increasing FPSD and fJSD | Fig. 18 (green) |
| Unseen random split noisy datasets | Noisy distributions with unseen samples | Increasing FPSD and fJSD | Fig. 18 (orange) |

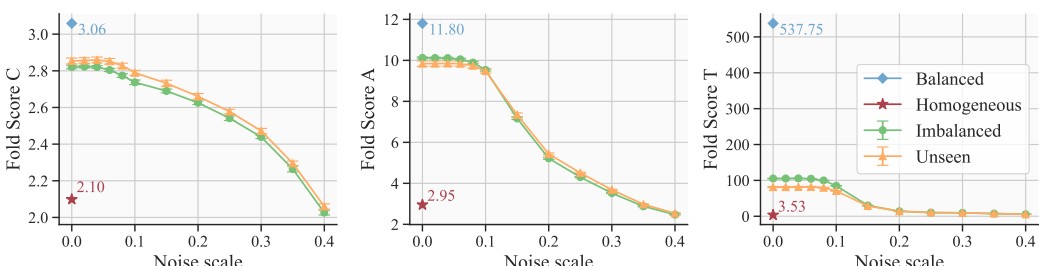

Figure 17: Fold Scores (C/A/T) metrics on balanced, homogeneous, imbalanced and unseen subsets of the PDB dataset, with varying levels of Gaussian noise (0.0 to 0.4Å) applied on the latter two.

3. **Edge and Message Passing:** Edge features are generated using a radial basis function (RBF) to capture spatial distance-based relationships, along with relative sequential positional encoding between atoms. Both features are 128-dimensional. Additionally, we incorporate clockwise angular features to break reflection symmetries.

4. **Relational Graph Convolution Layers:** The model includes 8 layers of Geometric Relational Graph Convolution, each aggregating information from neighboring atoms using the node and edge features. These layers employ MLPs to process inputs and update node representations, ensuring that the model captures different types of relational patterns between atoms.

5. **Output and Prediction:** After the convolutional layers, the atom features are aggregated using sum pooling to create a global protein representation. This global feature is further refined through an MLP layer. For classification, the model includes separate output heads for predicting three levels of CATH labels: T, A, and C, with output sizes of 1336, 43, and 5 classes, respectively.

Throughout the model, a dropout rate of 0.2 is applied to prevent overfitting, and leaky ReLU activation functions with a slope of 0.1 are used. The model is trained using the Adam optimizer with a learning rate of 0.0001, distributed across 8 GPUs with a batch size of 8 and a gradient accumulation step of 2. Training is run over 70,000 parameter update steps. On the test set, the model achieves a Micro Accuracy (Grandini et al., 2020) of 97.8% at the T-level, 98.1% at the A-level, and 99.2% at the C-level. Given the highly imbalanced nature of the CATH classes, we also report Macro Accuracy, achieving 94.0% at the T-level, 97.5% at the A-level, and 95.6% at the C-level. These results demonstrate that the classifier is highly effective in accurately predicting the fold labels of protein structures.

## G.4 METRIC VALIDATION

To validate the effectiveness of our metrics, we create two sets of experiments to observe the behavior of fS and FPSD, fJSD under different settings. We summarize these experiments, together with their expected results, in Tab. 12.

**Protein Fold Score (fS) Validation.** Using the PDB training dataset, we create three subsets to assess the behavior of the Protein Fold Score. All experiments are repeated with 20 different random seeds.

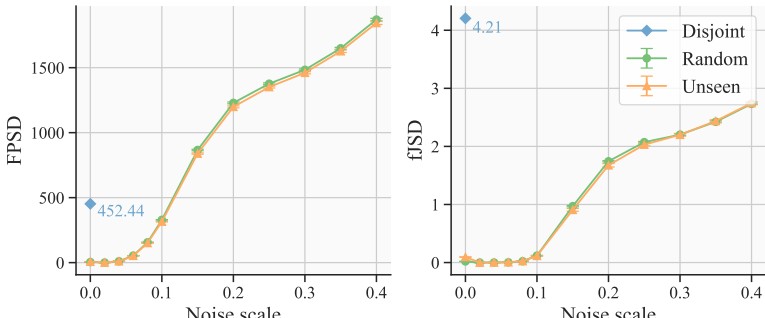

Figure 18: FPSD and fJSD metrics on *(i)* the fold-disjoint and *(ii)* random splits of the PDB training set, and *(iii)* a random split of the unseen PDB set, with Gaussian noise (0.0 to 0.4Å) applied to the latter two.

1. *Fold Class-Balanced Subset*: We randomly sample 300 T-level classes and then randomly sample approximately 16 proteins per class (with replacement) to create a total of 5,000 samples. This subset tests whether fS rewards a diverse, realistic, and class-balanced structure distribution.

2. *Homogeneous Subset*: We randomly sample 4 T-level classes and 1,250 proteins per class. This subset is designed to test whether fS penalizes distributions lacking fold diversity.

3. *Fold Class-Imbalanced Subset*: We randomly sample 5,000 proteins from the PDB dataset. Given PDB's inherent class imbalance (Fig. 3), this random sampling leads to a diverse but imbalanced distribution, so we expect this to lead to "intermediate" values for the metric.

Results for these three subsets are shown in Fig. 17. The results show exactly the expected behavior for the fS metric, with the *Fold Class-Balanced Subset* obtaining the highest score, the *Homogeneous Subset* the lowest, and the *Fold Class-Imbalanced Subset* standing in between these two extremes.

We additionally assess whether our metric is robust to noisy structures and structures unseen in the classifier's training dataset. For the former, we continue using the previously defined *Fold Class-Imbalanced Subset* and gradually add Gaussian noise to all structures, with the noise scale increasing from 0.0 to 0.4Å. We expect the fS score to decrease as the scale of the noise increases. For unseen structures, we randomly sample 5,000 structures from the full PDB dataset, without applying the CATH label filter, and apply Gaussian noise in the same manner.

We evaluate the Fold Score C/A/T on these noisy datasets, with the results shown in Fig. 17 (green and orange curves). As expected, as the noise scale increases, the quality of protein structures declines, leading to reduced classifier confidence and a corresponding gradual decrease in the Fold Score.

Overall, we find the Protein Fold Score is able to effectively measure the realism, diversity, and balance of distributions over protein structures. It remains robust to unseen samples and deteriorates gracefully for noisy samples, effectively detecting the lower quality of the noisy samples.

**Fréchet Protein Structure Distance (FPSD) and Protein Fold Jensen-Shannon Divergence (fJSD) Validation.** Similar to the fS validation, we curate two dataset splits based on the PDB training set, and measure FPSD and fJSD for the two splits.

1. *Fold Class-Disjoint Split*: We randomly draw 5,000 samples for each split, ensuring no overlap at A-level classes between the two splits. This setup tests whether FPSD and fJSD can distinguish different distributions. We exclude T-level classes here, as they are too fine-grained to produce sufficiently distinct distributions. We expect this split to yeild large values for both metrics.

2. *Random Split*: We randomly sample 5,000 proteins for each split from the PDB dataset. Since no constraints are applied during the split, both datasets are expected to follow the same distribution, and thus we expect low values for both metrics for this split.

We show results in Fig. 18, where we can observe that the metrics behave as expected for both splits. Specifically, the *Fold Class-Disjoint Split* yields a FPSD of 452.44 and fJSD of 4.21, while the *Random Split* yields significantly lower values for both metrics; $\approx 10$ for FPSD and $\approx 0$ for fJSD.

Table 13: FrameFlow's, RFDiffusion's, Genie2's and three variants of Proteína's cluster diversity values with # designable clusters in parentheses *under different TM-score clustering thresholds*. Best scores are **bold**.

| Threshold | 0.1 | 0.2 | 0.3 | 0.4 | 0.5 | 0.6 | 0.7 | 0.8 | 0.9 |
|---|---|---|---|---|---|---|---|---|---|
| FrameFlow | 0.051 (23) | 0.054 (24) | 0.067 (30) | 0.237 (105) | 0.523 (232) | 0.808 (358) | 0.934 (414) | **0.993 (440)** | 0.997 (442) |
| RFDiffusion | 0.046 (22) | 0.046 (22) | 0.048 (23) | 0.127 (60) | 0.447 (211) | 0.720 (340) | 0.834 (394) | 0.887 (419) | 0.936 (442) |
| Genie2 | 0.060 (29) | 0.060 (29) | 0.060 (29) | 0.144 (69) | 0.596 (284) | **0.915 (436)** | **0.978 (466)** | 0.991 (472) | **1.000 (476)** |
| $\mathcal{M}_{FS}$, $\gamma$=0.45 | **0.101 (49)** | **0.101 (49)** | **0.118 (57)** | **0.302 (146)** | **0.639 (308)** | 0.852 (411) | 0.943 (455) | 0.981 (473) | 0.995 (480) |
| $\mathcal{M}_{FS}^{\text{no-tri}}$, $\gamma$=0.45 | 0.083 (39) | 0.089 (42) | 0.095 (45) | 0.257 (121) | 0.626 (294) | 0.820 (385) | 0.916 (430) | 0.980 (460) | 0.993 (466) |
| $\mathcal{M}_{21M}$, $\gamma$=0.3 | 0.056 (28) | 0.060 (30) | 0.064 (32) | 0.147 (73) | 0.305 (151) | 0.442 (219) | 0.569 (282) | 0.723 (358) | 0.901 (446) |

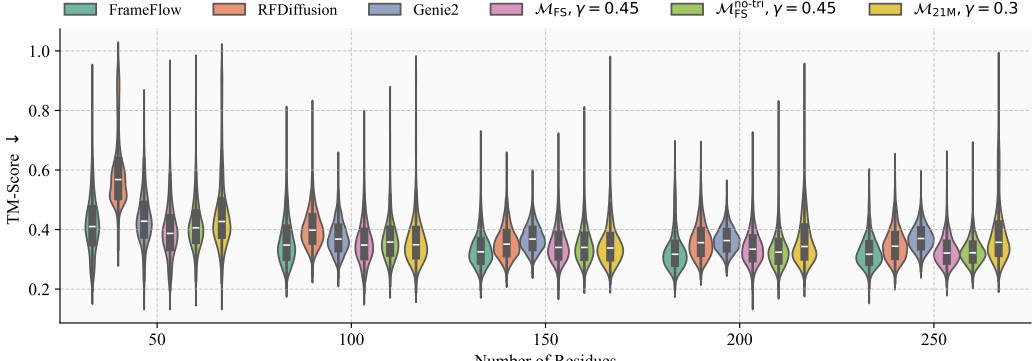

Figure 19: Pairwise TM-Score distributions of FrameFlow, RFDiffusion, Genie2 and three variants of Proteína across different residue lengths, with lower TM-Scores indicating better performance.

Additionally, we apply the same process used in the fS validation to create noisy and unseen dataset splits for testing the FPSD and fJSD metrics, computing them between the noisy and original datasets. The results in Fig. 18 (green and orange) indicate that as the noise scale increases, protein structure quality deteriorates, leading to increasignly higher FPSD and fJSD values.

In summary, FPSD and fJSD effectively recognize similarities and differences between structure distributions, remain robust to unseen samples and detect increasingly noisy samples.

# H ANALYSIS AND VALIDATION OF METRICS CALCULATIONS

## H.1 FINE-GRAINED DIVERSITY EVALUATIONS

**Cluster-based diversity with different thresholds.** We evaluate the cluster-based diversity metric under varying clustering thresholds for three of our best models and the most relevant and competitive baselines—Genie2, RFDiffusion, and FrameFlow—as shown in Tab. 13. For looser thresholds, our $\mathcal{M}_{FS}$ ($\gamma$=0.45) model outperforms the others. However, with very strict clustering thresholds, all models and baselines produce highly diverse results, covering a wide range of distinct clusters.

**Distribution of pairwise TM-scores.** The other diversity metric that we use in this work are average pairwise TM-scores. Here, we analyze the distributions of the pairwise TM-scores for samples generated by our models and the three most relevant baselines mentioned above. We draw violin plots across different lengths in Fig. 19. The results demonstrate that all models maintain reasonable TM-score distributions, showing no signs of mode collapse.

## H.2 EVALUATION OF METRICS FOR REFERENCE DATASETS

To provide reference values for our results in Tab. 1, we report metrics for two representative protein structure databases: the PDB (natural proteins) and the AFDB (synthetic proteins predicted by AlphaFold2). We use the representative subsets of the PDB and AFDB processed in App. G.2 as our reference distributions. Following the protocols outlined in App. F and App. G, we sample from these reference datasets and evaluate all metrics. The results, presented in Tab. 14, show novelty values close to 1, and very low FPSD and fJSD values, as expected. The designability of the two reference datasets aligns with previously reported results in Lin et al. (2024), with the AFDB exhibiting lower

Table 14: Reference metrics by sampling two reference datasets (AFDB and PDB) introduced in App. G.2.

| Dataset | Design-ability (%)↑ | Diversity Cluster↑ | TM-Sc.↓ | Novelty vs. PDB↓ | AFDB↓ | FPSD vs. PDB↓ | AFDB↓ | fS (C / A / T)↑ | fJSD vs. PDB↓ | AFDB↓ | Sec. Struct. % (α / β) |
|---|---|---|---|---|---|---|---|---|---|---|---|
| PDB ref. | 67.2 | 0.62 (209) | 0.31 | 0.99 | 0.87 | 3.433 | 71.20 | 2.94 / 9.48 / 90.45 | 0.05 | 0.51 | 35.0 / 16.8 |
| AFDB ref. | 33.6 | 0.91 (154) | 0.29 | 0.76 | 1.00 | 73.02 | 3.011 | 2.71 / 6.57 / 34.88 | 0.53 | 0.02 | 44.9 / 12.7 |

Table 15: Proteína's unconditional generation results *under different random sampling seeds*.

| Seed | Design-ability (%)↑ | Diversity Cluster↑ | TM-Sc.↓ | Novelty vs. PDB↓ | AFDB↓ | FPSD vs. PDB↓ | AFDB↓ | fS (C / A / T)↑ | fJSD vs. PDB↓ | AFDB↓ | Sec. Struct. % (α / β) |
|---|---|---|---|---|---|---|---|---|---|---|---|
| *Unconditional generation with Proteína model $\mathcal{M}_{FS}^{no\text{-}tri}$ and noise scale $\gamma = 0.45$.* | | | | | | | | | | | |
| 0 | 93.6 | 0.62 (294) | 0.36 | 0.70 | 0.76 | 322.2 | 306.1 | 1.80 / 4.72 / 18.59 | 1.85 | 1.36 | 70.5 / 6.1 |
| 1 | 92.8 | 0.61 (283) | 0.36 | 0.70 | 0.75 | 327.0 | 313.0 | 1.78 / 4.64 / 18.83 | 1.85 | 1.39 | 70.7 / 6.2 |
| 2 | 94.8 | 0.60 (283) | 0.36 | 0.69 | 0.75 | 318.4 | 303.2 | 1.83 / 4.74 / 19.35 | 1.79 | 1.34 | 71.5 / 5.3 |
| 3 | 94.4 | 0.58 (272) | 0.37 | 0.71 | 0.76 | 325.1 | 308.0 | 1.88 / 4.60 / 18.40 | 1.88 | 1.40 | 71.1 / 5.2 |
| 4 | 93.0 | 0.57 (263) | 0.37 | 0.71 | 0.76 | 316.6 | 300.3 | 1.79 / 4.70 / 18.93 | 1.83 | 1.38 | 71.6 / 4.9 |
| Mean | 93.7 | 0.60 (279) | 0.36 | 0.70 | 0.76 | 321.9 | 306.1 | 1.80 / 4.68 / 18.82 | 1.84 | 1.37 | 71.1 / 5.5 |
| Std. Dev. | 0.9 | 0.02 (12) | 0.01 | 0.01 | 0.01 | 4.4 | 4.8 | 0.02 / 0.06 / 0.36 | 0.03 | 0.03 | 0.5 / 0.6 |

designability due to the use of sequence clustering to avoid over-represented clusters. Additionally, we observe that the reference datasets display higher diversity than all models and baselines. This suggests that existing models still have room for improvement in optimizing diversity.

Note that our models and the baselines are able to achieve much higher designability among the generated proteins due to noise or temperature scaling during inference (or rotation schedule annealing), effectively modifying the generated distribution towards more well-structured protein backbones.

### H.3 STATISTICAL VARIATION OF METRICS

To assess the statistical stability of our results, we select the model $\mathcal{M}_{FS}^{no\text{-}tri}$ ($\gamma$=0.45) for efficiency and repeat the metrics evaluations five times using different random seeds. The results, presented in Tab. 15, show minimal variance across all metrics, indicating robustness of our evaluation process.

## I SAMPLING, AUTOGUIDANCE AND HIERARCHICAL FOLD CLASS GUIDANCE

Here, we provide more details about Proteína's inference-time sampling as well as our new hierarchical fold class guidance and autoguidance.

### I.1 ODE AND SDE SAMPLING

Generally, flow-matching (Lipman et al., 2023; Albergo & Vanden-Eijnden, 2023; Liu et al., 2023; Albergo et al., 2023) models like Proteína rely on a vector field $\mathbf{u}_t(\mathbf{x}_t)$ that describes the probability flow between noise and data (see Sec. 2). The default way to generate samples is to solve the flow's ODE

$$\mathrm{d}\mathbf{x}_t = \mathbf{u}_t(\mathbf{x}_t)\mathrm{d}t \tag{10}$$

from $t = 0$ to $t = 1$, initialized from random noise. In our case, the noise corresponds to a standard Gaussian prior with unit variance over the 3D $C_\alpha$ coordinates of the protein backbone. As discussed in Sec. 2, the flow's intermediate states $\mathbf{x}_t$ are in practice constructed through an interpolant between data $\mathbf{x}_1 \sim p(\mathbf{x}_1)$ and the Gaussian random noise distribution $\boldsymbol{\epsilon} \sim \mathcal{N}(\mathbf{0}, \boldsymbol{I})$, which takes the general form

$$\mathbf{x}_t = \alpha_t \mathbf{x}_1 + \sigma_t \boldsymbol{\epsilon} \tag{11}$$

for the time-dependent scaling and standard deviation coefficients $\alpha_t$ and $\sigma_t$, respectively. In our work, we rely on the rectified flow (Liu et al., 2023) (also known as conditional optimal transport (Lipman et al., 2023)) formulation, corresponding to a linear interpolant

$$\mathbf{x}_t = t\mathbf{x}_1 + (1 - t)\boldsymbol{\epsilon} \tag{12}$$

between noise $\boldsymbol{\epsilon}$ and data samples $\mathbf{x}_1$. This leads to the corresponding marginal vector field at intermediate $\mathbf{x}_t$

$$\mathbf{u}_t(\mathbf{x}_t) = \frac{\partial}{\partial t}\left(\mathbb{E}[t\mathbf{x}_1|\mathbf{x}_t] + \mathbb{E}[(1-t)\boldsymbol{\epsilon}|\mathbf{x}_t]\right) = \mathbb{E}[\mathbf{x}_1|\mathbf{x}_t] - \mathbb{E}[\boldsymbol{\epsilon}|\mathbf{x}_t]. \tag{13}$$

Further, we have that (see, e.g., proof in Ma et al. (2024))

$$\mathbf{s}_t(\mathbf{x}_t) := \nabla_{\mathbf{x}_t} \log p_t(\mathbf{x}_t) = -\frac{1}{1-t}\mathbb{E}[\boldsymbol{\epsilon}|\mathbf{x}_t]. \tag{14}$$

This allows one to derive a relation between $\mathbf{u}_t(\mathbf{x}_t)$ and $\mathbf{s}_t(\mathbf{x}_t)$:

$$\begin{aligned}
\mathbf{u}_t(\mathbf{x}_t) &= \mathbb{E}[\mathbf{x}_1|\mathbf{x}_t] - \mathbb{E}[\boldsymbol{\epsilon}|\mathbf{x}_t] \\
&= \frac{\mathbf{x}_t - (1-t)\mathbb{E}[\boldsymbol{\epsilon}|\mathbf{x}_t]}{t} - \mathbb{E}[\boldsymbol{\epsilon}|\mathbf{x}_t] \\
&= \frac{\mathbf{x}_t}{t} - \frac{1}{t}\mathbb{E}[\boldsymbol{\epsilon}|\mathbf{x}_t] \\
&= \frac{\mathbf{x}_t}{t} + \frac{1-t}{t}\mathbf{s}_t(\mathbf{x}_t),
\end{aligned} \tag{15}$$

or

$$\mathbf{s}_t(\mathbf{x}_t) = \frac{t\mathbf{u}_t(\mathbf{x}_t) - \mathbf{x}_t}{1-t}. \tag{16}$$

Furthermore, given the score of the probability path $p_t(\mathbf{x}_t)$, we can now obtain the Langevin dynamics SDE

$$d\mathbf{x}_t = \mathbf{s}_t(\mathbf{x}_t)dt + \sqrt{2}\,d\mathcal{W}_t, \tag{17}$$

with the Wiener process $\mathcal{W}_t$. This SDE, when simulated, in principle samples from $p_t(\mathbf{x}_t)$ at any fixed $t$. We can combine this with the flow's ODE (Eq. (10)) to obtain the SDE

$$d\mathbf{x}_t = \mathbf{u}_t(\mathbf{x}_t)dt + g(t)\mathbf{s}_t(\mathbf{x}_t)dt + \sqrt{2g(t)}\,d\mathcal{W}_t, \tag{18}$$

which, for any $g(t) \geq 0$, now simulates the stochastic flow along the marginal probability path $p_t(\mathbf{x}_t)$ from $t = 0$ to $t = 1$ with the stochastic paths due to the Langevin dynamics component.

In practice, we model $\mathbf{u}_t(\mathbf{x}_t)$ by the learnt neural network $\mathbf{v}_t^\theta(\mathbf{x}_t, \tilde{c})$ with parameters $\theta$, which we can use to obtain the corresponding learnt score $\mathbf{s}_t^\theta(\mathbf{x}_t, \tilde{c})$, using Eq. (16) above. As in the main paper, $\tilde{c}$ represents all conditioning information we may use in practice. Hence,

$$d\mathbf{x}_t = \mathbf{v}_t^\theta(\mathbf{x}_t, \tilde{c})dt + g(t)\mathbf{s}_t^\theta(\mathbf{x}_t, \tilde{c})dt + \sqrt{2g(t)}\,d\mathcal{W}_t. \tag{19}$$

Importantly, in practice the velocity and score are neural network-based approximations with respect to their ground truths, and the SDE is numerically discretized. Therefore, different choices of $g(t)$, which scales the stochastic Langevin component, can lead to different results in practice. For $g(t) = 0$, we recover the default flow ODE.

Moreover, we introduce a noise scale $\gamma$, as is common in generative models of protein structures, and in practice use the generative SDE

$$d\mathbf{x}_t = \mathbf{v}_t^\theta(\mathbf{x}_t, \tilde{c})dt + g(t)\mathbf{s}_t^\theta(\mathbf{x}_t, \tilde{c})dt + \sqrt{2g(t)\gamma}\,d\mathcal{W}_t. \tag{20}$$

For $\gamma = 1$, we simulate the "proper" marginal probability path, while lowering the noise scale often reduces the diversity of the generated results, oversampling the model's main modes. Although not principled, this can be empirically beneficial in protein structure generation, as the tails of the distribution can consist of undesired samples that, for instance, may not be designable. Since our models operate directly in 3D space, reduced noise injection results in more globular and well-structured backbones that tend to be more designable. In the main paper, we typically either use the default ODE for generation (corresponding to $g(t) = 0$) or the SDE with a reduced noise scale $\gamma < 1$ and some stochasticity schedule $g(t) > 0$. Moreover, in practice it can be sensible to only simulate with $g(t) > 0$ up to some cutoff time $t < 1$, due to the diverging denominator in Eq. (16), required to calculate the score from the vector field when using $g(t) > 0$.

## I.2 CLASSIFIER-FREE GUIDANCE AND AUTOGUIDANCE

As mentioned in the main text, we are leveraging both classifier-free guidance (CFG) (Ho & Salimans, 2021) and autoguidance (Karras et al., 2024) in selected experiments in this paper. To the best of our knowledge, neither of the two methods have been explored in flow- or diffusion-based protein

structure generation. In both approaches, different scores are combined to obtain a "higher quality" score that leads to improved samples.

Let us assume we have access to the densities $p_t^A(\mathbf{x}_t)$ and $p_t^B(\mathbf{x}_t)$. Now let us define the "guided" score (Karras et al., 2024)

$$\mathbf{s}_t^{\text{guided}}(\mathbf{x}_t) := \nabla_{\mathbf{x}_t} \log p_t^\omega(\mathbf{x}_t) := \nabla_{\mathbf{x}_t} \log \left( p_t^B(\mathbf{x}_t) \left[ \frac{p_t^A(\mathbf{x}_t)}{p_t^B(\mathbf{x}_t)} \right]^\omega \right), \tag{21}$$

where $\omega \geq 0$ denotes the guidance weight. In practice, $p_t^A(\mathbf{x}_t)$ corresponds to the density we are primarily interested in and for $\omega = 1$ we recover $\mathbf{s}_t^{\text{guided}}(\mathbf{x}_t) = \nabla_{\mathbf{x}_t} \log p_t^A(\mathbf{x}_t)$. But if we now choose a density $p_t^B(\mathbf{x}_t)$ that is more spread out than $p_t^A(\mathbf{x}_t)$, then, for $\omega > 1$, the term

$$\left[ \frac{p_t^A(\mathbf{x}_t)}{p_t^B(\mathbf{x}_t)} \right]^\omega \tag{22}$$

is typically $> 1$ for $\mathbf{x}_t \sim p_t^A(\mathbf{x}_t)$ and the overall score is essentially scaled according to the ratio between $p_t^A(\mathbf{x}_t)$ and $p_t^B(\mathbf{x}_t)$. This can be leveraged to construct a guided $\mathbf{s}_t^{\text{guided}}(\mathbf{x}_t)$ that emphasizes the difference between $p_t^A(\mathbf{x}_t)$ and $p_t^B(\mathbf{x}_t)$, as we now explain (see Karras et al. (2024) for details).

In *classifier-free guidance* (Ho & Salimans, 2021), $p_t^A(\mathbf{x}_t)$ corresponds to a conditional density and $p_t^B(\mathbf{x}_t)$ to a unconditional one and $\omega > 1$ emphasizes the conditioning information, leading to samples that are more characteristic for the given class, and often also improving quality. *Autoguidance* (Karras et al., 2024) disentangles the effects of improved class adherence and improved quality and instead uses for $p_t^B(\mathbf{x}_t)$ a "bad" version of $p_t^A(\mathbf{x}_t)$ that is trained for fewer steps or uses a smaller, less expressive network. Due to the maximum likelihood-like objectives of diffusion and flow models their learnt densities generally tend to be mode covering and can be somewhat broader than the ideal target density. Hence, even the "good" $p_t^A(\mathbf{x}_t)$ will usually not be a "perfect" model that models the distribution of interest perfectly and the "bad" $p_t^B(\mathbf{x}_t)$ will make the same errors like $p_t^A(\mathbf{x}_t)$, but stronger and the density will be even broader. Guidance with $\omega > 1$ then emphasizes the quality difference between $p_t^A(\mathbf{x}_t)$ and $p_t^B(\mathbf{x}_t)$ and can result in sharper outputs, essentially extrapolating in distribution space beyond $p_t^A(\mathbf{x}_t)$ towards the true desired distribution of interest.

CFG also often improves quality because a similar effect happens there, just entangled with the conditioning: The unconditional density used in CFG also represents a broader density than the conditional one, which means the guided score does not only emphasize the class conditioning, but also pushes samples towards modes, which often correspond to less diverse, but high-quality samples. Note that autoguidance is general and can be used both for conditional and unconditional generation, whereas classifier-free guidance contrasts a conditional and an unconditional model and hence is only applicable when such a conditional model is available.

Let us now derive the exact guidance equation used in our work. Decomposing the logarithm term in Eq. (21) yields

$$\mathbf{s}_t^{\text{guided}}(\mathbf{x}_t) := \omega \nabla_{\mathbf{x}_t} \log p_t^A(\mathbf{x}_t) + (1-\omega) \nabla_{\mathbf{x}_t} \log p_t^B(\mathbf{x}_t), \tag{23}$$

or short

$$\mathbf{s}_t^{\text{guided}}(\mathbf{x}_t) := \omega \, \mathbf{s}_t^A(\mathbf{x}_t) + (1-\omega) \mathbf{s}_t^B(\mathbf{x}_t). \tag{24}$$

Inserting Eq. (16) that relates the score and the vector field, we find that the vector fields obey the analogous equation

$$\mathbf{u}_t^{\text{guided}}(\mathbf{x}_t) := \omega \, \mathbf{u}_t^A(\mathbf{x}_t) + (1-\omega) \mathbf{u}_t^B(\mathbf{x}_t). \tag{25}$$

In practice, we use learnt conditional models. And in that case, we can now introduce the interpolation parameter $\alpha \in [0, 1]$ that interpolates between classifier-free guidance and autoguidance in a unified formulation (analogous to Karras et al. (2024) in their Appendix B.2). We get

$$\mathbf{v}_t^{\theta,\text{guided}}(\mathbf{x}_t, \tilde{c}) = \omega \, \mathbf{v}_t^\theta(\mathbf{x}_t, \tilde{c}) + (1-\omega) \left[ (1-\alpha)\mathbf{v}_t^\theta(\mathbf{x}_t, \emptyset) + \alpha \, \mathbf{v}_t^{\theta,\text{bad}}(\mathbf{x}_t, \tilde{c}) \right], \tag{26}$$

where $\mathbf{v}_t^\theta(\mathbf{x}_t, \tilde{c})$ is the main model with conditioning $\tilde{c}$, $\mathbf{v}_t^\theta(\mathbf{x}_t, \emptyset)$ corresponds to the unconditional version, and $\mathbf{v}_t^{\theta,\text{bad}}(\mathbf{x}_t, \tilde{c})$ denotes the "bad" model required for autoguidance. For $\alpha = 0$, we get regular classifier-free guidance, while for $\alpha = 1$, we get regular autoguidance. In that case, setting $\tilde{c} = \emptyset$, i.e. autoguidance for unconditional modeling, is still applicable. In this paper, we do not use

any intermediate $\alpha$, but only explore either pure CFG or pure autoguidance, though. An analogous formula can be written for the scores,

$$\mathbf{s}_t^{\theta,\text{guided}}(\mathbf{x}_t, \tilde{c}) = \omega\, \mathbf{s}_t^{\theta}(\mathbf{x}_t, \tilde{c}) + (1-\omega)\left[(1-\alpha)\mathbf{s}_t^{\theta}(\mathbf{x}_t, \emptyset) + \alpha\, \mathbf{s}_t^{\theta,\text{bad}}(\mathbf{x}_t, \tilde{c})\right], \qquad (27)$$

required in the generative SDE in Eq. (20).

Note that when we apply self-conditioning (Sec. 3.2) during sampling we generally feed the same clean data prediction

$$\hat{\mathbf{x}}(\mathbf{x}_t) = \mathbf{x}_t + (1-t)\mathbf{v}_t^{\theta}(\mathbf{x}_t, \tilde{c}) \qquad (28)$$

as conditioning to all different models of the guidance equations (Eqs. (26) and (27)). Self-conditioning is optional, though, since we train with the self-conditioning input in only 50% of the training iterations. Proteína can be used both with and without self-conditioning.

## I.3 GUIDANCE WITH HIERARCHICAL FOLD CLASS LABELS

In order to be able to apply classifier-free guidance during inference, one typically learns a model that can be used both as a conditional and an unconditional one, by randomly dropping out the conditioning labels during training and feeding a corresponding $\emptyset$-embedding that indicates unconditional generation. As discussed in detail in Sec. 3.2, we drop out our hierarchical fold class labels in a hierarchical manner, thereby enabling guidance with respect to all different levels of the hierarchy.

Here, we summarize the corresponding guidance equations. Note that we do not explicity indicate the time step $t$ conditioning as well as the self-guidance conditioning, to keep the equations short and readable.

**T-level guidance.** If we guide with respect to the finest fold class T, we use

$$\begin{aligned}
\mathbf{v}_t^{\theta,\text{guided}}(\mathbf{x}_t, \{C_\mathbf{x}, A_\mathbf{x}, T_\mathbf{x}\}_{\text{CAT}}) &= \omega\, \mathbf{v}_t^{\theta}(\mathbf{x}_t, \{C_\mathbf{x}, A_\mathbf{x}, T_\mathbf{x}\}_{\text{CAT}}) \\
&+ (1-\omega)\left[(1-\alpha)\mathbf{v}_t^{\theta}(\mathbf{x}_t, \{\emptyset, \emptyset, \emptyset\}_{\text{CAT}}) + \alpha\, \mathbf{v}_t^{\theta,\text{bad}}(\mathbf{x}_t, \{C_\mathbf{x}, A_\mathbf{x}, T_\mathbf{x}\}_{\text{CAT}})\right],
\end{aligned} \qquad (29)$$

and correspondingly for the score $\mathbf{s}_t^{\theta,\text{guided}}(\mathbf{x}_t, \{C_\mathbf{x}, A_\mathbf{x}, T_\mathbf{x}\}_{\text{CAT}})$. As mentioned above, $\omega$ is the guidance strength. For autoguidance, we have $\alpha = 1$, and for CFG we have $\alpha = 0$. Note that we also feed the "coarser" C- and A-level labels that are the parents of the T-level label in the hierarchy.

**A-level guidance.** If we guide with respect to the fold class A, we use

$$\begin{aligned}
\mathbf{v}_t^{\theta,\text{guided}}(\mathbf{x}_t, \{C_\mathbf{x}, A_\mathbf{x}, \emptyset\}_{\text{CAT}}) &= \omega\, \mathbf{v}_t^{\theta}(\mathbf{x}_t, \{C_\mathbf{x}, A_\mathbf{x}, \emptyset\}_{\text{CAT}}) \\
&+ (1-\omega)\left[(1-\alpha)\mathbf{v}_t^{\theta}(\mathbf{x}_t, \{\emptyset, \emptyset, \emptyset\}_{\text{CAT}}) + \alpha\, \mathbf{v}_t^{\theta,\text{bad}}(\mathbf{x}_t, \{C_\mathbf{x}, A_\mathbf{x}, \emptyset\}_{\text{CAT}})\right],
\end{aligned} \qquad (30)$$

and correspondingly for the score $\mathbf{s}_t^{\theta,\text{guided}}(\mathbf{x}_t, \{C_\mathbf{x}, A_\mathbf{x}, \emptyset\}_{\text{CAT}})$.

**C-level guidance.** If we guide with respect to the fold class C, we use

$$\begin{aligned}
\mathbf{v}_t^{\theta,\text{guided}}(\mathbf{x}_t, \{C_\mathbf{x}, \emptyset, \emptyset\}_{\text{CAT}}) &= \omega\, \mathbf{v}_t^{\theta}(\mathbf{x}_t, \{C_\mathbf{x}, \emptyset, \emptyset\}_{\text{CAT}}) \\
&+ (1-\omega)\left[(1-\alpha)\mathbf{v}_t^{\theta}(\mathbf{x}_t, \{\emptyset, \emptyset, \emptyset\}_{\text{CAT}}) + \alpha\, \mathbf{v}_t^{\theta,\text{bad}}(\mathbf{x}_t, \{C_\mathbf{x}, \emptyset, \emptyset\}_{\text{CAT}})\right],
\end{aligned} \qquad (31)$$

and correspondingly for the score $\mathbf{s}_t^{\theta,\text{guided}}(\mathbf{x}_t, \{C_\mathbf{x}, \emptyset, \emptyset\}_{\text{CAT}})$.

**No fold class guidance.** If we do not guide with respect to a fold class, but we still want to apply autoguidance in its unconditional setting, we have $\alpha = 1$ and

$$\mathbf{v}_t^{\theta,\text{autoguided}}(\mathbf{x}_t, \{\emptyset, \emptyset, \emptyset\}_{\text{CAT}}) = \omega\, \mathbf{v}_t^{\theta}(\mathbf{x}_t, \{\emptyset, \emptyset, \emptyset\}_{\text{CAT}}) + (1-\omega)\mathbf{v}_t^{\theta,\text{bad}}(\mathbf{x}_t, \{\emptyset, \emptyset, \emptyset\}_{\text{CAT}}), \qquad (32)$$

and correspondingly for the score $\mathbf{s}_t^{\theta,\text{autoguided}}(\mathbf{x}_t, \{\emptyset, \emptyset, \emptyset\}_{\text{CAT}})$.

In practice, for the "bad" models required for autoguidance, we use early training checkpoints of our main models. We do not train separate, smaller dedicated models just for the purpose of autoguidance, but this would be an interesting future endeavor.

### I.4 STEP SIZE AND STOCHASTICITY SCHEDULES

As discussed in the previous section, sampling Proteína involves simulating the SDE

$$d\mathbf{x}_t = \mathbf{v}_t^\theta(\mathbf{x}_t, \tilde{c})dt + g(t)\mathbf{s}_t^\theta(\mathbf{x}_t, \tilde{c})dt + \sqrt{2g(t)\gamma}\,d\mathcal{W}_t \qquad (33)$$

from $t = 0$ to $t = 1$, where the vector field and score can also be subject to guidance. This is exactly Eq. (3) in the main text, repeated here for convenience. In practice, we simulate the SDE using the Euler-Maruyama method detailed in Algorithm 1. For all our experiments, we use $N = 400$ discretization steps and $g(t) = 1/(t + 0.01)$ for $t \in [0, 0.99]$ and $g(t) = 0$ for $t \in (0.99, 1)$ (we empirically observed that numerically simulating the SDE may lead to unstable simulation for $t$ close to 1. We avoid this by switching to the ODE, setting $g(t) = 0$, for the last few steps). We explore multiple values for the noise scaling parameter $\gamma$, which leads to different trade-offs between metrics (see Tabs. 1 and 3). We discretize the unit interval using logarithmically spaced points. More precisely, in PyTorch code (Paszke et al., 2019), we get $[t_0, t_1, ..., t_N]$ by the following three steps

```
t = 1.0 - torch.logspace(-2, 0, nsteps + 1).flip(0)
t = t - torch.min(t)
t = t / torch.max(t),
```

where the last two operations ensure that $t_0 = 0$ and $t_1 = 1$.

---

**Algorithm 1** Euler-Maruyama numerical simulation scheme

**Input:** Number of steps $N$
**Input:** Discretization of the unit interval $0 = t_0 < t_1 < t_2 < ... < t_N = 1$
**Input:** Stochasticity schedule $g(t)$
**Input:** Noise scaling parameter $\gamma$
**Input:** Conditioning variables $\tilde{c}$

$\mathbf{x}_0 \sim \mathcal{N}(0, I)$
**for** $n = 1$ **to** $N - 1$ **do**
 $\epsilon_n \sim \mathcal{N}(0, I)$
 $\delta_n = t_n - t_{n-1}$
 $\mathbf{x}_{t_n} = \mathbf{x}_{t_{n-1}} + \left[\mathbf{v}_{t_{n-1}}^\theta(\mathbf{x}_{t_{n-1}}, \tilde{c}) + g(t_{n-1})\,\mathbf{s}_{t_{n-1}}^\theta(\mathbf{x}_{t_{n-1}}, \tilde{c})\right]\delta_n + \sqrt{2\delta_n g(t_n)\gamma}\,\epsilon_n$
**end for**
**Output:** $\mathbf{x}_1$

---

## J   ON THE RELATION BETWEEN FLOW MATCHING AND DIFFUSION MODELS

A question that frequently comes up is the relation between flow matching (Lipman et al., 2023; Liu et al., 2023; Albergo & Vanden-Eijnden, 2023) and diffusion models (Sohl-Dickstein et al., 2015; Ho et al., 2020; Song et al., 2021). For Proteína, we opted for a flow matching-based approach, but in protein structure generation, both approaches have been leveraged in the past. Hence, here we discuss the two frameworks.

Crucially, we would first like to point out that we are using flow matching to couple the training data distribution (the protein backbones for training) *with a Gaussian noise distribution*, from which the generation process is initialized when sampling new protein backbones after training. In this case, i.e. when coupling with a Gaussian distribution, flow matching models and diffusion models can in fact be shown to be equivalent up to reparametrizations. This is because diffusion models generally use a Gaussian diffusion process, thereby also defining Gaussian conditional probability paths, similar to the Gaussian conditional probability paths in flow matching with a Gaussian noise distribution.

For instance, when using a Gaussian noise distribution, one can rewrite the velocity prediction objective used in flow matching as a noise prediction objective, which is frequently encountered in diffusion models (Ho et al., 2020). Different noise schedules in diffusion models can be related to different time variable reparametrizations in flow models (Albergo et al., 2023). Most importantly, for Gaussian flow matching, we can derive a relationship between the *score function* $\nabla_{\mathbf{x}_t} \log p_t(\mathbf{x}_t)$ of

the interpolated distributions and the flow's velocity (see Eq. (2) as well as App. I.1). The score is the key quantity in score-based diffusion models (Song et al., 2021). Using this relation, diffusion-like stochastic samplers for flow models can be derived, as well as flow-like deterministic ODE samplers for diffusion models (Ma et al., 2024). In conclusion, we could in theory look at our Proteína flow models equally as score-based diffusion models. With that in mind, from a pure performance perspective flow matching-based approaches and diffusion-based approaches should in principle perform similarly well when coupling with a Gaussian noise distribution. In practice, performance boils down to choosing the best training objective formulation, the best time sampling distribution to give appropriate relative weight to the objective (see Sec. 3.2), etc.—these aspects dictate model performance, independently of whether one approaches the problem from a diffusion model or a flow matching perspective.

In fact, we directly leverage the connections between diffusion and flow models when developing our stochastic samplers (see App. I.1) and guidance schemes. Both classifier-free guidance (Ho & Salimans, 2021) and autoguidance (Karras et al., 2024) were proposed for diffusion models, but due to the relations between score and velocity, we can also apply them to our flow models (to the best of our knowledge, our work is the first to demonstrate classifier-free guidance and autoguidance for flow matching of protein backbone generation). Please see App. I.2 for all technical details regarding guidance in Proteína.

Considering these relations, why did we overall opt for the flow matching formulation and perspective? *(i)* Flow matching can be somewhat simpler to implement and explain, as it is based on simple interpolations between data and noise samples. No stochastic diffusion processes need to be considered. *(ii)* Flow matching offers the flexibility to be directly extended to more complex interpolations, beyond Gaussians and diffusion-like methods. For instance, we may consider optimal transport couplings (Pooladian et al., 2023; Tong et al., 2024) to obtain straighter paths for faster generation or we could explore other, more complex non-Gaussian noise distributions. We plan to further improve Proteína in the future and flow matching offers more flexibility in that regard. At the same time, when using Gaussian noise, all tricks from the diffusion literature still remain applicable. *(iii)* The popular and state-of-the-art large-scale image generation system Stable Diffusion 3 is similarly based on flow matching (Esser et al., 2024). This work demonstrated that flow matching can be scaled to large-scale generative modeling problems.

We would like to point out that the relations between flow-matching and diffusion models have been discussed in various papers. One of the first works pointing out the relation is Albergo & Vanden-Eijnden (2023) and the same authors describe a general framework in Stochastic Interpolants (Albergo et al., 2023), unifying a broad class of flow, diffusion and other models. Some of the key relations and equations can also be found more concisely in Ma et al. (2024). The relations between flow matching and diffusion models have also been highlighted in the Appendix of Kingma & Gao (2023). The first work scaling flow matching to large-scale text-to-image generation is the above mentioned Esser et al. (2024), which also systematically studies objective parametrizations and time sampling distributions, similarly leveraging the relation between flow and diffusion models.

## K   NEW TIME $t$ SAMPLING DISTRIBUTION

A crucial parameter in diffusion and flow matching models is the $t$ sampling distribution $p(t)$, which effectively weighs the objective (Eq. (1) for Proteína). Enhanced sampling of $t$ closer to $t = 1$ encourages the model to focus capacity on synthesizing accurate local details, which are generated at the end of the generative process, while sampling more at smaller $t$ can improve larger-scale features. In image generation it is common to increase sampling at intermediate $t$ (Karras et al., 2022; Esser et al., 2024), but this is not necessarily a good choice for protein structures—even slightly perturbing a structure could lead to unphysical residue arrangements and bond lengths. Hence, as discussed in Sec. 3.2 we designed a new $t$ sampling function focusing more on large $t$,

$$p(t) = 0.02\,\mathcal{U}(0, 1) + 0.98\,\mathcal{B}(1.9, 1.0),$$

where $\mathcal{B}(\cdot, \cdot)$ is the Beta distribution, to encourage accurate local details. We mix in uniform sampling to avoid zero sampling density when $t \to 0$. In Fig. 20, we show our novel distribution, a naive uniform distribution, and the logit-normal distribution that recently achieved state-of-the-art image synthesis in a similar rectified flow objective (Esser et al., 2024). Ablations can be found in App. L.

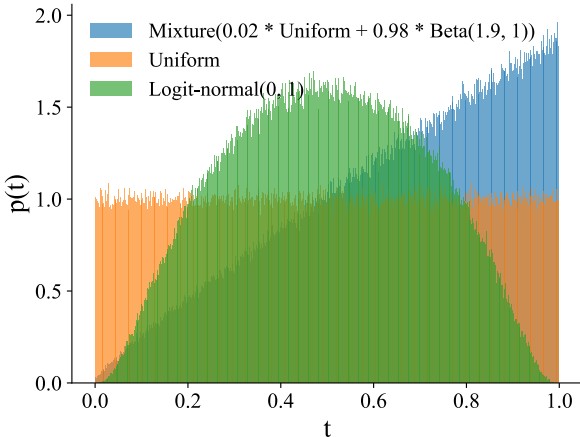

Figure 20: $t$-**sampling distributions.** We show our novel $t$-sampling distribution from Sec. 3.2 that mixes a Beta and a uniform distribution, a naive uniform distribution, and the logit-normal distribution that recently achieved state-of-the-art image synthesis in a similar rectified flow objective (Esser et al., 2024).

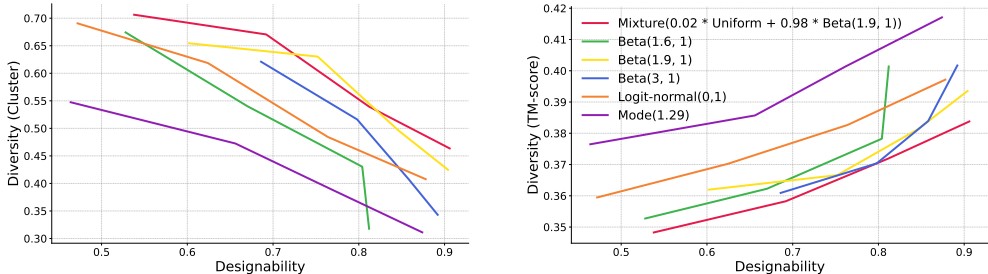

Figure 21: Designability-Diversity trade-offs achieved when training a "small" model using different t-sampling distributions. The curves are obtained by sampling each trained model for multiple noise scaling parameters $\gamma$ between 0.25 and 0.55.

## L ABLATION STUDIES

This section presents multiple ablations we carried out while developing the model. We tested multiple distributions to sample the time $t$ during training (App. L.1), different stochasticity schedules $g(t)$ (App. L.2), and explored various architectural choices (App. L.3). We note that these ablations were done at different stages during model development, and are thus not always directly comparable between each other, nor with the results presented in the main paper, as they were carried out with different (often small) models or sampling schemes. However, these ablations informed our decisions while developing our model and training regime.

### L.1 SAMPLING DISTRIBUTIONS FOR $t$

We consider multiple choices for the $t$-sampling distribution $p(t)$: the `mode(1.29)` distribution from Esser et al. (2024), the `Logit-normal(1, 0)` distribution from Esser et al. (2024), `Beta(3, 1)`, `Beta(1.6, 1)`, `Beta(1.9, 1)`, `Mixture(0.02*Uniform+0.98*Beta(1.9, 1))`, and `Uniform(0, 1)`. For each of these we trained a "small" model (30M parameters, no pair updates) on the $\mathcal{D}_{FS}$ dataset for 150K steps, using 4 GPUs with a batch size of 25 per GPU. Noting that training losses are not directly comparable for different $t$-sampling distributions, we compare the models by studying the designability-diversity trade-off they achieve when sampled under different noise scaling parameters $\gamma$. Results are shown in Fig. 21, where we show diversity metrics (cluster diversity and TM-score diversity, see App. F for details) as a function of designability. The curves are obtained by sweeping the noise scaling parameter $\gamma$ between 0.25 and 0.55. We can observe that the mixture distribution

consistently achieves the best trade-offs for both diversity metrics. Note that the training run using the uniform distribution displayed somewhat unstable behavior, producing `nan` values during training, so we did not sample it.

We performed this ablation early during model development, using the stochasticity schedule $g(t) = (1-t)/(t+0.01)$ and a uniform discretization of the unit interval (with 400 steps). We emphasize that these results are not comparable with the ones in the main papar and other sections in the Appendix, as they were obtained using a significantly smaller model, trained for less steps using less compute, and sampled using a different numerical simulation scheme.

### L.2 STOCHASTICITY SCHEDULES $g(t)$

In addition to the schedule $g(t) = 1/(t + 0.01)$ presented in App. I.4, used for all results we report in the main paper, we also tested the schedules $g_{1-t}(t) = (1 - t)/(t + 0.01)$ and $g_{\tan}(t) = \frac{\pi}{2} \tan\left((1-t)\frac{\pi}{2}\right)$.[4] A comparison of these three schedules is shown in Fig. 22, where it can be observed that $g_{\tan}$ and $g_{1-t}$ inject significantly less noise for times $t \approx 1$. We sampled and evaluated our $\mathcal{M}_{\mathrm{FS}}$ model for these three schedules. Results for deisgnability, diversity and novelty (w.r.t. PDB) are shown in Fig. 23.

It can be observed that the stochasticity schedule used has a strong effect in the model's final performance, with $g(t)$ leading to better results than $g_{\tan}(t)$ and $g_{1-t}(t)$. Note that, in principle, for $\gamma = 1$ all stochasticity schedules yield the same marginal distributions during the sampling process. In practice, however, the SDE is simulated numerically, and we use a noise scaling parameter $\gamma < 1$ (common in diffusion and flow-based generative methods for protein backbone design). These two factors have a nontrivial interaction with the stochasticity schedule, explaining the differences in results for the different $g(t)$ considered.

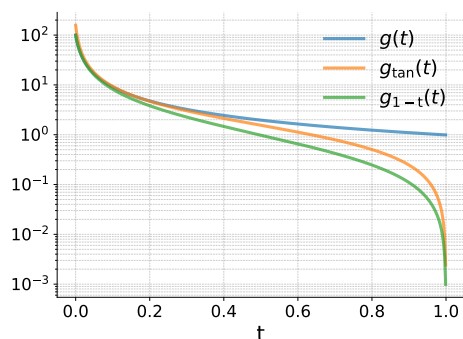

Figure 22: Different stochasticity schedules tested as a function of $t$.

### L.3 QK LAYER
NORM, REGISTERS AND ROPE EMBEDDINGS

We also ablated several choices for the architecture, with an interesting one being the addition or removal of pair updates with triangular multiplicative layers (model $\mathcal{M}_{\mathrm{FS}}$ against model $\mathcal{M}_{\mathrm{FS}}^{\text{no-tri}}$). These two models were compared in the main text in Tab. 1. While the use of pair updates with triangular multiplicative updates leads to better performance, it also has a negative impact on the model's scalability. In Tab. 1 we observed that our $\mathcal{M}_{\mathrm{FS}}^{\text{no-tri}}$ model is still competitive while being significantly more computationally efficient, which enabled us to scale to protein backbones of up to 800 residues, as discussed in the main text.

Other architectural choices we ablated involved the use of QK layer norm, registers, and rotary positional embeddings (RoPE) (Su et al., 2024) for the attention in the network's trunk. These changes only affect the architecture, so training losses are directly comparable. Training small models (see App. L.1) we observed that the use of registers and QK layer norm led to slightly improved training losses, so we included them in our final architecture. The use of RoPE embeddings, on the other hand, led to a slight increase in the training loss, so we did not include it in our final architecture.

---

[4]For numerical stability we compute $g_{\tan}(t) = \frac{\pi}{2} \frac{C(t)}{S(t)+0.01}$ where $C(t) = \cos\left((1 - t)\frac{\pi}{2}\right)$ and $S(t) = \sin\left((1 - t)\frac{\pi}{2}\right)$

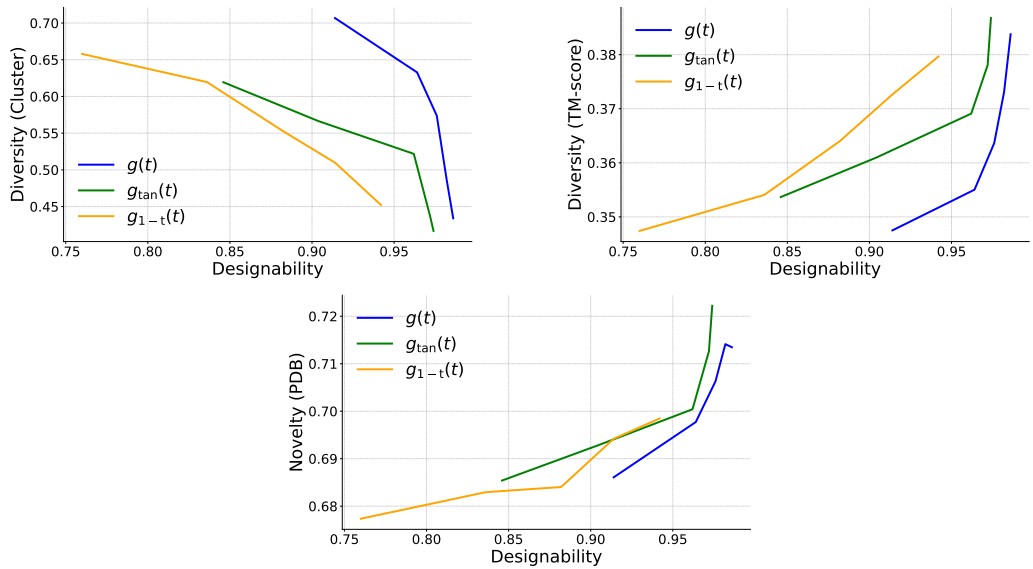

Figure 23: Results of sampling $\mathcal{M}_{\text{FS}}$ under three different schedules, $g(t)$, $g_{\tan}(t)$ and $g_{1-t}(t)$. Curves are obtained by sweeping the noise scaling parameter $\gamma$ between 0.3 and 0.5.

## M  DATA PROCESSING

### M.1  PDB PROCESSING, FILTERING AND CLUSTERING

For PDB datasets, we use metadata from the PDB directly to filter for single chains with lengths 50-256, resolution below 5Å and structures that do not contain non-standard residues. We also include chains from oligomeric proteins. We include structure-based filters, namely a max. coil proportion of 0.5 and a max. radius of gyration of 3.0nm. Together, this leads to 114,076 protein chains.

For LoRA-based fine-tuning, we prepare a subset of the dataset above with only designable structures. For this, we feed all 114,076 chains from the first dataset through the designability pipeline (Protein-MPNN, ESMFold), only keeping chains that have a scRMSD below 2Å. With this, we reduce the dataset size to 90,423 proteins, indicating that 79.26% of the samples from the original PDB dataset described above were designable.

### M.2  ALPHAFOLD DATABASE PROCESSING, FILTERING AND CLUSTERING

Processing the full AFDB-Uniprot dataset takes more than 20TB in disk space and includes around 214M individual file objects, both of which make it hard to work with this data. To remedy that, we leverage FoldComp (Kim et al., 2023) as a tool to enable efficient storage and fast access. FoldComp leverages NeRF (Natural Extension Reference Frame) to encode structure information in 13 bytes per residue and allow a fast compression/decompression scheme with minimal reconstruction loss.

Combined with FoldComp we use the MMseqs2 database format (Steinegger & Söding, 2017) to filter the AFDB-UniProt database into custom databases based on our filters and allow fast random-access and on-the-fly decompression at training time. These filters include sequence length, pLDDT values (mean and variance over the sequence), secondary structure content and radius of gyration.

In addition, we want to avoid random sampling of these databases since, after filtering, these datasets tend to be biased towards certain fold families. We therefore either utilise pre-computed clustering of the AFDB such as the AFDB-Foldseek clusters (Barrio-Hernandez et al., 2023), or cluster the databases according to sequence similarity via MMseqs2. We then use the mapping from cluster representative to cluster members to define a PyTorch Sampler that iterates over all clusters during one epoch, picking a random cluster member for each cluster.

Via this data processing pipeline, we prepare the different datasets that are described in the main section of the paper:

**1. High-quality filtered AFDB subset, size $\approx$21M, $\mathcal{D}_{21M}$:** We filtered all $\approx$214M AFDB structures for a max. length of 256 residues, min. average pLDDT of 85, max. pLDDT standard deviation of 15, max. coil percentage of 50%, and max. radius of gyration of 3nm. After additional subsampling this led to 20,874,485 structures. We further clustered the data with MMseqs2 (Steinegger & Söding, 2017) using a 50% sequence similarity threshold. During training, we sample clusters uniformly and draw random structures within.

**2. Foldseek AFDB clusters, only representatives, size $\approx$0.6M, $\mathcal{D}_{FS}$:** This dataset corresponds to the data that was also used by Genie2 (Lin et al., 2024), based on sequential filtering and clustering of the AFDB with the sequence-based MMseqs2 and the structure-based Foldseek (van Kempen et al., 2024; Barrio-Hernandez et al., 2023). This data uses *cluster representatives only*, i.e. only one structure per cluster. Like Genie2, we use a length cutoff at 256 residues and a pLDDT filter of 80, as well as a minimum length cutoff of 32 residues. We found that with this processing we obtained 588,318 instead of 588,571 AFDB structures compared to Genie2. We attribute this difference to two reasons: first, some pLDDT values that are listed as 80 in the AFDB are rounded to 80 and do not equal 80 when computed as an average directly from the per-residue pLDDT data, leading to samples that are in the dataset used by Genie2 but not in our dataset. Second, the AFDB clustering (Barrio-Hernandez et al., 2023) was done on version 3 of the AFDB, whereas in version 4 some of the structures were re-predicted with better confidence values. Several structures that were excluded in the data used by Genie2 due to having low pLDDT values at the time of version 3 had significantly improved pLDDT values in version 4. This leads to samples that are in our version of the dataset but not in Genie2's.

During long-length fine-tuning, we extend $\mathcal{D}_{FS}$ by progressively increasing the maximum length considered, up to a maximum chain length of 768 residues.

## M.3 CATH LABEL ANNOTATIONS FOR PDB AND AFDB

In order to make our model more controllable via methods like Classifier-Free Guidance (CFG), we leverage hierarchical CATH fold class labels (Dawson et al., 2016) for both experimental and predicted structures. These labels come in a hierarchy with multiple different levels (also see Fig. 3 in the main text):

**Class (C):** describes the overall secondary structure content of the protein domain, similar to SCOPe class (Lo Conte et al., 2000).

**Architecture (A):** describes how secondary structure elements are arranged in space (for example sandwiches, rolls and barrels).

**Topology (T):** describes how secondary structure elements are arranged and connected to each other.

**Homologous superfamily (H):** describes how likely these domains are evolutionarily related, often supported by sequence information.

Since we are mostly interested in structural features for guidance, we focus on the CAT levels of the hierarchy, discarding the H level. In addition, we focus mostly on the three major C classes ("mostly alpha", "mostly beta" and "mixed alpha/beta") and ignore the smaller special classes ("few secondary structure", "special").

**PDB CATH labels:** To obtain CATH labels for the individual PDB chains that we use as data points, we leverage the SIFTS resource (Structure Integration with Function, Taxonomy and Sequences resource) (Velankar et al., 2012), which is regularly updated and provides residue-level mappings between UniProt, PDB and other data resources such as CATH (Dana et al., 2019). For each of our samples, we map from the PDB ID and chain ID to the corresponding UniProt ID via the `pdb_chain_uniprot.tsv.gz` mapping, and from there to the corresponding CATH IDs and CATH codes via the `pdb_chain_cath_uniprot.tsv.gz` mapping. Some chains have more than one domain and then also more than one CATH code. For these, we use all CATH codes available (we randomly sample the conditioning CATH code fed to the model during training in those cases). We then truncate the labels to remove the H level and end up with CAT labels only.

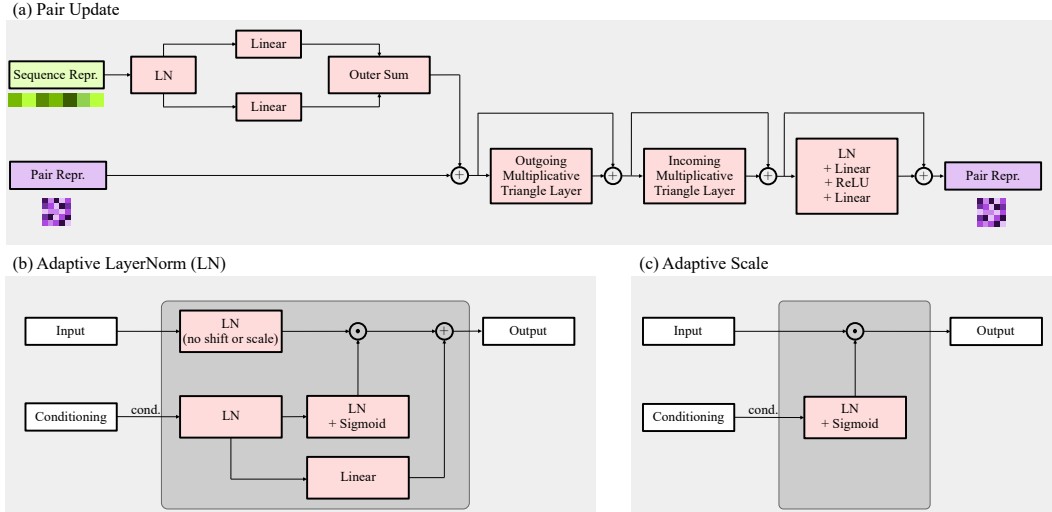

Figure 24: **Additional modules of Proteína's transformer architecture.** *(a)* Pair Update. *(b)* Our adaptive LayerNorm *(c)* Adaptive Scale.

**AFDB CATH labels**: To obtain CATH labels for the individual AFDB chains that we use as data points, we leverage the TED resource (The Encyclopedia of Domains) (Lau et al., 2024b) to map from the AFDB UniProt identifier to the corresponding CAT/CATH codes. Again we use all available CATH codes and remove the H-level information if it is present.

## N ADDITIONAL NEURAL NETWORK ARCHITECTURE DETAILS

Visualizations of the additional *Pair Update*, *Adaptive LayerNorm (LN)*, and *Adaptive Scale* modules are shown in Fig. 24.

When creating the pair representation (see Fig. 5 (c)), the pair and sequence distances created from the inputs $\mathbf{x}_t$, $\hat{\mathbf{x}}(\mathbf{x}_t)$ and the sequence indices are discretized and encoded into one-hot encodings. Specifically, for the pair distances from $\mathbf{x}_t$ we use 64 bins of equal size between 1Å and 30Å with the first bin being <1Å and the last one being >30Å. For the pair distances from $\hat{\mathbf{x}}(\mathbf{x}_t)$ we use 128 bins of equal size between 1Å and 30Å with the first bin being <1Å and the last one being >30Å. For the sequence separation distances we use 127 bins for sequence separations $[<-63, -63, -62, -61, ..., 61, 62, 63, >63]$. As shown in Fig. 5 this pair representation can be (optionally) updated throughout the network using pair update layers. These feed the sequence representation through linear layers to update the pair representation, which is additionally updated using triangular multiplicative updates (Jumper et al., 2021), as shown in Fig. 24. While powerful, these triangular layers are computationally expensive; hence, we limit their use in our models. In fact, as shown in Tab. 16, our $\mathcal{M}_{FS}^{\text{no-tri}}$ model completely avoids the use of these layers (leading to a much more scalable model), while $\mathcal{M}_{FS}$ and $\mathcal{M}_{21M}$ use 5 and 4 triangular multiplicative update layers, respectively. In this work we did not explore the use of triangular attention layers (Jumper et al., 2021), as these are even more memory and computationally expensive, limiting the models' scalability.

We generally use 10 register tokens in all models when constructing the sequence representation. Sequence conditioning and pair representation are zero-padded accordingly.

The MLP used when creating the sequence conditioning (see Fig. 5 (b)) corresponds to a Linear–SwiGLU–Linear–SwiGLU–Linear architecture (Shazeer, 2020).

Specific architecture hyperparameters like the number of layers, attention heads and embedding sizes used during training of different Proteína models can be found in App. O.

## O  EXPERIMENT DETAILS AND HYPERPARAMETERS

This section provides details about our model architectures as well as training and sampling configurations for our experiments.

### O.1  TRAINED PROTEÍNA MODELS

Tab. 16 presents the hyperparameters used to define the three architectures considered in this paper, giving details about number of layers, dimensions of each feature, and number of trainable parameters, among others. It also offers details on the training of our models, such as the number of GPUs used, the number of training steps, and the batch size per GPU. All our models were trained using Adam (Kingma, 2014) with $\beta_1 = 0.9$ and $\beta_2 = 0.999$. We use random rotations to augment training samples.

Table 16: Hyperparameters for Proteína model training.

| Hyperparameter | Pre-training | | | Fine-tuning | |
|---|---|---|---|---|---|
| | $\mathcal{M}_{\text{FS}}$ | $\mathcal{M}_{\text{FS}}^{\text{no-tri}}$ | $\mathcal{M}_{\text{21M}}$ | $\mathcal{M}_{\text{LoRA}}$ | $\mathcal{M}_{\text{long}}$ |
| **Proteína Architecture** | | | | | |
| initialization | random | random | random | $\mathcal{M}_{\text{FS}}$ | $\mathcal{M}_{\text{FS}}^{\text{no-tri}}$ |
| sequence repr dim | 768 | 768 | 1024 | 768 | 768 |
| # registers | 10 | 10 | 10 | 10 | 10 |
| sequence cond dim | 512 | 512 | 512 | 512 | 512 |
| $t$ sinusoidal enc dim | 256 | 256 | 256 | 256 | 256 |
| idx. sinusoidal enc dim | 128 | 128 | 128 | 128 | 128 |
| fold emb dim | 256 | 256 | 256 | 256 | 256 |
| pair repr dim | 512 | 512 | 512 | 512 | 512 |
| seq separation dim | 128 | 128 | 128 | 128 | 128 |
| pair distances dim ($\mathbf{x}_t$) | 64 | 64 | 64 | 64 | 64 |
| pair distances dim ($\hat{\mathbf{x}}(\mathbf{x}_t)$) | 128 | 128 | 128 | 128 | 128 |
| pair distances min (Å) | 1 | 1 | 1 | 1 | 1 |
| pair distances max (Å) | 30 | 30 | 30 | 30 | 30 |
| # attention heads | 12 | 12 | 16 | 12 | 12 |
| # tranformer layers | 15 | 15 | 18 | 15 | 15 |
| # triangle layers | 5 | — | 4 | 5 | — |
| # trainable parameters | 200M | 200M | 400M | 7M | 200M |
| **Proteína Training** | | | | | |
| # steps | 200K | 360K | 180K | 11K | 220K/80K |
| batch size per GPU | 4 | 10 | 4 | 6 | 2/1 |
| # GPUs | 128 | 96 | 128 | 32 | 128 |
| # grad. acc. steps | 1 | 1 | 1 | 2 | 1/2 |

### O.2  UNCONDITIONAL GENERATION EXPERIMENTS

This section presents precise details for all results for unconditional generation shown in Tabs. 1 and 3 (not including LoRA fine-tuning, covered in App. O.3). All experiments follow the sampling algorithm described in App. I.4.

Tab. 1 shows results obtained for our $\mathcal{M}_{\text{FS}}$ and $\mathcal{M}_{\text{FS}}^{\text{no-tri}}$ models under multiple noise scales.[5] We sampled the $\mathcal{M}_{\text{FS}}$ model without self-conditioning, since we observed that this yielded better trade-offs between designability, diversity, and novelty in the unconditional setting. On the other hand, we used self-conditioning with the $\mathcal{M}_{\text{FS}}^{\text{no-tri}}$ model, since we observed it often led to slightly improved performance. The noise scales $\gamma \in \{0.35, 0.45, 0.5\}$ shown for $\mathcal{M}_{\text{FS}}$ were chosen to show different points in the Pareto front between different metrics, while still retaining high designability values. On the other hand, $\mathcal{M}_{\text{FS}}^{\text{no-tri}}$ was sampled for a single noise scale ($\gamma = 0.45$) to show that even without a pair track (i.e., no updates to the pair representation, yielding significantly improved scalability) our model still performs competitively.

Tab. 1 also shows results for the $\mathcal{M}_{\text{21M}}$ model for two different noise scales $\gamma \in \{0.3, 0.6\}$. These two runs have different purposes. The one with lower noise scale aims to show that we can achieve

---

[5]While the $\mathcal{M}_{\text{FS}}^{\text{no-tri}}$ model was trained for 360k steps, we observed better designability-diversity trade-offs for earlier training checkpoints. Therefore, for that model, we show results after 80k steps.

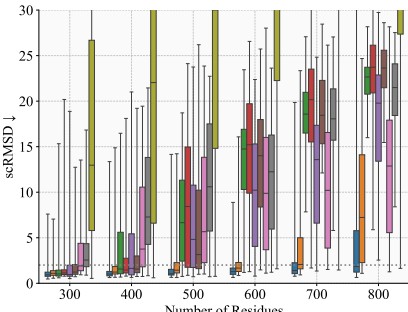 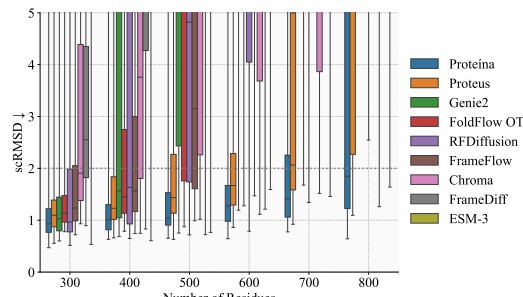

Figure 25: scRMSD values for long protein generation, with zoomed-out view (left, y-axis: 0 Å to 30 Å) and zoomed-in view (right, y-axis: 0 Å to 5 Å).

extremely high designability values by training on a large dataset filtered for high quality structures (to achieve this we use self-conditioning for this run), while the other one attempts to show better trade-offs between different metrics (achieved without self-conditioning).

Finally, Tab. 3 shows results for ODE sampling for the $\mathcal{M}_{FS}$ and $\mathcal{M}_{21M}$ models (both runs with self-conditioning, which yields better designability values). These runs can be observed to produce significantly better values for our new metrics, FPSD, fS and fJSD. This is expected since scaling the noise term in the SDE is known to modify the distribution being sampled.

## O.3  LoRA Fine-tuning on PDB

To enhance our model's ability to generate both designable and realistic samples, we curate a high-quality, designable PDB dataset as outlined in App. M.1. We then fine-tune our best unconditional model, $\mathcal{M}_{FS}$, on this processed dataset. To prevent overfitting and enable efficient fine-tuning, we apply LoRA (Hu et al., 2022) with a rank of 16 and a scaling factor of 32, introducing trainable low-rank decomposition matrices into all embedding and linear layers of the model. This reduces the number of trainable parameters to 7M, significantly lower than the original 200M parameters. The complete training configuration is detailed in Tab. 16. For inference, we observe that enabling self-conditioning consistently improves designability, so we adopt it for this model.

## O.4  Conditional Generation Experiments

For conditional generation, we follow the same schedule as unconditional sampling, with self-conditioning enabled, as we find it improves designability and re-classification probabilities. In Tab. 2, we explore the effect of classifier-free guidance (CFG) by sweeping the guidance weight among 1.0, 1.5, and 2.0 for the model $\mathcal{M}_{FS}^{cond}$ with a noise scale of $\gamma = 0.4$.

During sampling, we account for the compatibility of the (protein chain length, CATH code) combinations to avoid generating unrealistic lengths for certain classes. We divide the lengths into buckets ranging from 50 to 1,000, with a bucket size of 25. An empirical label distribution is then constructed for each length bucket based on the datasets $\mathcal{D}_{FS}$ and $\mathcal{D}_{21M}$. For each length in conditional sampling, we randomly select a CATH code from the empirical distribution corresponding to that length.

**Class-specific Guidance.**   To showcase the utility of guidance at different hierarchical levels, we also perform class-specific guidance where we guide the model only via class labels ("mainly alpha", "mainly beta", "mixed alpha/beta") to control secondary structure content in samples while still maintaining high designability and diversity. We sample the model conditionally with a guidance weight of 1, and a noise scale of $\gamma = 0.4$ for the conditional classes and $\gamma = 0.45$ for the unconditional case. With this configuration, we generate samples of length 50, 100, 150, 200, and 250 with 100 examples each, totaling 500 samples, which are used to report designability, diversity, novelty, and secondary structure content in Tab. 4.

### O.5 Long Length Generation Experiments

For the long length generation results in Fig. 8 we first take the model $\mathcal{M}_{\text{FS}}^{\text{no-tri}}$ after 360K steps and fine-tune it for long-length generation; for this, we train it for 220K steps on the AFDB Cluster representatives filtered to minimum average pLLDT 80, minimum length 256 and maximum length 512. We then train it for 80K more steps on the same dataset, but with the maximum length increased to 768. We sample this final model $\mathcal{M}_{\text{long}}$ with a noise scale of $\gamma = 0.35$ and 400 steps to generate samples of lengths 300, 400, 500, 600, 700 and 800 with 100 examples per length. These samples are then subject to the previously described metric pipeline calculating designability and diversity. Similarly, for Fig. 8, we sample these lengths from each baseline in accordance with App. P. Also see Fig. 25 for scRMSD plots of Proteína and the baselines for the long protein generation experiment.

In addition, we combine these long length generation capabilities of our model with class-specific guidance (i.e. conditional sampling of the model while providing labels at the C level of the CATH hierarchy) to obtain large proteins with controlled secondary structure content.

### O.6 Autoguidance Experiments

In Fig. 7, we show both conditional and unconditional sampling of our model, $\mathcal{M}_{\text{21M}}$, using the full distribution mode (ODE). The checkpoint at 10K training steps serves as a "bad" guidance checkpoint, corresponding to the "reduced training time" degradation discussed in the original paper (Karras et al., 2024). For both conditional and unconditional sampling, we apply self-conditioning while keeping all other inference configurations consistent with those described earlier.

## P  Baselines

In this section, we briefly list the models that we sampled for benchmarking and the sampling configurations we used.

**Genie2**: We used the code from the Genie2 public repository. We loaded the base checkpoint that was trained for 40 epochs. The noise scale was set to 1 for full temperature sampling and 0.6 for low temperature sampling. The sampling was run in the provided docker image.

**RFDiffusion**: We used the code from the RFDiffusion public repository. The sampling was run in the provided docker image. Default configurations of the repository were used for sampling.

**ESM3**: We followed the instruction in the ESM3 public repository to install and load the publicly available weights through the HugginFace API. When sampling structures, we set the temperature to 0.7, and number of steps to be $L * \frac{3}{2}$ where $L$ is the length of the protein sequence. It is noteworthy that ESM3 performs relatively poorly on metrics evaluating unconditional generation. This may be expected as ESM3 is trained on many metagenomic sequences which are less designable.

**FoldFlow**: We used the code from the FoldFlow public repository. When sampling the FoldFlow-base model, we set both the `ot_plan` and `stochastic_path` in the `flow_matcher` configuration to `False`. When sampling the FoldFlow-OT model, we set the `ot_plan` to `True`. Lastly, when sampling the FoldFlow-SFM model, we set both the `ot_plan` and the `stochastic_path` to `True`, with the `noise_scale` set to 0.1. For all three models, we set the configuration `flow_matcher.so3.inference_scaling` to 5 as we empirically found that such setting yields a performance closest to the results reported in the FoldFlow paper (Bose et al., 2024).

**FrameFlow**: We installed FrameFlow from its public repository. The model weights are downloaded from Zenodo. Default settings are used for unconditional sampling.

**Chroma**: We used the code from the Chroma public repository. Model weights were downloaded through the API following the instructions. Default settings were used for unconditional sampling. For conditional sampling using CAT labels, we used the default `ProClassConditioner` provided in the repository to guide the generation.

**FrameDiff**: We used the code from the FrameDiff public repository, using the public weights from the paper located in `./weights/paper_weights.pth`. The default configuration of the repository was used for sampling. The sampling was run in the provided conda environment.

**Proteus**: We used the code from the Proteus public repository, using the public weights from the paper located in `./weights/paper_weights.pt`. The default configuration of the repository was used for sampling. The sampling was run in the provided conda environment.

