# OpenReview forum: "Proteina: Scaling Flow-based Protein Structure Generative Models"
_ICLR.cc/2025/Conference — ICLR 2025 Oral_

### Official Review · Reviewer_v57v · 2024-11-03

**Soundness:** 4
**Presentation:** 4
**Contribution:** 4
**Rating:** 8
**Confidence:** 4

**Summary:**

This paper presents Proteina, a new large-scale flow-based protein backbone generator that utilizes hierarchical fold class labels for conditioning and relies on a tailored scalable transformer architecture. The model is trained on up to 21 million protein structures, a significant increase compared to previous work. Proteina achieves state-of-the-art performance on de novo protein backbone design and produces diverse and designable proteins at unprecedented length, up to 800 residues. The hierarchical conditioning offers novel control, enabling high-level secondary-structure guidance as well as low-level fold-specific generation. In addition, the paper also offers numerous practical insights that can serve as valuable references, such as LoRA fine-tuning, autoguidance, and new evaluation metrics.

**Strengths:**

- The paper demonstrates the current state-of-the-art capabilities in protein backbone design.
- The paper addresses the longstanding issue of insufficient $\beta$-sheets in this field by introducing the CAT lable.
- The paper introduces new evaluation metrics that offer a fresh perspective for examining the model.
- The paper provides valuable insights by exploring numerous practical techniques, e.g., LoRA fine-tuning and autoguidance, in the field of protein design.

**Weaknesses:**

- Although the article provides detailed implementation details, Proteina, as a complex system engineering project, requires open-source code and data as essential resources for reproducibility.
- The paper does not provide a discussion on the scaling trends of the data and model, meaning we cannot determine whether the 21 million protein structure data points and 400 million model parameters are necessary conditions for achieving the current performance of Proteina.

**Questions:**

- As a method for backbone design, is the input sequence representation in the model composed entirely of a sequence made up of the same special symbol?
- Will the newly proposed metrics, which are based on the fold classifier, create an unfair competitive advantage for Proteina, as it uses CAT as an input? Could a more general classifier be used to improve the design of these metrics?
- For a single protein design, are all amino acids input using the same CAT label (e.g., amino acids A and B both use the label X), or is the CAT label adjusted specifically for each amino acid (e.g., amino acid A uses Xa and amino acid B uses Xb)?

---

> ### Author Response · Authors · 2024-11-22
> **Response by Authors. Part 1**
>
> *We would like to thank the reviewer for their very positive review. We are grateful for highlighting our state-of-the-art performance, our new evaluation metrics, our improved $\beta$-sheet generation thanks to the fold class conditioning, as well as our other contributions such as LoRA fine-tuning and autoguidance. Below, we address the raised questions:*
>
>
> >1. Code and data availability for reproducibility.
>
> Proteina will be made available publicly after publication. Also note that all data sources are publicly available under permissive licenses, and the data preparation protocol is described in detail in Appendix B. Moreover, all training, sampling and hyperparameter details are covered in Appendices F and K.
>
>
> >2. Training on 21 million protein structures, model size, and scaling trends.
>
> Training and fully evaluating new models on the large D_21M dataset is beyond the scope of the rebuttal. However, we would like to point out that our motivation to train this M_21M model is to show a proof of concept that one can curate very large-scale high-quality training sets from the AFDB that result in extremely high designability, despite the fully synthetic nature of the AFDB database, while leveraging scalable architectures like ours (previous training sets are more than an order of magnitude smaller, see Sec. 3.1). Future work could build on this observation and curate different large-scale training sets from the AFDB and other sources, using different filters, tailored to different needs.
>
> Furthermore, in the updated version we performed a scaling study with respect to the flow matching objective and model size, similar to recent seminal work in image generation [1]. See Figures 19 in Appendix M.1. We see that scaling the model size systematically reduces the loss, implying a strong scaling behavior of our architecture.
>
> Moreover, we added the number of parameters as part of Tables 9, 10 and 11 in Appendix M.2. We list the number of parameters for all our models as well as the most relevant baselines. Our Proteina models may seem large and expensive by parameter count, but what matters in practice is not an absolute parameter count, but a model’s memory consumption and sampling speed, along with performance (which is reported in the main paper already, in Table 1). Therefore, in Figure 20 and Tables 9, 10 and 11, we also show the maximum batch size per GPU that our models and the most important baselines can achieve during inference, as well as sampling speed, both in single sample mode and in batched mode, normalized per sample. We see that while our models employ many parameters, they remain highly memory efficient and are also still as fast as or faster than the baselines. This is due to our efficient non-equivariant transformer architecture. Note that for this table, we also trained an additional smaller model with around 60M parameters, a model size more similar to the baselines – this model is particularly efficient, yet still performs very well (see Table 8). Please see the entire discussion in the newly added Appendix M.2.
>
> >Q1. Input sequence representation.
>
> The input *sequence representation* (see architecture Fig. 6) only consists of the embedded backbone coordinates, as well as a sequence index and an optional self-conditioning input. We call this sequence representation, because the embedded coordinates come in sequential order, and this naming is also consistent with the transformer literature. However, we are *not* modeling any amino acids and our sequence representation does *not* correspond to an amino acid sequence, and there are also no special symbols. Proteina focuses on modeling protein backbone structures.

---

> ### Author Response · Authors · 2024-11-22
> **Response by Authors. Part 2**
>
> >Q2. Fold class classifier and its use in the new metrics.
>
> No, there is no unfair advantage. Our fold class classifier is trained purely for the purpose of evaluation of all models and never used during inference (for sampling we use *classifier-free* guidance).
>
> The experimental setup is analogous to the situation in image generation, where an Inception network-based ImageNet classifier is used universally across the literature to evaluate model performance and calculate FID and IS scores, including for class-conditional generative models trained on ImageNet. We would also like to point out that most of our evaluations (Table 1) are on unconditional generation anyway. We now split off conditional generation in the updated version of the manuscript (Table 2) and evaluated fold class-conditioned Chroma (which uses its own, separate classifier for classifier guidance during inference) as another baseline. Chroma conditions on the same labels as Proteina, thereby making the comparison fair.
>
> It may be possible to train other classifiers or embedding networks for the FPSD metric. However, we leave this to future research. As discussed in the paper, leveraging the fold class labels to train the classifier necessarily results in a classifier sensitive to different protein structures, which is what we require to evaluate our protein structure generative model.
>
> >Q3. Do all residues of protein backbones generated with fold class conditioning use the same CAT label?
>
> Yes, all residues use the same CAT label, this is, each backbone has one global label that defines its overall global fold class. Future work could explore generation with multiple fold classes at the same time, for instance in a multimer generation setting.
>
>
> *We hope that we were able to answer all your questions. We would also like to point out our additional new results that we added to the paper; please see the reply to all reviewers.*

---

> ### Comment · Reviewer_v57v · 2024-11-26
>
> Thank you to the author for their response. Regarding the open-sourcing of code and data, a more proactive approach should be adopted, such as sharing via anonymous URLs. Additionally, research on scaling should include studies on data scaling, which can be easily achieved by reducing the data volume. Overall, the author’s response addresses some concerns, but I still have lingering doubts. I hope the author can address my concerns more proactively, even though I initially gave a high score.

---

### Official Review · Reviewer_WugM · 2024-11-04

**Soundness:** 3
**Presentation:** 4
**Contribution:** 2
**Rating:** 6
**Confidence:** 5

**Summary:**

The paper introduces a novel flow-based generative model for protein backbone design. The model leverages a non-equivariant transformer architecture, adding to the ongoing debate on the applications of SE(3)-equivariant architectures. The proposed model incorporates hierarchical fold class conditioning for controlled generation, and has been trained on a large dataset of 21 million protein structures, more than any of the previous approaches. The authors also introduce distribution-level metrics for evaluation of generative models of protein structures.

**Strengths:**

1. The paper is written exceptionally well. The authors provide a lot of auxillary information and details on nearly every aspect of the work.

2. This work pushes the frontier in terms of the training data scale and parameter count for the protein structure generation models. The authors demonstrate in detail their efficient architecture that enables scaling.

3. The extensive ablation studies and the details on the architecture and experiments are valuable for the community.

**Weaknesses:**

1. The main issue is the experimental design: in the fold class-conditional generation, Proteina is not compared to any other models, while the unconditional generation setting is invalid, since the baselines are trained on different data while being compared to PDB and AFDB. This renders the statements on "vastly outperforming all baselines" false. Another concern about the experimental design is self-evaluation bias: the work introduces a dataset, a model, and a set of evaluation metrics that use their own classifier without external validation.

2. The proposed metrics (FPSD and fJSD) are highly correlated and redundant. Calculation of Frechet distances to PDB and AFDB datasets does not make sense until the models are trained on those datasets. Generative models should generate samples resembling their respective training data, not PDB or AFDB. Either all the baselines should be trained on the same dataset and FPSD and fJSD calculated for this dataset, or another set of metrics should be chosen to evaluate unconditional generation. When using distributional distance metrics, it is a good idea to provide a sense of scale by measuring such distance between random samples in the dataset itself.

3. I agree with the Authors' critique of the existing metrics like diversity and novelty. But there is more to it. The TM-score diversity metric is not informative and lacks an optimal reference point. Providing distributions over TM-scores (e.g., violin plots) would be much more informative than the provided mean over different means over different length ranges. The TM-score expected from the generative model depends on the dataset it is trained on, so providing such value for a subset of a dataset is a good idea. Another issue with metrics is that a single threshold for cluster diversity is not enough as a 50% level cannot detect mode collapse inside the clusters. For example, when each cluster consists of identical structures memorized by the model.

4. The use of a classifier is justified in the experiments that include predictions on the noised data, however, the classifier has not been trained on the noisy data which makes the claim that it can evaluate protein realism weak. If the classifier was trained on the noisy structures, it could be used for classifier guidance.

5. The task scope is limited, which hinders our ability to assess the model. The Authors show unconditional generation and fold class-conditional generation. However, the comparison with other models is poorly set up in the case of unconditional generation and is not performed at all in the case of fold conditioning. The remedy could come from solving other tasks where several baselines exist, for example, motif-scaffolding/inpainting.

**Questions:**

Could you please provide the details on how the GearNet-based classifier was trained? From scratch with contrastive objective as in the original paper, or from the pretrained model, if so, how exactly?

---

> ### Author Response · Authors · 2024-11-22
> **Response by Authors. Part 1**
>
> *We would like to thank the reviewer for their comprehensive review. We appreciate that the reviewer considers our work exceptionally well-written, we thank the reviewer for pointing out that our work pushes the frontier in scaling protein structure generation, and we are grateful for highlighting our extensive experiments and ablations. Below, we address the raised questions:*
>
>
> >1. Performance of Proteina compared to previous work, experimental design, task scope and additional fold class-conditioned baseline.
>
> We would like to point out that **we do indeed outperform all baselines in terms of designability and diversity** in the unconditional generation benchmark in Table 1. Different models and sampling parameters (noise scale gamma) show different tradeoffs, but for instance the M_FS (gamma=0.45) model achieves a designability of 96.4% as well as a cluster-based diversity of 0.63 (305) and a TM-score diversity of 0.36. No baseline scores as well. FrameFlow has similar TM-score diversity, but much worse designability and cluster-based diversity. FoldFlow can achieve slightly higher designability, but has much worse diversity, and both our M_FS (gamma=0.35) model as well as our M_21M (gamma=0.3) model outperform FoldFlow not only in designability, but also diversity. Increasing the noise scale to gamma=0.5 for the M_FS model further boosts diversity, far beyond all baselines, while maintaining above 90% designability. Meanwhile, our M_21M (gamma=0.3) model achieves 99% designability, also far beyond all baselines. In novelty, we are only slightly behind Genie2, which we outperform in designability and diversity.
>
> Moreover, in long chain generation (see Sec. 4.2), we outperform all baselines by a large margin.
>
> Regarding **fold class-conditional generation**, there is only one previous work, Chroma [1], which qualitatively demonstrated fold class conditioning. We now also systematically sampled Chroma with fold-class conditioning and compared with our results in the updated version of the paper. We find that we significantly outperform Chroma in designability and TM-score diversity, and we also generate more designable clusters (see Table 2). Further, we also outperform Chroma in re-classification accuracy, which validates whether specific classes are generated correctly, see Table 6 in Appendix I.1. Chroma fails to reliably generate protein structures according to input query fold class labels. We would like to stress that our work is the first that quantitatively studies fold class conditioning, opening the door towards more advanced control in protein backbone generation and new applications. Note that we have split Table 1 into Table 1 and Table 2 in the updated version to better emphasize that the conditional and unconditional settings are not directly comparable.
>
> Considering the findings in these different experiments, we do believe that we overall indeed outperform relevant previous works, achieving state-of-the-art performance. To be more precise, we slightly rephrased our claim in the last paragraph of the introduction to “*(v)* We achieve state-of-the-art designable and diverse protein backbone generation performance …”.
>
>
> >2. Role of the new metrics (FPSD, FS, fJSD).
>
> In practice, a protein designer is interested in diverse, novel and designable protein backbones. Hence, these are the primary metrics that should be optimized. With this in mind, our new metrics (FPSD, fS, fJSD), which also involve our fold class classifier, are *not* meant to represent “yet another benchmark metric to beat”. Instead, they can provide novel and detailed insights, help analyze results, and guide model development. They are therefore complementary to the other, established metrics. For instance, as discussed in section 4.1, they have been useful to better understand the results on full distribution and fold class-conditional generation. To emphasize that, while the new metrics can bring new insights, they should not be the target of optimization and used to rank models, we have now removed all bold and underline highlighting from the tables for these metrics. We have also added a brief discussion on this at the end of section 3. Our claims regarding state-of-the-art designable and diverse protein structure generation performance are not related to these new metrics, but based on the results discussed above on the conventional metrics. Also, since we are the first to propose such new distribution-focused metrics, we naturally had to train our own classifier, which we rigorously validated and which does not represent any unfair advantage, see Appendix E.

---

> ### Author Response · Authors · 2024-11-22
> **Response by Authors. Part 2**
>
> >3. Use of different training datasets and reference datasets in new metrics.
>
> In fact, all models and baselines in Table 2 are trained on different datasets. Some models use subsets of the AFDB, like Genie 2 and Proteina, while other models leverage subsets of the PDB. Those works using PDB proteins all use different filtering strategies, resulting in different training sets and results with different designability, diversity and novelty tradeoffs.
>
> The crucial point is that in the protein structure generation literature, the choice of the dataset is treated as a hyperparameter itself. This is because from a protein designer’s perspective all that matters is that the model can produce diverse, novel and designable proteins, while a protein designer is less interested in the employed training dataset. Considering that all baselines trained on different data, it is not feasible to re-train all models on the same data.
>
> That being said, our M_FS and M_FS^no-tri models are trained on exactly the same AFDB subset as Genie2, which is the strongest among the baselines. As pointed out above, we do outperform Genie2 in designability and diversity.
>
> Generally, the PDB (natural proteins) and the AFDB (synthetic proteins predicted from AlphaFold2) represent the two main protein structure databases, from which different works and models construct their training datasets. Therefore, we chose a representative subset from the PDB as well as a representative subset from the AFDB as our reference distributions in the FPSD, fJSD and novelty metrics. For novelty, this practice simply follows Genie2, with which we wanted to be consistent. From the protein design perspective, this allows one to study how close in distribution a model’s generated proteins are to either the synthetic proteins from the AFDB or the natural proteins from the PDB, which provides valuable insights. We do agree with the reviewer that the value of the metrics always needs to be considered with a model’s training data in mind. That is another reason why we now removed all bold/underline highlighting for the new metrics (see above). We kept the “higher/lower is better” arrow indications, to indicate general trends; this is, a lower FPSD/fJSD indicates a closer match to the reference distribution and a higher fS indicates more diverse samples.
>
> That aside, the training data for our M_FS and M_FS^no-tri models and for Genie2 is in fact the same as the AFDB reference data used in the different metrics (hence, lower FPSD and fJSD scores indeed indicate a better match of the training data in those cases).
>
>
> >4. TM-score-based diversity, violin plots, and different clustering thresholds for cluster-based diversity.
>
> As discussed above, from a protein designer’s perspective a more diverse model is generally desirable, implying that a lower TM-score-based diversity value is generally better. That said, we agree with the reviewer that it is helpful to study diversity in the context of the diversity observed in the PDB and AFDB, from which the models construct their different training sets. In Table 13 in Appendix M.4, we now report all metrics when evaluated on our PDB and AFDB reference sets used. We added a discussion in the appendix, but would like to highlight the TM-score results here; we find that the data generally has a higher diversity (lower TM-score) than all models and baselines, which is expected and implies that existing models can potentially still be optimized further for diversity.
>
> We also agree that it can be helpful to better analyze the distribution over TM-scores. Following the reviewers’ suggestion, we added violin plots for the TM-score for different lengths for our main models as well as the most relevant baselines (Genie 2, RFDiffusion, FrameFlow), see Figure 21 in Appendix M.3. We find that all models show reasonable TM-score distributions without mode collapse.
>
> We further follow the reviewer’s suggestion and report cluster-based diversity with different thresholds for three of our “best” models as well as the most relevant Genie2, RFDiffusion and FrameFlow baselines (see Table 12 in Appendix M.3). We find that for looser thresholds our M_FS (gamma=0.45) model generally “wins”, while when considering very strict clustering thresholds, all models and baselines produce highly diverse results with many different clusters.

---

> ### Author Response · Authors · 2024-11-22
> **Response by Authors. Part 3**
>
> >5. Fold class classifier and noisy data.
>
> Indeed, if a model generates noisy data, then the classifier may not perform well. This will be reflected in the classifier’s prediction and in its internal feature maps, thereby negatively influencing the FPSD, fJSD and fS metrics that rely on this classifier. However, this is *desirable*, because in that way the metrics can immediately catch noisy generated structures. This is what is validated in the experiments in Appendix E.4. This approach follows the practice from the image generation literature, where the inception-based classifier widely used in the FID [1] and IS [2] metrics is also not specifically trained for noise robustness, thereby leading to metrics sensitive to noisy data.
>
> Finally, we agree that a classifier trained on noisy data could be used for classifier guidance. However, Proteina leverages classifier-free guidance for fold class conditioning and the classifier is only used for evaluation purposes, never during sample generation itself, which ensures that all evaluations are fair.
>
>
> >Q1. GearNet-based classifier details.
>
> We trained the GearNet fold class predictor from scratch, without contrastive pretraining. We leverage the GearNet architecture, but trained the model in a fully supervised fashion using the CAT labels. Note that the original GearNet architecture processes both sequence and structure data; we modify GearNet to focus solely on predicting fold classes based on structure. Please see Appendix E.3, where the fold class predictor training is explained in detail.
>
>
> *We hope we were able to address your questions and concerns. In that case, we would like to kindly ask you to consider raising your score accordingly. Otherwise, please let us know if there is anything else we can clarify. Finally, please see the reply to all reviewers for an overview over all additional new results that we added to the paper.*
>
>
> [1] Heusel et al., GANs Trained by a Two Time-Scale Update Rule Converge to a Local Nash Equilibrium. NeurIPS, 2017.
>
> [2] Salimans et al., Improved Techniques for Training GANs. NeurIPS, 2016.

---

> ### Comment · Reviewer_WugM · 2024-11-28
>
> I appreciate that authors have made several improvements to the manuscript. Adding Chroma to fold class conditioning comparison provides some context for conditional generation evaluation. I appreciate the plots of TM-score distributions and TM-score clustering at different thresholds. These experiments support the concern that mean pairwise TM-score is a rather non-informative metric. The provided clustering and visualization of the TM-score distributions reveal much more details about the generated samples, e.g. a small fraction of duplicate structures. Finally, I found the provided additional details on the training of the GearNet-based classifier important for reproducibility. It would be great to update the manuscript accordingly.
>
> Still, several **major methodological concerns** remain:
>
> 1. Self-evaluation bias - using their own encoder/classifier and metrics without **external validation** raises concerns about objectivity. I acknowledge that many established protein structure encoders like ProteinMPNN or GVP-Transformer cannot be directly applied to CA-only backbones and use 4- and 3-atom-per-residue backbones. Nonetheless, the authors could validate their approach by comparing their encoder/classifier representations of protein backbones with those from established methods on tasks where both are applicable, or by demonstrating performance on independent classification tasks using a separate validation dataset.
>
> 2. The authors' argument that "dataset is a hyperparameter" sidesteps fundamental scientific reproducibility concerns. Furthermore, comparing distributional distances across models trained on different data remains problematic from a methodological perspective, as it confounds model improvements with dataset effects.
>
> 3. The proposed metrics (FPSD/fJSD) may inadvertently favor models trained on data more similar to reference sets. Without controlled comparisons, it becomes difficult to determine whether better scores come from improved modeling capabilities or simply from dataset similarity.
>
> 4. The apparent correlation between FPSD and fJSD metrics raises questions about their redundancy. This correlation could stem from either inherent metric similarity, similar performance patterns across evaluated models, or biases introduced by training and using the classifier on related datasets.
>
> To address these concerns, I suggest the following improvements:
>
> 1.1. Use multiple established structure encoders alongside the self-trained encoder/classifier to provide independent validation
>
> 1.2. Through independent tasks demonstrate what unique structural information is captured by the self-trained encoder/classifier compared to established encoders
>
> 1.3. Add ablation studies using different encoders for the distributional metrics, carefully selecting the data to alleviate potential data leakage.
>
> 2. Add controlled experiments comparing models trained on the same datasets
>
> 3. Provide correlation analysis between the proposed metrics
>
> 4. (Minor) Include reference measurements between random splits of training data to establish scales for distributional metrics
>
>
> Without solving problems with FPSD/fJSD/fS (need for independently validated encoder and inconclusive comparisons between models trained on different datasets), the generation benchmark boils down to one comparison in unconditional generation (with Genie2) and one in conditional (with Chroma) evaluated on Designability, Diversity and Novelty. Given the fair criticism of these metrics from the authors (Section 3.5), with which I completely agree, it becomes crucial to establish more rigorous evaluation protocols. The paper presents important technical advances in scaling protein structure generation, but without addressing these methodological concerns, the comparative results should be interpreted with appropriate caution. I encourage the authors to focus on strengthening the evaluation framework to match the impressive technical achievements demonstrated in the work.

---

> > ### Author Response · Authors · 2024-12-02
> > **Response to Reviewer by Authors. Part 1**
> >
> > We are glad that the reviewer appreciates our additional results and extensive updates of the manuscript, including the Chroma comparison for fold class-conditioned generation as well as the TM-score analyses and the GearNet-based fold class classifier details. We thank the reviewer for highlighting *“the impressive technical achievements demonstrated in the work”*.
> >
> > We also appreciate the additional questions raised by the reviewer and the extensive suggestions for further experiments. **However, we would like to respectfully point out that this is against ICLR’s guidelines: The reviewer waited with their additional reply for 7 days since we presented our initial rebuttal, which includes extensive discussion on all points raised initially as well as many additional experiments addressing all concerns. The reviewer then posted their follow-up reply only on November 28, *after the deadline extension*. The email from the ICLR Program Chairs that informed about the deadline extension clearly states:**
> >
> > *“The intention of this extension is to allow reviewers and authors to address minor issues, and not for the authors to make major changes to the paper. Consequently, between November 26th and December 3rd, authors are only able to post on the forum and not revise the PDF. Reviewers are instructed to not ask for significant experiments, and area chairs are instructed to discard significant requests from reviewers.”*
> >
> > **Yet, the reviewer ignores this guidance and asks for a long list of additional experiments.**
> >
> > That being said, below we will reply to the raised questions one by one:
> >
> >
> > >**Finally, I found the provided additional details on the training of the GearNet-based classifier important for reproducibility. It would be great to update the manuscript accordingly.**
> >
> > We are glad that the description of the training procedure of the GearNet-based classifier is useful. As mentioned in our previous reply, this information is already contained in Appendix E.3.
> >
> >
> > >**1. Using other protein structure encoders or classifiers.**
> >
> > The reviewer questions the use of our classifier that we trained as a fold class predictor and C_alpha backbone embedder for use in our new metrics, suggesting use of *external* models instead. Yet, simultaneously the reviewer acknowledges that applicable external models do not exist. To reiterate, existing models for protein structure representation learning and classification typically use full backbones with amino acid sequences and atomistic details and are not applicable to the setting where only C_alpha backbone atoms are modeled, as in our work (and many others). An exception to this is ProteinMPNN, which can also be run on C_alpha-only backbones, as is done as part of the designability pipeline. However, ProteinMPNN was trained purely for the purpose of inverse folding, that is, sequence prediction, and its internal representations may be less well suited to extract embeddings that are sensitive to both local and global structure variations. Even more importantly, ProteinMPNN predicts sequences, not fold classes, and hence could not be used in our fJSD and FS metrics that directly rely on fold class predictions.
> >
> > Therefore, we believe training a new classifier is an appropriate choice. As we extensively discuss in Appendix E, training on fine-grained hierarchical fold classes that classify different structures naturally enforces the classifier’s embeddings and predictions to be sensitive to complex structure variations. This is validated extensively in Appendix E.4. We would like to point in particular to Table 5 and Figures 11 and 12.
> >
> > That being said, based on the reviewer’s suggestion we did recalculate our FPSD metrics with ProteinMPNN-based embeddings. We cannot update the pdf, but the results are presented at the end of our reply. The finding is that ProteinMPNN-based embeddings overall lead to similar conclusions as using our classifier’s embeddings, just the absolute scale of the metric is different. Note that ProteinMPNN can be considered an *external* model, thereby addressing the reviewer’s concern. The similar results further support that using our own classifier does not imply any unfair advantages. We will include this analysis in the final version of the paper.
> >
> > The reviewer also suggests experiments on other independent classification tasks. Yet, it is not clear what sort of other independent tasks are meant here. Moreover, these comparisons would necessarily be besides the point and inconclusive because, as discussed, existing models generally use other inputs (sequence and/or all-atom structure), which would make a meaningful comparison difficult. Moreover, our classifiers are only meant to perform fold class prediction with internal embeddings sensitive to backbone structure variations. As pointed out above, our classifiers are validated extensively for that, and they are not meant to do any other independent tasks.

---

> ### Author Response · Authors · 2024-12-02
> **Response to Reviewer by Authors. Part 2**
>
> Note that while developing the classifier, we did explore different architectures. In fact, we also experimented with a GVP architecture [1]. Both GearNet and GVP led to very similar, strong performance. Because GearNet is more efficient in this case, we chose it over GVP.
>
> We would also like to stress that the classifier is purely used for evaluation purposes. We never use the classifier during generative model training or inference. Therefore, the use of the classifier is fair among all baselines and there is no “self-evaluation bias”. We would also like to stress that our practice exactly follows what is being done in the large literature on image generative modeling. The Frechet Inception Distance (FID) and the Inception Score (IS) for evaluation of image generators [2,3] are based on an inception network classifier, trained on ImageNet. FID and IS have been used in hundreds of works to evaluate generative models trained both on ImageNet itself, and other datasets. The way in which this inception classifier is used and how it is sensitive to global and local image and texture variations is exactly analogous to how our GearNet classifier is used and how it is sensitive to complex protein structure variations. We are following standard and well-established practices from the large image generation literature.
>
>
> >**2. Models and baselines trained on different datasets. Reproducibility.**
>
> We respectfully disagree with the reviewer on this point. Training models on different datasets has nothing to do with reproducibility. All datasets used in our work are publicly and freely available to both academic and commercial entities and all data processing is explained in detail in the Appendix. Moreover, we now made a version of the code available to the reviewers (for reviewing purposes only), and we will publicly release code upon publication. We believe that we extensively address scientific reproducibility.
>
> Another question is to which degree different models can be compared. We acknowledge that the training dataset does of course play an important role in a model’s performance. However, *all works in protein structure generation do indeed use different datasets*. In this literature, the dataset is indeed treated as a form of hyperparameter. This is because one looks at the problem from the perspective of a protein designer, who does not care about training datasets, but only about the ability of the models to generate designable, diverse and novel proteins — hence, models are compared on these aspects, irrespective of data and architecture details.
>
> To be clear, let us give some examples:
> - FoldFlow trained on 22,248 monomeric samples from the PDB, using samples with <5A resolution, <50% loops, sampling uniformly over clusters with 30% sequence similarity.
> - For FrameFlow, there exist two versions, a stronger one trained on a somewhat similar PDB subset as FoldFlow, and another one trained on a smaller subset of just 3,938 backbones (SCOPe data).
> - Proteus trained on 50,773 structures from the PDB, also using oligomers split into individual chains.
> - Genie 2 trained on 588,570 structures from the AFDB, the same as our D_FS.
> - Chroma is trained on 28,819 protein structures. These contain non-membrane, X-ray structures with resolution <2.8A from the PDB, clustered at 50% sequence identity, taking one sample from each cluster. The data is further enriched with 1726 non-redundant antibodies clustered at 90% sequence identity.
> - RFDiffusion first trains RoseTTAFold, a folding model, additionally leveraging sequence information, in contrast to all other models. Only then it is fine-tuned for generation without sequences. It uses 25,600 samples from the PDB, and the exact filtering and clustering details are unclear.
>
> The chain lengths the models are trained on vary, too. We see that all previous works use different datasets with different preprocessing, filtering, clustering or pre-training steps, all of which matter. And they also all use different architectures. Nonetheless, it is common practice in the literature to compare these models with each other, as has been done in all previous papers in the area (which we cite in our related work section). We would like to acknowledge that we sympathize with the reviewer’s point, though. It could indeed be helpful if the field of protein structure generation established more well-defined benchmarks. However, this is beyond the scope of our paper, and we follow the current practices of the field in our work. In fact, it is one of our key contributions to demonstrate successful training on large-scale protein backbone datasets using efficient and scalable transformer architectures.

---

> > ### Author Response · Authors · 2024-12-02
> > **Response to Reviewer by Authors. Part 3**
> >
> > >**3. Reference datasets, new metrics, and their interpretability.**
> >
> > The reviewer is correct that the new FPSD and fJSD metrics may favor models trained on data more similar to their reference datasets. However, this is expected. The reference datasets are constructed to be representative of the natural proteins from the PDB on the one hand, and of the synthetic AlphaFold2-predicted structures from the AFDB on the other hand. As we discussed in the previous reply and in the reply to all reviewers, these metrics are not meant to be optimized and to serve as scores to rank models. Hence, there are no “better” scores in that sense. Rather, the metrics can measure similarity with the PDB/AFDB reference sets. This becomes very insightful, for instance, when comparing our main unconditional generation results (Table 1), the fold-class conditional generation results (Table 2), and the full-distribution sampling results (Table 3). Fold-class conditional generation significantly reduces FPSD and fJSD, thereby indicating an overall closer match to the PDB and AFDB. Full-distribution sampling further reduces FPSD and fJSD. This does not mean that these different models are better or worse. However, these FPSD and fJSD values are indicative of distributional similarity to PDB or AFDB, which is interesting and relevant in itself. These aspects cannot be measured by the existing metrics as easily.
> >
> > Note that we removed the bold/underline highlighting from the tables for these metrics to make clear that they should not be used to rank models, as pointed out in our initial reply.
> >
> >
> > >**4. Redundancy of FPSD and fJSD metrics.**
> >
> > Yes, the reviewer is correct that FPSD and fJSD are correlated, and this is expected and can be explained. In short, FPSD embeds generated and reference structures in the feature space of the classifier and measures similarity of the feature distributions. fJSD instead predicts fold class labels for generated and reference structures with the same classifier and measures similarity of the categorical label distributions. We see that both metrics follow a very similar recipe, the only difference being that the similarity is either measured in categorical label space or in the continuous feature space (of the same model that is used to also predict the labels).
> >
> > FPSD and fJSD represent two options that one can use to measure the same aspect of protein backbone generative models, this is, the similarity of generated samples with reference distributions, in our case representative subsets of the PDB and AFDB. Future work could use both metrics, or pick one of them only, due to their close relationship. We do not think that this represents a problem. It is clear why the metrics are correlated and our work simply presents both options. We will include discussion on this in the final version of the paper (we cannot update the pdf anymore during this stage of the rebuttal, as mentioned).
> >
> >
> > >**Comparisons and contributions of the paper.**
> >
> > *We would like to point out that we compare our models not only to Genie2 and Chroma, but also FoldFlow, FrameFlow, FrameDiff, Proteus, ESM3 and RFDiffusion, representing all relevant works with openly available models.* Yes, the models are trained on different datasets, but as mentioned above this is standard practice in the field.
> >
> > We would also like to point out that we compare on unconditional generation (Table 1), fold class-conditional generation (Table 2), generation without noise scale or similar modifications (Table 3), as well as long chain generation (Figure 4).
> >
> > We would also like to clarify the role of the different metrics once again. Models are typically scored on novelty, diversity and designability. From a protein designer’s perspective, we believe these are sensible metrics, and scoring the final models on those metrics can be reasonable. However, when developing the underlying generative models, then these metrics can occasionally be somewhat opaque. That is where our new metrics come in. However, our new metrics should only provide additional insights during model development, but they should not serve as optimization targets.

---

> > > ### Author Response · Authors · 2024-12-02
> > > **Response to Reviewer by Authors. Part 4**
> > >
> > > We would also like to remind that our work makes a number of further contributions. In long chain generation we outperform all baselines by a huge margin. Our work is the first to successfully demonstrate diverse and designable backbone generation with up to 800 residues, opening the door to new, larger protein design tasks. The novel fold class conditioning similarly can enable new protein design capabilities. For instance, it can directly enforce secondary structure diversity through conditioning, overcoming the problem of previous models that almost all generate predominantly alpha helices and very few beta sheets (see Table 4). We also showcase LoRA-based fine-tuning of protein structure generators as well as autoguidance, both of which haven’t been demonstrated before for protein structure generation. Our work is also the first to demonstrate that a non-equivariant, highly scalable transformer architecture can be successful in protein backbone generation. We demonstrate successful and efficient training on much more data and with significantly larger models than any previous work. We hope this will inspire future work, further improving and scaling protein structure generative models and their training data and network architectures.
> > >
> > >
> > > *Overall, we would like to thank the reviewer for their feedback. Their review helped us to significantly improve the manuscript with additional experiments and updates. However, as pointed out above, during this stage of the rebuttal phase no new experiments should be requested according to the guidelines, as there is insufficient time to run them, and the manuscript pdf cannot be updated anymore. Despite that, we nonetheless did run a requested experiment, using ProteinMPNN as structure embedder (see below). We will update the final version of the paper with this experiment and consider adding further experiments, analyses and discussion related to the raised points in the final version of the paper. Thank you.*
> > >
> > >
> > > [1] Jing et al., Learning from Protein Structure with Geometric Vector Perceptrons. ICLR, 2021.
> > >
> > > [2] Heusel et al., GANs Trained by a Two Time-Scale Update Rule Converge to a Local Nash Equilibrium. NeurIPS, 2017.
> > >
> > > [3] Salimans et al., Improved Techniques for Training GANs. NeurIPS, 2016.
> > >
> > >
> > > ## **Experiment: Using ProteinMPNN as structure embedder in FPSD.** ##
> > >
> > > Here, we would like to report the results of one additional experiment that the reviewer suggested and that we did run. Instead of relying on our own classifier, we use ProteinMPNN to embed generated and reference structures in the FPSD calculation. Specifically, we take the representations after ProteinMPNN’s three encoder layers, averaging over all residues in the protein to get 128-dim. embeddings.
> > >
> > > We recalculate all FPSD metrics for all experiments, including unconditional, fold class-conditional and full-distribution generation. The results are shown in the table below. We find that we observe similar trends as using our own classifier for embedding, showing significant correlation between our GearNet classifier-based and the ProteinMPNN-based FPSD calculation. For instance, we again see that conditional generation and full-distribution generation reduces the FPSD, indicating a higher match with the reference datasets. The absolute scale of the metric is significantly different, though, which can be explained by the difference in dimensionality of the internal feature layers of ProteinMPNN (128-dim.) and GearNet (512-dim.) and by the difference in the scale of the activations, which is due to the architectural differences. However, in practice only relative comparisons matter, and those generally align well.
> > >
> > > This result indicates that using an *external* model like ProteinMPNN does lead to similar conclusions, implying that there are no unfair conclusions due to the use of our own classifier. We will include these results in the final version of the manuscript.
> > >
> > > **See next message for table with results.**

---

> > > > ### Author Response · Authors · 2024-12-02
> > > > **Response to Reviewer by Authors. Part 5 (results table)**
> > > >
> > > > ### **Table: Results Comparison Between ProteinMPNN FPSD and GearNet FPSD**
> > > >
> > > > | **Model**                  | **PMPNN FPSD** |  |  **GearNet FPSD**      |        |
> > > > |----------------------------|----------------|------------------|--------|--------|
> > > > |                            | **PDB**        | **AFDB**         | **PDB** | **AFDB** |
> > > > |                            |                |                  |        |        |
> > > > | **Unconditional Generation** |||
> > > > |   FrameDiff.               | 0.110          | 0.217            | 194.2  | 258.1  |
> > > > |   FoldFlow (base)          | 0.504          | 0.579            | 601.5  | 566.2  |
> > > > |   FoldFlow (stoc)         | 0.428          | 0.503            | 543.6  | 520.4  |
> > > > |   FoldFlow (OT)            | 0.308          | 0.377            | 431.4  | 414.1  |
> > > > |   FrameFlow                | 0.111          | 0.214            | 129.9  | 159.9  |
> > > > |   ESM3                     | 0.465          | 0.292            | 933.9  | 855.4  |
> > > > |   Chroma                   | 0.116          | 0.176            | 189.0    | 184.1  |
> > > > |   RFDiffusion              | 0.157          | 0.254            | 253.7  | 252.4  |
> > > > |   Proteus                  | 0.154          | 0.213            | 225.7  | 226.2  |
> > > > |   Genie2                   | 0.179          | 0.265            | 350.0    | 313.8  |
> > > > |                            |                |                  |        |        |
> > > > | **$M_{FS}, \gamma=0.35$**           | 0.190          | 0.267            | 411.2  | 392.1  |
> > > > | **$M_{FS}, \gamma=0.45$**           | 0.158          | 0.207            | 388    | 368.2  |
> > > > | **$M_{FS}, \gamma=0.5$**            | 0.128          | 0.184            | 380.1  | 359.8  |
> > > > | **$M^{no-tri}_{FS}, \gamma=0.45$**  | 0.158          | 0.207            | 322.2  | 306.2  |
> > > > | **$M_{21M}, \gamma=0.3$**           | 0.164          | 0.257            | 280.7  | 319.9  |
> > > > | **$M_{21M}, \gamma=0.6$**           | 0.083          | 0.156            | 280.7  | 301.8  |
> > > > | **$M_{LoRA}, \gamma=0.5$**          | 0.109          | 0.232            | 274.1  | 336    |
> > > > |                            |                |                  |        |        |
> > > > | **Fold class-conditional Generation**   |                |                  |        |        |
> > > > |   Chroma                   | 0.100          | 0.132            | 157.8  | 131    |
> > > > | **$M^{cond}_{FS}, \omega=1.0$**     | 0.098          | 0.156            | 121.1  | 127.6  |
> > > > | **$M^{cond}_{FS}, \omega=1.5$**     | 0.094          | 0.143            | 106.1  | 113.5  |
> > > > | **$M^{cond}_{FS}, \omega=2.0$**     | 0.101          | 0.139            | 103.0    | 108.3  |
> > > > |                            |                |                  |        |        |
> > > > | **Full-distribution Generation**           |                |                  |        |        |
> > > > |   Genie2                   | 0.045          | 0.018            | 104.7  | 29.94  |
> > > > | **$M_{FS}$**                   | 0.043          | 0.008            | 85.39  | 21.41  |
> > > > | **$M_{21M}$**                  | 0.025          | 0.049            | 50.14  | 44.98  |
> > > > | **$M_{LoRA}$**                 | 0.028          | 0.130            | 68.56  | 138.6  |
> > > > | **$M^{cond}_{FS}$**            | 0.038          | 0.010            | 71.46  | 19.45  |

---

### Official Review · Reviewer_deDq · 2024-11-04

**Soundness:** 4
**Presentation:** 4
**Contribution:** 3
**Rating:** 5
**Confidence:** 3

**Summary:**

This paper introduces Proteina, a novel flow-based model for protein backbone generation using a scalable non-equivariant transformer architecture. Proteina incorporates fold class conditioning and supports generation of diverse structures up to 800 residues long.

**Strengths:**

1. Very well-written paper and very easy to follow.
2. The authors show that large-scale non-equivariant flow models also succeed on unconditional protein structure generation.

**Weaknesses:**

1. The authors claim to significantly outperform all previous works; however, evidence supporting this assertion is not found in the experimental results table. Excluding unconditional models, there are no direct competitors, and comparisons can only be made with unconditional results. Even if the bold results are accepted as outperforming based on FPSD, FS, fJSD, and TM-score metrics, this model exhibits the lowest diversity.
2. RFdiffusion, ESM3, and Genie 2 were trained on different datasets, while the comparison in Table 1 uses PDB and AFDB, the comparison should be made against their respective training datasets. FPSD, FS, fJSD, and TM-score are uninformative metrics because they lack a defined optimum. To make metrics like FPSD, fS, fJSD, and TM-score meaningful, all competitors should be trained on the same dataset for a fair comparison. Instead of aiming for lower values (fS upper), the goal should be to achieve metrics that match the training dataset. Moreover, comparisons are made with PDB and AFDB without clarifying whether these datasets were part of the training set. If these examples were not used in training, there's no reason to expect lower (fS upper) values. The metrics also lack a clear scale.
3. There are no direct competitors for conditional generation; however, RFdiffusion is capable of conditional generation and could potentially be used for CAT. For a proper comparison with competitors, other tasks such as motif scaffolding, inpainting, and fold conditioning should be addressed.
4. The diversity TM-score is not a particularly informative metric because it lacks a clear optimal value. Instead of simply aiming for a lower score, we should be targeting diversity that aligns with the training dataset. Additionally, only the mean is reported without any measure of deviation, reducing its informativeness. The violin plot provides a better insight, showing the distribution and indicating that there are no identical structures (no collapse mode). When performing an unconditional comparison of the method with itself, the results appear unchanged, suggesting a lack of sensitivity in the metric.
5. The results lack statistical significance.
6. The code is not provided.

**Questions:**

1. The quality of a fold class predictor can be evaluated through classifier-guidance.
2. How was the multiview contrastive learning pretraining conducted for GearNet architechture? Did you initialize with GearNet's existing weights or carry out full pretraining?
3. Can you please justify the use of the exponent in the Inception score equation if you are utilizing a variant without the exponent. Additionally, the Inception score has become less popular in recent years.

---

> ### Author Response · Authors · 2024-11-22
> **Response by Authors. Part 1**
>
> *We would like to thank the reviewer for their comprehensive review. We appreciate that the reviewer considers our work very well-written and easy to follow, and we thank the reviewer for highlighting Proteina’s successful large-scale non-equivariant flow-based generation of protein structures. Below, we address the raised questions:*
>
> >1. Performance of Proteina compared to previous work and additional fold class-conditioned baseline.
>
> We would like to point out that **we do indeed outperform all baselines in terms of designability and diversity** in the unconditional generation benchmark in Table 1. Different models and sampling parameters (noise scale gamma) show different tradeoffs, but for instance the M_FS (gamma=0.45) model achieves a designability of 96.4% as well as a cluster-based diversity of 0.63 (305) and a TM-score diversity of 0.36. No baseline scores as well. FrameFlow has similar TM-score diversity, but much worse designability and cluster-based diversity. FoldFlow can achieve slightly higher designability, but has much worse diversity, and both our M_FS (gamma=0.35) model as well as our M_21M (gamma=0.3) model outperform FoldFlow not only in designability, but also diversity. Increasing the noise scale to gamma=0.5 for the M_FS model further boosts diversity, far beyond all baselines, while maintaining above 90% designability. Meanwhile, our M_21M (gamma=0.3) model achieves 99% designability, also far beyond all baselines. In novelty, we are only slightly behind Genie2, which we outperform in designability and diversity.
>
> Moreover, in the long chain generation experiments (see Sec. 4.2), we outperform all baselines by a large margin.
>
> Regarding fold class-conditional generation, there is only one previous work, Chroma [1], which qualitatively demonstrated fold class conditioning. We now also systematically sampled Chroma with fold-class conditioning and compared with our results in the updated version of the paper. We find that we significantly outperform Chroma in designability and TM-score diversity, and we also generate more designable clusters (see Table 2). Further, we also outperform Chroma in re-classification accuracy, which validates whether specific classes are generated correctly, see Table 6 in Appendix I.1. Chroma fails to reliably generate protein structures according to input query fold class labels. We would like to stress that our work is the first that quantitatively studies fold class conditioning, opening the door towards more advanced control in protein backbone generation and new applications. Note that we have split Table 1 into Table 1 and Table 2 in the updated version to better emphasize that the conditional and unconditional settings are not directly comparable.
>
> Considering the findings in these different experiments, we do believe that we overall indeed outperform relevant previous works, achieving state-of-the-art performance. To be more precise, we slightly rephrased our claim in the last paragraph of the introduction to “*(v)* We achieve state-of-the-art designable and diverse protein backbone generation performance …”.
>
> >2. Role of the new metrics (FPSD, FS, fJSD).
>
> In practice, a protein designer is interested in diverse, novel and designable protein backbones. Hence, these are the primary metrics that should be optimized. With this in mind, our new metrics (FPSD, fS, fJSD) are *not* meant to represent “yet another benchmark metric to beat”. Instead, they can provide novel and detailed insights, help analyze results, and guide model development. They are therefore complementary to the other, established metrics. For instance, as discussed in section 4.1, they have been useful to better understand the results on full distribution and fold class-conditional generation. To emphasize that, although the new metrics can bring new insights, they should not be the target of optimization and used to rank models, we have now removed all bold and underline highlighting from the tables for these metrics. We have also added a brief discussion on this at the end of section 3. Our claims regarding state-of-the-art designable and diverse protein structure generation performance are not related to these new metrics, but based on the results discussed above.
>
> Regarding the scale of the metrics, we would like to emphasize that we are primarily interested in relative comparisons between the methods, not absolute values. We could easily re-scale the metrics to vary in any desired interval, but this would not lead to additional insights.

---

> ### Author Response · Authors · 2024-11-22
> **Response by Authors. Part 2**
>
> >3. Use of different training datasets and reference datasets in new metrics.
>
> In fact, not only RFDiffusion, ESM3 and Genie2 were trained on different datasets, but all baselines use different training sets (those works training on the PDB all use differently filtered PDB subsets). In the protein structure generation literature, the choice of the dataset is treated as a hyperparameter itself. This is because from the protein designer perspective all that matters is that the model can produce diverse, novel and designable proteins, while a protein designer is not interested in the training dataset (hence, a better TM-score-based diversity is generally desirable, irrespective of the training data). Considering that all baselines trained on different data, it is not feasible to re-train all models on the same data.
>
> That being said, our M_FS and M_FS^no-tri models are trained on exactly the same AFDB subset as Genie2, which is the strongest among the baselines. As pointed out above, we do outperform Genie2 in designability and diversity. Regarding the question whether PDB/AFDB data was part of training, please see Sec. 3. In short, our model employs two different AFDB subsets. PDB training data is only leveraged in the LoRA fine-tuning experiment.
>
> Generally, the PDB (natural proteins) and the AFDB (synthetic proteins predicted from AlphaFold2) represent the two main protein structure databases, from which different works and models construct their training datasets. Therefore, we chose a representative subset from the PDB as well as a representative subset from the AFDB as our reference distributions in the FPSD and fJSD metrics. From the protein design perspective, this allows one to study how close in distribution a model’s generated proteins are to either the synthetic proteins from the AFDB or the natural proteins from the PDB, which provides valuable insights. We do agree with the reviewer that the value of the metrics always needs to be considered with a model’s training data in mind. That is another reason why we now removed all bold/underline highlighting for the new metrics (see above). We kept the “higher/lower is better” arrow indications, to indicate general trends; this is, a lower FPSD/fJSD indicates a closer match to the reference distribution and a higher fS indicates more diverse samples.
>
>
> >4. Competitors for fold-class conditional generation.
>
> RFDiffusion can be conditioned on pairwise distance maps between residues, which can be used to also specify certain folds in a fine-grained manner, but RFDiffusion cannot be conditioned on the broad and universal CAT fold class labels that we leverage and that provide a semantic and hierarchical way to control the backbone generation at different levels of detail. Therefore, a meaningful comparison with RFDiffusion is not possible. However, Chroma [1] can indeed be used with CAT fold class conditioning, leveraging classifier guidance (in contrast to our *classifier-free* guidance). Hence, in the updated version we evaluated fold class-conditioned Chroma as a new baseline, which we outperform (discussed already above in point 1).
>
>
> >5. TM-score-based diversity and violin plots.
>
> As discussed above, from a protein designer’s perspective a more diverse model is generally desirable, implying that a lower TM-score-based diversity value is generally better. That said, we agree with the reviewer that it is helpful to study diversity in the context of the diversity observed in the PDB and AFDB, from which the models construct their different training sets. In Table 13 in Appendix M.4, we now report all metrics when evaluated on our PDB and AFDB reference sets used. We added a discussion in the appendix, but would like to highlight the TM-score results here; we find that the data generally has a higher diversity (lower TM-score) than all models and baselines, which is expected and implying that existing models can still potentially be optimized further for diversity.
>
> We also agree that it can be helpful to better analyze the distribution over TM-scores. Following the reviewers’ suggestion, we added violin plots for the TM-score for different lengths for our main models as well as the most relevant baselines (Genie 2, RFDiffusion, FrameFlow), see Figure 21 in Appendix M.3. We find that all models show reasonable TM-score distributions without mode collapse.

---

> ### Author Response · Authors · 2024-11-22
> **Response by Authors. Part 3**
>
> >6. Statistical significance and sensitivity in the metric.
>
> In the newly added Appendix section M.5 (see Table 14), we repeatedly evaluate one of our models with different seeds, observing very little variation of the results, thereby demonstrating that our results can be considered statistically significant. Estimating the exact standard deviations for all models and baselines through repeated evaluation is unfortunately computationally infeasible due to the massive amount of sampling required and hence beyond the scope of the rebuttal. However, we can expect similarly accurate results for all other models and baselines as for the one for which we did the error estimation here.
>
> We would also like to point out that all metrics vary across our different models and the baselines, thereby indicating that all metrics are sensitive to the different generated samples.
>
>
> >7. The code is not provided.
>
> Proteina will be made available publicly after publication. Also note that all data sources are publicly available under permissive licenses, and the data preparation protocol is described in detail in Appendix B. Moreover, all training, sampling and hyperparameter details are covered in Appendices F and K.
>
>
> >Q1. The quality of a fold class predictor can be evaluated through classifier-guidance.
>
> It is not clear to us what exactly is meant with this question. However, we would like to point out that our fold class predictor is validated by evaluation on a held-out test set (see Appendix E.3). Further, whether our fold class-conditioned Proteina models accurately generate proteins from the correct fold classes is then evaluated by re-classifying the generated backbones with the fold class predictor and seeing if the predictions match the input conditioning (see Appendix H). We would like to kindly ask the reviewer to clarify their question, in case our reply is not what they were looking for.
>
>
> >Q2. GearNet-based classifier details.
>
> We trained the GearNet fold class predictor from scratch, without contrastive pretraining. We leverage the GearNet architecture, but trained the model in a fully supervised fashion using the CAT labels. Note that the original GearNet architecture processes both sequence and structure data; we modify GearNet to focus solely on predicting fold classes based on structure. Please see Appendix E.3, where the fold class predictor training is explained in detail.
>
>
> >Q3. Use of inception score and exponentiation.
>
> Our definition of the fold Score (fS) follows the standard definition of the inception score in computer vision [2], *which includes the exponentiation* of the Kullback-Leibler term, see definition in Sec. 3.5. This exponentiation ensures that the scale of the score is in ~[1, N], where N is the number of classes, making the metric more interpretable. Although indeed less popular, the inception score nonetheless remains a relevant metric to quantify a generative model’s coverage of a class distribution. This is particularly interesting when quantifying a class-conditional model like ours, which builds on classifier-free guidance. In fact, the inception score, together with the FID, was used in the original classifier-free guidance paper to analyze the results [3]. Since we similarly use classifier-free guidance (for the first time) for protein generation, we believe it is similarly a good idea to study our fold Score (fS), our protein backbone version of the inception score.
>
>
> *We hope we were able to address your questions and concerns. In that case, we would like to kindly ask you to consider raising your score accordingly. Otherwise, please let us know if there is anything else we can clarify. Finally, please see the reply to all reviewers for an overview over all additional new results that we added to the paper.*
>
>
> [1] Ingraham et al., Illuminating protein space with a programmable generative model. Nature 2023.
>
> [2] Salimans et al., Improved Techniques for Training GANs. NeurIPS, 2016.
>
> [3] Ho and Salimans, Classifier-Free Diffusion Guidance. arXiv, 2022.

---

### Official Review · Reviewer_KGtK · 2024-11-07

**Soundness:** 3
**Presentation:** 3
**Contribution:** 3
**Rating:** 6
**Confidence:** 4

**Summary:**

The paper introduces Proteina, which is a flow-matching based protein structure generative model. The authors make efforts to explore the scalability of the new structure and the training data which results in the largest model with impressive performance. The model introduces several interesting techniques for boosting performance including the recent proposed auto-guidance. The paper proposes several new evaluation metrics with extensive baselines to demonstrate the effectiveness of the Proteina.

**Strengths:**

1. The paper is certainly well written and I do enjoy the reading.

2. The paper makes several very interesting yet important explorations and observations.  For example, though AF3 already observes the Equivariant vs Non-equivariant properties, it would be nice to further explore the scalability with non-equivariant transformers; The auto guidance parts of generation also provides some new insights into the protein structure generation; Studying protein structure generation in scale is also an important research direction for the community.

3. I appreciate the efforts of providing comprehensive and sound evaluations for the protein structure generation tasks. Specifically, apart from the previously used simple quality/diversity measurements, the introduced FPSD/FjSD/fs could provide a fresh evaluation view at the distributional level. And I believe it could make more impact in facilitating future research.

**Weaknesses:**

1. Though with a scaled structure, it would be better to understand the training in a more systematic way, e.g. scaling laws. The trained flow matching model in general could still obtain the corresponding likelihood generally, could Proteina also conduct a likelihood evaluation over the protein structures? Is it possible to study the scaling laws based on that?

2. The notation of Table 1 for models with different configs is not very clear which makes it hard to read and analyze.  I also suggest the authors list the parameter size of the proposed models and baseline in an extra column. So considering the claimed fact that the proposed model is 5 times more params than previous models, the unconditional performance of M_{21M} is worse than previous methods in the sense of FPSD/ FS/diversity, for example, FrameFlow, etc. This raises a major concern of whether the general framework and formulation could provide benefits. I would like to suggest the authors elaborate more on this.

I would like to consider increasing my score if the concerns are fixed.

**Questions:**

refer to above

---

> ### Author Response · Authors · 2024-11-22
> **Response by Authors.**
>
> *We would like to thank the reviewer for their comprehensive review. We appreciate that the reviewer considers our work well-written and enjoyed reading it. We would also like to thank the reviewer for highlighting our various interesting explorations and evaluations, pointing out the impact on future work. Below, we address the raised questions:*
>
> >1. Scaling Laws.
>
> In our case of flow matching with Gaussian conditional probability paths, the flow matching loss can be seen as a rescaled version of the evidence lower bound (see [1,2,3] for details), thereby being closely related to the likelihood. Similar to recent seminal work in image generation [4], which analyzed the scaling of the flow matching objective, we therefore also directly analyze the flow matching loss for differently-sized Proteina transformer models. In the newly added Fig. 19 in Appendix M.1, we see that scaling model size indeed systematically reduces the loss, implying a strong scaling behavior of our architecture.
>
> >2. Models in Table 1, and number of parameters.
>
> We trained three separate models in our paper, *M_FS*, *M_FS^no-tri*, and *M_21M*, see first paragraph of Sec. 4. The “FS/21M” subscript denotes which data the model was trained on (see Sec. 3). “No-tri” indicates that no triangular multiplicate layers were used (see discussion in Sec. 3.3). Each model is trained such that it can be sampled both fold-class conditioned, or in an unconditional manner (see Sec. 3.2). Moreover, varying the noise scale $\gamma$ during inference also leads to different results. These aspects result in different evaluation setups, presented in the rows of Tables 1 and 2.
>
> To not make these tables even larger, we added the number of parameters as part of the Tables 9, 10 and 11 in Appendix M.2. We list the number of parameters for all our models as well as the most relevant baselines. Our Proteina models may seem large and expensive by parameter count, but what matters in practice is not an absolute parameter count, but a model’s memory consumption and sampling speed, along with performance (which is reported in the main paper already, in Table 1). Therefore, in Figure 20 and  Tables 9, 10 and 11, we also show the maximum batch size per GPU that our models and the most important baselines can achieve during inference, as well as sampling speed, both in single sample mode and in batched mode, normalized per sample. We see that while our models employ many parameters, they remain highly memory efficient and are also still as fast as or faster than the baselines. This is due to our efficient non-equivariant transformer architecture. Note that for this table, we also trained an additional smaller model with around 60M parameters, a model size more similar to the baselines – this model is particularly efficient, yet still performs very well (see Table 8). Please see the entire discussion in the newly added Appendix M.2.
>
> >3. Performance of the *M_21M* model.
>
> The reviewer also pointed to the performance of the *M_21M* model. The fact that this model exhibits a different performance tradeoff compared to our other models and several baselines is **not** because of the model architecture. It is because it was trained on the *D_21M* dataset, which is subject to careful filtering to only contain high-quality well-structured backbones (see Sec. 3.1). Hence, this leads to slightly lower diversity, but state-of-the-art 99% designability (see Tab. 1). Our motivation to train this model is to show a proof of concept that one can curate such very large-scale high-quality training sets from the AFDB that result in extremely high designability, despite the fully synthetic nature of the AFDB database, while leveraging scalable architectures like ours (previous training sets are more than an order of magnitude smaller, see Sec. 3.1). Future work could build on this observation and curate different large-scale training sets from the AFDB and other sources, using different filters, tailored to different needs.
>
>
>
> *We hope we were able to address your questions and concerns. In that case, we would like to kindly ask you to consider raising your score accordingly. Otherwise, please let us know if there is anything else we can clarify.*
>
> *Finally, we would also like to point out our other additional new results that we added to the paper; please see the reply to all reviewers.*
>
> [1] Ho et al., Denoising Diffusion Probabilistic Models. NeurIPS, 2020.
>
> [2] Albergo et al., Stochastic Interpolants: A Unifying Framework for Flows and Diffusions. arXiv, 2023.
>
> [3] Kingma and Gao, Understanding Diffusion Objectives as the ELBO with Simple Data Augmentation. NeurIPS, 2023.
>
> [4] Esser et al., Scaling Rectified Flow Transformers for High-Resolution Image Synthesis. ICML 2024.

---

> > ### Comment · Reviewer_KGtK · 2024-11-25
> > **Thanks for your response**
> >
> > The response fixes my main concerns about performance and scalability. I appreciate the efforts of the authors in replying my questions.
> >
> > Besides, I am curious that considering the performance only, would flow matching-based approaches outperform the diffusion-based approaches over protein backbone generation? If so, it is there any intuitions or insights behind that?

---

> ### Author Response · Authors · 2024-11-25
> **Response to Follow-up Question. Part 1**
>
> *We are glad that we were able to address your concerns and are happy to also provide some insights regarding your follow-up question.*
>
> >I am curious that considering the performance only, would flow matching-based approaches outperform the diffusion-based approaches over protein backbone generation? If so, it is there any intuitions or insights behind that?
>
> To answer this question, we would like to first point out that we are using flow matching to couple the training data distribution (the protein backbones for training) with a *Gaussian noise distribution*, from which the generation process is initialized when sampling new protein backbones after training. In this case, i.e. when coupling with a Gaussian distribution, flow matching models and diffusion models can actually be shown to be equivalent, up to reparametrizations. This is because diffusion models generally use a Gaussian diffusion process, thereby also defining Gaussian conditional probability paths, similar to the Gaussian conditional probability paths in flow matching with a Gaussian noise distribution.
>
> For instance, when using a Gaussian noise distribution, one can rewrite the velocity prediction objective used in flow matching as a noise prediction objective, which is frequently encountered in diffusion models. Different noise schedules in diffusion models can be related to different time variable reparametrizations in flow models. Most importantly, for Gaussian flow matching, we can derive a relationship between the *score* of the interpolated distributions $\nabla_{\mathbf{x}_t}\log p_t(\mathbf{x}_t)$ and the flow’s velocity (see Eq. (3) in our paper as well as Appendix F.1). The score is the key quantity in score-based diffusion models. Using this relation, diffusion-like stochastic samplers for flow models can be derived, as well as flow-like deterministic ODE samplers for diffusion models (see below). In conclusion, we could in theory equally consider our flow models score-based diffusion models. With that in mind, to answer your question, from a pure performance perspective when coupling with a Gaussian noise distribution flow matching-based approaches and diffusion-based approaches should in principle perform similarly well. In practice, it all boils down to choosing the best training objective formulation, the best time sampling distribution to give appropriate relative weight to the objective (see Section 3.2), etc. – these aspects dictate model performance, independently of whether one approaches the problem from a diffusion model or a flow matching perspective.
>
> In fact, we directly leverage the connections between diffusion and flow models when developing our stochastic samplers (see Appendix F.1) and guidance schemes. Both classifier-free guidance [1] and autoguidance [2] were proposed for diffusion models, but due to the relations between score and velocity, we can also apply them to our flow models (to the best of our knowledge, our work is the first to demonstrate classifier-free guidance and autoguidance for flow matching of protein backbone generation). Please see our Appendix F.2, which has all technical details.
>
> Considering these relations, why did we then overall opt for the flow matching formulation and perspective? (i) Flow matching is somewhat simpler to implement and explain, as it is based on simple interpolations between data and noise samples. No complex stochastic diffusion processes need to be considered. (ii) Flow matching offers the flexibility to be directly extended to more complex interpolations, beyond Gaussians and diffusion-like methods. For instance, we may consider optimal transport couplings [3,4] to obtain straighter and faster generation paths or we could explore other more complex non-Gaussian noise distributions. We plan to further improve Proteina in the future and flow matching offers more flexibility in that regard. At the same time, when using Gaussian noise, all tricks from the diffusion literature still remain applicable. (iii) The popular and state-of-the-art large-scale image generation system Stable Diffusion 3 is similarly based on flow matching [5]. This work unambiguously demonstrated that flow matching can be scaled to large-scale generative modeling problems.
>
> *(continued below)*

---

> > ### Author Response · Authors · 2024-11-25
> > **Response to Follow-up Question. Part 2**
> >
> > We would like to point out that the relations between flow-matching and diffusion models have been discussed in various papers. One of the first works pointing out the relation is [6] and the same authors describe a general framework in *Stochastic Interpolants* [7], unifying a broad class of flow, diffusion and other models. Some of the key relations and equations can also be found more concisely in [8]. The relations between flow matching and diffusion models have also been highlighted in Appendix D of [9]. The first work scaling flow matching to large-scale text-to-image generation is the above mentioned [5], which also systematically studies objective parametrizations and time sampling distributions, similarly leveraging the relation between flow and diffusion models.
> >
> > The equivalence between the flow matching and diffusion model frameworks has been pointed out in our paper at end of the third paragraph of the background section 2 (lines 133-135). It is also briefly discussed in Section 3.4 and further used in Appendices F.1 and F.2.
> >
> > *We hope that we were able to provide helpful insights and explanations. We would like to kindly ask you to consider raising your score, also considering our previous reply that addressed your previous concerns, as well as the extended results (see reply to all reviewers). Thank you!*
> >
> >
> > [1] Karras et al., Guiding a Diffusion Model with a Bad Version of Itself. NeurIPS, 2024.
> >
> > [2] Ho and Salimans, Classifier-Free Diffusion Guidance. arXiv, 2022.
> >
> > [3] Pooladian et al., Multisample flow matching: straightening flows with minibatch couplings. ICML, 2023.
> >
> > [4] Tong et al., Improving and generalizing flow-based generative models with minibatch optimal transport. TMLR, 2024.
> >
> > [5] Esser et al., Scaling Rectified Flow Transformers for High-Resolution Image Synthesis. ICML, 2024.
> >
> > [6] Albergo and Vanden-Eijnden, Building Normalizing Flows with Stochastic Interpolants. ICLR 2023.
> >
> > [7] Albergo et al., Stochastic Interpolants: A Unifying Framework for Flows and Diffusions. arXiv, 2023.
> >
> > [8] Ma et al., SiT: Exploring Flow and Diffusion-based Generative Models with Scalable Interpolant Transformers. ECCV, 2024.
> >
> > [9] Kingma and Gao, Understanding Diffusion Objectives as the ELBO with Simple Data Augmentation. NeurIPS, 2023.

---

> > > ### Comment · Reviewer_KGtK · 2024-11-26
> > > **Thanks for your discussion！**
> > >
> > > Thanks a lot for your detailed discussion which help me better understand the motivation behind the framework. I hope the corresponding discussion can be summarized and integrated into the updated draft. I update my score and vote for acceptance.

---

> > > > ### Author Response · Authors · 2024-11-26
> > > > **Thank you!**
> > > >
> > > > Thank you! We are glad that we were able to provide helpful insights. We have refined and now added the discussion to the manuscript in an additional Appendix N.

---

### Official Review · Reviewer_Yrch · 2024-11-08

**Soundness:** 4
**Presentation:** 4
**Contribution:** 4
**Rating:** 8
**Confidence:** 3

**Summary:**

This paper introduces Proteína, a foundational model for generating protein backbones with innovative fold class conditioning for precise control over structure. Proteína achieves state-of-the-art results, synthesizing diverse backbones up to 800 residues long using a scalable transformer model. It leverages a 21M-sized dataset from the AFDB and, at over 400M parameters, can generate highly designable proteins even with synthetic data. The model pioneers classifier-free, autoguidance, and LoRA-based fine-tuning in protein structure generation, and introduces novel metrics for assessing generator behavior.

**Strengths:**

1. The paper introduces novel metrics that address previously omitted distribution-level aspects of protein generation, which is both valuable and innovative, allowing for a more comprehensive evaluation of model performance. Additionally, the scaling of both training data and model aligns with the evolution of the field of protein generation.
2. The paper proposes an innovative $t$ sampling method that effectively captures the unique characteristics of protein data. This is also the first application of classifier-free guidance (CFG) and autoguidance in protein structure generation.
3. The methodology is presented with commendable clarity and detail.

**Weaknesses:**

In line 119, a partial derivative seems mistakenly written as a total derivative, and the divergence is incorrectly labeled as a gradient. I believe the right form of the continuity equation should be like $\partial p_t(\boldsymbol x_t)/\partial t=-\nabla_{\boldsymbol x_t}\cdot(p_t(\boldsymbol x_t)\boldsymbol u_t(\boldsymbol x_t))$.
Additionally, the differential symbol should be formatted in upright type, as $\mathrm{d}$, to follow standard conventions.

**Questions:**

1. Given that the non-equivariant model still relies on random rotations as data augmentation, is the choice to adopt a non-equivariant transformer a fundamental advantage over equivariant models, or simply a practical compromise? In other words, is it reasonable to expect that, with the current dataset and model architecture, introducing equivariance into the model could further improve performance?
2. The three newly introduced metrics (FPSD, fS, fJSD) each seem to reflect certain distributional similarities between the generated data and the reference dataset. Could the authors clarify the specific focus of each metric? Additionally, is there a possibility of merging these metrics to produce a more unified and intuitive measure?

---

> ### Author Response · Authors · 2024-11-22
> **Response by Authors. Part 1**
>
> *We would like to thank the reviewer for their positive evaluation of our work. Below, we address the raised questions:*
>
> >Continuity equation and differential symbols.
>
> We thank the reviewer for pointing this out. We have fixed the continuity equation in the updated version of the manuscript and also modified all differential symbols to be upright.
>
> >Q1. Choice of non-equivariant transformer, data augmentations, and performance.
>
> We chose a non-equivariant transformer as the main backbone architecture in Proteina primarily for practical and performance reasons. Building strict equivariance into networks often comes with severe architecture constraints, and the most powerful equivariant architectures often scale less gracefully than non-equivariant models when training on large and complex data. For instance, in protein structure generation, triangle attention and triangle multiplicative layers are common network components employed in high-performance equivariant architectures [1,2,3]. However, their memory consumption scales cubically with sequence length, making them highly expensive when applied to large biomolecular systems. Other equivariant networks involve very computationally expensive and memory consuming higher-order tensor products [4]. In contrast, non-equivariant transformers have been shown to scale to large-scale generative modeling tasks in the language and image modeling literature. This is the main reason why we primarily rely on non-equivariant transformer architectures also in our work. In fact, this scalability is what allowed us for the first time to scale protein backbone generation to unprecedented chain lengths of up to 800 residues (see Sec. 4.2 in the paper).
>
> Note that in some models we do in fact also use triangle layers to boost performance, but our main driving backbone architecture is a transformer that performs extremely well also on its own, as manifested for instance in the long chain generation experiments.
>
> Since the group over 3D rotations is “small”, we can nonetheless achieve approximate equivariance even with our non-equivariant architectures through data augmentation, improving the model’s generalization (also see the related analysis in Appendix G). It is an interesting question to explore whether re-introducing strict equivariance into our network architecture itself would further boost performance. However, as discussed, strictly equivariant architectures often do not scale well. Hence, it is questionable whether this would lead to actual benefits, when relying on current equivariant network layers. Future work could explore developing novel, highly expressive, yet computationally efficient and scalable equivariant architectures for protein structure generation.

---

> ### Author Response · Authors · 2024-11-22
> **Response by Authors. Part 2**
>
> >Q2. Clarifying the specific focus of each of the new metrics (FPSD, fS, fJSD), and possibility to merge into a more unified measure.
>
> The Frechet Protein Structure Distance (FPSD) is inspired by the Frechet Inception Distance [5] widely used in image generation. It embeds sets of generated and reference protein backbones into a feature space and measures a Wasserstein distance between the embedding distributions. Therefore, FPSD measures distributional similarity between generated and reference backbones at the structure level, including detailed structure variations.
>
> The fold Jensen-Shannon Divergence (fJSD) is a somewhat similar measure, but only operates at the coarser fold class level. Specifically, sets of generated and reference backbones are classified with our fold class classifier, and then the predicted categorical fold class label distributions are compared, using the Jensen-Shannon Divergence. In contrast to the FPSD, fJSD is less sensitive to local structure variations, but instead measures fold class-level similarity between generated and reference data sets.
>
> The fold Score (fS) is inspired by the Inception Score in computer vision. This metric does not rely on a reference distribution and instead measures how well a set of generated protein backbones covers all possible fold classes. Hence, we can interpret the fS as a novel fold class-based diversity measure.
>
> Both fJSD and fS can be evaluated at different levels of the CAT fold class hierarchy, thereby operating at coarser or more fine-grained fold class levels (we aggregate the fJSD for all different levels of the CAT hierarchy into one score). We would like to point the reviewer to Appendix E, where the metrics are discussed, introduced and validated in detail.
>
> In theory, one could probably merge all new metrics into a single score. However, we believe that this would not be intuitive nor helpful, and goes against our intentions with these metrics. As discussed, the different metrics capture different aspects, and these would not be discernable anymore if combined into one “average” unified score. We would like to emphasize that we did not introduce these metrics just to have “yet another benchmark metric to beat”, but rather to provide novel and detailed insights (see discussions in Sec. 4.1). To this end, it is beneficial to be able to study the metrics individually, enabling different analyses.
>
>
> *We hope we were able to answer your questions. Please let us know if there is anything else we can clarify. We would also like to point out our additional new results that we added to the paper; please see the reply to all reviewers.*
>
> [1] Jumper et al., Highly accurate protein structure prediction with Alphafold. Nature, 2021.
>
> [2] Lin et al., Out of Many, One: Designing and Scaffolding Proteins at the Scale of the Structural Universe with Genie 2. arXiv, 2024.
>
> [3] Huguet et al., Sequence-Augmented SE(3)-Flow Matching For Conditional Protein Backbone Generation. arXiv, 2024.
>
> [4] Duval et al., A Hitchhiker's Guide to Geometric GNNs for 3D Atomic Systems. arXiv, 2023.
>
> [5] Heusel et al., GANs Trained by a Two Time-Scale Update Rule Converge to a Local Nash Equilibrium. NeurIPS, 2017.
>
> [6] Salimans et al., Improved Techniques for Training GANs. NeurIPS, 2016.

---

> > ### Comment · Reviewer_Yrch · 2024-11-26
> >
> > Thank you for the detailed reply, which has addressed my questions raised. I keep my positive rating for this work.

---

> ### Author Response · Authors · 2024-12-02
> **Thank you.**
>
> We are glad that we were able to address the questions. Thank you for the helpful review and constructive feedback.

---

### Author Response · Authors · 2024-11-22
**Overall Response by Authors**

We would like to thank all reviewers for their feedback. We are delighted to see that our work is generally well received and that not only our various technical contributions, such as the fold class conditioning, and our experimental results have been highlighted, but also our paper’s presentation. We appreciate that several reviewers acknowledged that Proteina advances the state of the art in protein backbone design and pushes the frontier in terms of scale with respect to data and model size.

Considering the reviewers’ feedback, we added several experiments and made modifications to the manuscript. We aggregated most additional experimental results in an additional section M in the appendix to make them easy to find for the reviewers. For the camera-ready version, we will rearrange the appendix.

### **We made the following additions and modifications:**


1. As an **additional baseline for fold class-conditional generation**, we now also evaluated Chroma, the only previous work that can condition on fold class labels (via classifier guidance). See Table 2, main paper, and Appendix I.2. We find that we outperform Chroma; in particular, Chroma has poor designability (*Table 2*), and we show that it fails to accurately generate the correct fold classes (*Appendix I.2*).
2. We analyze **how the optimization of the flow matching loss scales with model size**, following similar analyses in previous work on state-of-the-art image generation (Esser et al., Scaling Rectified Flow Transformers for High-Resolution Image Synthesis. ICML, 2024). We find that we can consistently improve the objective as we scale the model size. *See Appendix M.1.*
3. We analyzed **inference speed, memory consumption and model size** for our main models and the baselines. We find that even though our models are large in terms of parameters, and show state-of-the-art quantitative performance, they are still memory efficient and can be sampled quickly, thanks to our efficient transformer architecture. See *Appendix M.2.*
4. We performed a more **fine-grained analysis of the diversity metrics**. For the diversity metric based on pairwise TM-scores, we show violin plots for our models and key baselines. We find that all models show reasonable distributions without mode collapse. Further, we analyse the effect of the clustering threshold for the clustering-based diversity metric. *See Appendix M.3.*
5. Some of our analyses rely on representative reference subsets from the PDB (natural proteins) and the AFDB (synthetic AF2 protein structures), and also our models and the baselines construct their different training sets from the PDB and AFDB databases. **We evaluated all metrics with respect to the two PDB and AFDB reference sets**, thereby providing valuable context, such as the designability or diversity of the data itself. *See Appendix M.4.*
6. For one of our models, we repeatedly evaluate the model with different seeds, observing little variation of the results, thereby demonstrating that **our results can be considered statistically significant**. *See Appendix M.5.*
7. **Other updates:** We have split the original Table 1 in the main text into Table 1 and 2, separately focusing on unconditional and fold class-conditional backbone generation. We now also emphasize more that our new metrics are meant to guide model development and provide new insights, but they are not meant as optimization targets to rank models. Hence, we removed all bold/underline highlighting for the new metrics in the results tables. We also added a discussion at the end of section 3. To save space, we moved our novel t-sampling distribution figure and its discussion to a new Appendix section G.

*We believe that these additional results further improve our work, and we would like to thank the reviewers for the suggestions.*

---

### Meta-Review · Area_Chair_oGxy · 2024-12-22

**Metareview:**

The paper introduces a flow-based generative model for protein backbone design, leveraging hierarchical fold class labels and a scalable non-equivariant transformer architecture, and it also introduces novel evaluation metrics for protein generation. Reviewers appreciated the paper's clarity of presentation, the introduction of novel metrics and the detailed empirical evaluation using them, the ablations studies, and the model's state-of-the-art capabilities in backbone design. However, the reviewers also raised concerns about comparisons being complicated by using new metrics relying on an internal classifier, self-evaluation bias in the new metrics, baselines being trained on different datasets complicating direct comparisons, statistical significance, and reproducibility being better supported by open-source resources. The rebuttal and discussion addressed many of these issues by adding additional baseline comparisons (with Chroma for fold class-conditional generation), repeated experiments with different seeds for evaluating statistical significance, additional experiments with using ProteinMPNN as structure embedded in FPSD to address potential self-evaluation bias, and sharing code. Four out of five reviewers recommend accepting the paper, mainly due to its state-of-the-art performance and significant technical contributions to scaling protein structure generation. Therefore, I recommend accepting the paper.

The acceptance is conditioned on the authors making the code publically available for the camera version of the paper, since code availability played a central role in the review process.

**Additional Comments On Reviewer Discussion:**

The discussion and changes are summarised in the meta-review.

---

### Decision · Program_Chairs · 2025-01-22

Accept (Oral)